# German methane fluxes estimated top-down using ICON–ART – Part 1: Ensemble-enhanced scaling inversion

Valentin Bruch[1], Thomas Rösch[1], Diego Jiménez de la Cuesta Otero[1], Beatrice Ellerhoff[1], Buhalqem Mamtimin[1], Niklas Becker[1], Anne-Marlene Blechschmidt[1], Jochen Förstner[1], and Andrea K. Kaiser-Weiss[1]

[1]Deutscher Wetterdienst, Frankfurter Str. 135, 63067 Offenbach

**Correspondence:** Valentin Bruch (valentin.bruch@dwd.de) and Andrea K. Kaiser-Weiss (andrea.kaiser-weiss@dwd.de)

**Abstract.** This two-part study explores the quantification of greenhouse gas emissions using atmospheric observations in order to validate national emission inventories. Inverse methods can support emission quantification at the national scale based on observations and atmospheric transport simulations, yet, they are often limited by the observation coverage, transport model uncertainties, and inversion methodologies. Here, we introduce a system for regional estimation of methane fluxes and apply this to Central Europe with a focus on Germany, where we distinguish emissions from different anthropogenic sectors. We evaluate the robustness of the method using sensitivity tests with in-situ observations from the Integrated Carbon Observation System (ICOS). Using synthetic observation experiments, we estimate the impact of transport errors on the flux estimates. The atmospheric transport is calculated employing the numerical weather prediction model ICON with its module ART at 6.5 km resolution, sampling the meteorological uncertainty with a 12-member transport ensemble. The same transport ensemble is used to generate pseudo-observations with a simulated transport uncertainty. Posterior fluxes are estimated with a synthesis inversion method for three different approximations of the model–observation error covariance matrix. We find that using ensemble-estimated transport uncertainties can significantly reduce the random error of emission estimates. Our results highlight the importance of analyzing biases in flux inversions for reliable, observation-based emission estimates.

## 1 Introduction

Quantifying greenhouse gas (GHG) emissions is essential for effective mitigation of anthropogenic climate change. Atmospheric GHG inversions provide such quantification by connecting the observed atmospheric composition to surface fluxes using transport models. This so-called "top-down" approach is complementary to "bottom-up" emission estimates, which are based on activity data and emission factors (IPCC et al., 2019). Top-down emission estimates can be used to validate national bottom-up GHG inventories reported to the United Nations Framework Convention on Climate Change (UNFCCC) (Manning et al., 2003, 2011; Henne et al., 2016). Such national-scale estimates are typically limited by the observation coverage (Petrescu et al., 2023) and uncertainties in atmospheric transport modeling (Gerbig et al., 2008). This motivates estimating methane emissions in the comparably well-observed Central Europe using a high-resolution transport model and applying methods from numerical weather prediction (NWP) to estimate the transport uncertainty.

Regional top-down estimates of long-lived GHG can be based on different types of transport models. Lagrangian models calculate trajectories from selected locations by moving with air parcels transported by the wind. They have been widely used for inversions of trace gases like halocarbons, nitrous oxide and methane ($CH_4$) in European regions, see e.g., Stohl et al. (2009); Ganesan et al. (2015); Henne et al. (2016). In contrast, Eulerian models – such as ICON–ART – continuously transport trace gas concentrations through three-dimensional grid boxes. Although they are computationally more expensive for cases where a relatively small number of trajectories would suffice, they become superior when the amount of data grows and, as Engelen et al. (2002) pointed out, open the road for data assimilation methods as used in NWP. Among the Eulerian models, also NWP models have been used for regional flux inversions of $CO_2$ (Lauvaux et al., 2013) and $CH_4$ (Steiner et al., 2024b). Regardless whether Lagrangian or Eulerian or even combined approaches (Rigby et al., 2011) are applied, the top-down estimation requires solving an inverse problem (Enting, 2002). Eulerian transport model based inversions may employ emission ensembles, as in Steiner et al. (2024b) with a localized Kalman filter, and other data assimilation methods (see, e.g., Meirink et al., 2008). Alternatively, the method of synthesis inversion scales a set of a priori emission categories (Kaminski et al., 2001).

In this work, we introduce a system for national-scale top-down estimation of $CH_4$ emissions based on modeling experience from NWP. We analyze the benefit of constraining the transport uncertainty using a meteorological ensemble as proposed by Ghosh et al. (2021) and Steiner et al. (2024a). A synthesis inversion method is used to estimate emissions with a focus on Germany based on high-resolution a priori emissions from national reporting and in situ observations of atmospheric $CH_4$ concentrations.

In the present Part 1 of this two-part study, we describe our new inversion system and evaluate its performance. Section 2 introduces the method with a detailed description of the uncertainty estimation. The description of the inversion system is completed by the input data described in Sect. 3. In Sect. 4, we analyze the performance using synthetic observation experiments and test the sensitivity to tuning parameters with real observations. We conclude in Sect. 5 and refer to Part 2 (Bruch et al., 2025a) for a discussion of the emission estimates obtained using real observations.

## 2 Method

We use a synthesis inversion method (Kaminski et al., 2001) that scales the $CH_4$ fluxes to optimize the agreement of model predictions and observations. In this method, the fluxes are initially grouped into a manageable set of flux categories. Here, these are 46 categories that subdivide the fluxes by region and emission sector. With the Eulerian transport model, the concentration from each flux category is calculated separately at all grid cells and time points. At the location and time of the observations, the model writes out the predicted concentrations from the flux category contributions and their sum is compared to the observed concentration. The inversion then minimizes the mismatch between model prediction and observations by scaling each of the flux categories by one number – the scaling factor – making use of the linear relation between fluxes and concentrations in the atmosphere. Thus, the inversion result consists of one scaling factor for each flux category. By multiplying the a priori fluxes with the scaling factors we obtain the a posteriori fluxes. This scaling method cannot provide a correction where a

priori fluxes are zero (Kountouris et al., 2018). However, this is less of a problem for CH$_4$, as inventories can collect where methane-emitting activities are normally located, but emission factors which translate the activities into bottom-up emissions are uncertain (Dammers et al., 2024).

The described method relies on high quality model predictions as well as accurate concentration observations. To match these requirements, we have carefully chosen the configuration of the transport model (Sect. 2.1) and consider the specific difficulties in modeling strong plumes (2.2). Selected observational data are employed to remedy model boundary effects and therefore improve the overall model predictions (Sect. 2.3). In Sect. 2.4, we introduce the Bayesian inversion framework. To assess whether deviations between model and observations contain information on the fluxes, we estimate the model uncertainty

and error correlations. We compare three different methods for estimating these uncertainties and correlations (Sections 2.5 and 2.6). Furthermore, we define the time window and a priori uncertainties of the inversion (Sections 2.7 and 2.8). A summary of the method and data streams will be provided in Sect. 3.5.

## 2.1   Transport simulation

### 2.1.1   Transport model

The atmospheric transport is simulated using the NWP model ICON (Zängl et al., 2015) in a configuration close to operational NWP at Germany's Meteorological Service (DWD), extended with the module for Aerosol and Reactive Trace gases (ART) (Rieger et al., 2015; Schröter et al., 2018). The model is run in limited area mode for a domain covering large parts of the European continent (latitudes $34°\,N$ to $70°\,N$, longitudes $21°\,W$ to $59°\,E$, see Fig. 1) with a horizontal resolution of $6.5\,km$ (ICON grid R3B8) and 74 vertical levels up to a maximal height of $22.77\,km$. The ICON model simulates the meteorology

and the tracer transport. Re-initialization of the meteorological fields every $24\,h$ with operationally produced analysis fields ensures that the meteorology stays close to reality. The surface CH$_4$ fluxes are provided to the transport model using the online emission module (Jähn et al., 2020; Steiner et al., 2024b). We do not simulate any chemical reactions, because the typical lifetime of CH$_4$ in the atmosphere is much longer than the time that an air parcel typically spends in our modeling domain.

    For long living tracers like methane, the correct treatment of the lateral boundary concentrations is of importance. Therefore,

we extended the model by implementing lateral boundary nudging for ART tracers in order to obtain smooth fields and avoid strong spatial gradients. The nudging is limited to a boundary zone of width $< 250\,km$. Further, so-called meteogram output has been implemented for ART tracers, providing model output in the vicinity of observation locations with high temporal resolution.

### 2.1.2   Meteorological ensemble

For improved uncertainty estimates, we run a meteorological ensemble of 12 members. Each ensemble member uses different meteorological initial and lateral boundary conditions from the operational ensemble data assimilation used for global NWP at DWD (Schraff et al., 2016; Reinert et al., 2025). Since our meteorological input fields and the transport model setup are taken

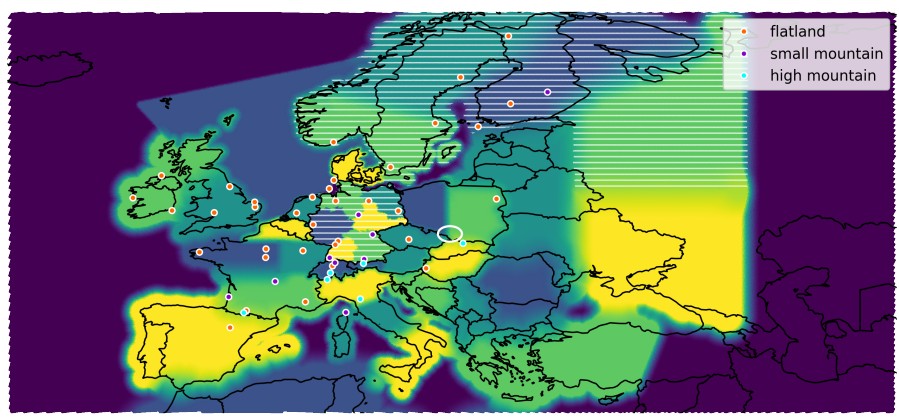

**Figure 1.** Model domain, colored to distinguish 35 patches defining regional flux categories. Observation sites (dots) are colored by the choice of model equivalent height (see Table C1). Dark blue at the domain boundary indicates regions for which emissions are not categorized and therefore not modified in the inversion. Other colors only distinguish neighboring patches. In white hatched regions, natural fluxes are also categorized and scaled. A white ellipse marks the Upper Silesian Coal Basin, in which fugitive emissions define their own flux category. In Germany, the map shows the six regions used for the agricultural sector. For other sectors in Germany, we use four regions: south (yellow and light green), west (dark blue), north (light green), and east (dark green and yellow).

from operational NWP at DWD, the ensemble provides a reasonable estimate for the meteorological uncertainty in our model, including uncertainties in the simulated wind field and atmospheric stability.

In the following, we distinguish a so-called deterministic model run providing the best estimate of the modeled $CH_4$ concentration, and the ensemble runs providing 12 different $CH_4$ concentrations to estimate the uncertainty. The ensemble will only be used to estimate model uncertainties and error covariances (see Sect. 2.5), and to generate pseudo-observations (Sect. 3.4).

### 2.1.3 Definition of flux categories

Estimating $CH_4$ fluxes in $> 10^5$ grid cells based on 50 observation sites seems impossible without reducing the number of
degrees of freedom of the fluxes. Here, we reduce the degrees of freedom drastically by parametrizing the fluxes using only 46 basis vectors. A basis vector in this parametrization is a flux category that contains all fluxes from one region, possibly limited to specific emission sectors. For example, we define all anthropogenic emissions from Denmark as one flux category. We thereby assume that the distribution of anthropogenic emissions within Denmark is correct in the a priori and only allow the inversion to adjust the total emissions from Denmark.

We define the flux categories with the primary aim of providing an accurate estimate of emissions from Germany, resolving federated states where possible, to address the requirements of potential stakeholders. When distinguishing emission sectors, we stay close to the national reporting by using definitions from the gridded aggregated nomenclature for reporting (GNFR, Veldeman et al., 2013). For the agricultural sector (GNFR sectors K+L), which contributes roughly two thirds of all German $CH_4$ emissions, we distinguish six regions within Germany as depicted in Fig. 1. For the sum of all other sectors –

**Table 1.** Overview of sectors distinguished in the inversion and number of flux categories. We distinguish the focus region, well-observed regions near the focus region, and regions in large distance from the focus region ("remote"). The latter are split in very large flux categories with low a priori uncertainty. Natural plus LULUCF fluxes are separated from other anthropogenic emissions only in regions where the natural fluxes are strong and in Germany. One extra category in the well-observed regions is the Upper Silesian Coal Basin (marked* in the last column). See Fig. 1 for the definition of flux categories on the map.

| Classification | Countries and regions | Sectors | # of areas | # of flux categories |
|---|---|---|---|---|
| focus region | Germany | agriculture, LULUCF + natural, other | 6 agr., 4 other, 1 LULUCF | 11 |
| focus region | Netherlands | agriculture, other | 1 | 2 |
| well observed | Sweden, Norway | LULUCF + natural, anthropogenic | 2 | 4 |
| well observed | DK, PL, CZ, AU, SK, HU, SV, HR, BA, CH, FR, BE, LU, UK, IE, northern IT, North Sea | anthropogenic (excl. LULUCF) | 16 | 17* |
| remote | Finland, north-western Russia | LULUCF + natural, anthropogenic | 2 | 4 |
| remote | other | anthropogenic (excl. LULUCF) | 8 | 8 |

excluding natural and LULUCF fluxes – we distinguish four regions, i.e., the federated states south: Baden-Wuerttemberg and Bavaria, west: North Rhine-Westphalia, Hesse, Rhineland-Palatinate and Saarland, north: Lower Saxony, Bremen, Hamburg and Schleswig-Holstein, as well as east: Mecklenburg-Western Pomerania, Brandenburg, Berlin, Saxony, Saxony-Anhalt and Thuringia. Natural plus LULUCF fluxes in Germany are treated as a single flux category.

Outside Germany, we do not distinguish sectoral emissions, with one exception. Agriculture emissions in the Netherlands form their own category, as we found that they strongly influence the $CH_4$ concentrations in Germany, caused by the proximity and high emission rates in the Netherlands. We define further categories by area for anthropogenic emissions excluding LULUCF such that a comparably high resolution is obtained in regions near Germany with high observation coverage. These area-defined flux categories follow borders as feasible for the inversion. Areas with small expected influence on inversion results for Germany are combined in large categories, such as Spain plus Portugal, Türkiye plus Greece, and large areas east of Poland. All area-defined categories are shown in Fig. 1 and an overview of the sector resolution is given in Table 1.

We treat natural plus LULUCF fluxes separately and categorize them only in Germany, Scandinavia, and the north-eastern part of our domain (hatched regions in Fig. 1). This is motivated by strong $CH_4$ emissions from wetlands in summer in Scandinavia and northern Russia in our prior (Segers and Houweling, 2020). Uncategorized fluxes – whether natural or anthropogenic – are not scaled in the inversion, but still included in the transport simulation such that no fluxes are discarded. To avoid strong spatial gradients in the concentration fields, the boundaries between different area-defined categories are smoothened as visualized in Fig. 1.

We furthermore define a separate flux category for the strongest $CH_4$ plume in Central Europe to mitigate the plume localization problem described below (Sect. 2.2). These are fugitive emissions from the Upper Silesian Coal Basin with yearly emissions of $567\,kt$ in our prior (white ellipse in Fig. 1).

### 2.1.4 Tracer assignment in the transport model

In the transport simulation, we consider not only the categorized fluxes, but also the $CH_4$ from lateral boundaries and from uncategorized emissions. Overall, we simulate the transport of 50 tracer fields in the deterministic model run:[1]

(i) **Sum of all anthropogenic emissions excluding LULUCF.** This constitutes a single, common tracer.

(ii) **Sum of all natural plus LULUCF fluxes.** This constitutes another single, common tracer, which summed with (i) covers all a priori emissions in the domain.

(iii) **Far field.** The far field contains the $CH_4$ from initial and lateral boundary conditions.

The sum of (i)–(iii) is the total a priori $CH_4$ concentration. The a posteriori concentration is not computed directly. Instead, we treat the deviation of the posterior concentration from the prior as a perturbation. To compute this perturbation, we simulate the transport of each flux category:

(iv) **Flux categories.** For each of the 46 flux categories an own tracer field is defined. To avoid the accumulation of categorized $CH_4$ beyond the time scale on which we consider the modeled transport reliable, we set an artificial decay rate of these concentrations. After emission, the concentration in these tracer fields decays exponentially with a mean lifetime of five days. This technical feature constitutes a localization in time similar to the commonly used localization in space (e.g., Steiner et al., 2024b) and allows a waning of sectoral and regional attribution over a few days. This regulates that any attribution of a $CH_4$ anomaly to a certain region or sector is only attempted if the emission was fresh or a few days ago. Furthermore, this allows us to save computing time by limiting the transport of these flux category tracer fields to altitudes below $8\,km$. The artificial decay rate affects the posterior concentration and the sensitivity of the inversion to changes in the emissions. However, assuming that the typical time between emission and observation is short compared to the artificial lifetime and in the presence of transport model errors, we expect that this feature of our inversion system leads to more robust results.

(v) **Auxiliary field for plume detection.** For the purpose of investigating the model uncertainty due to the plume from the Upper Silesian Coal Basin, an auxiliary tracer is added (see Sect. 2.6.1). This tracer is never added to the total $CH_4$ concentration but only serves as an indicator for the plume location.

---

[1]Technically, the simulation includes 58 tracers in an attempt to split up the sector "other" in Germany in three sectors. Since we do not use these additional data here, we describe the setup for the 50 tracers we actually used.

## 2.2 Plume localization problem

In our transport simulation and inversion, we address the specific challenge posed by plumes from high emissions in small areas. The inversion may be biased for such plumes due to the so-called double penalty issue (Vanderbecken et al., 2023). In cases where our model falsely predicts that the plume reaches an observation site, the inversion will reduce the emissions to improve the agreement with the observation. In the opposite case, when the model fails to predict that a plume reaches the observation, the inversion will not change the plume emission amount but will wrongly increase emissions in other areas instead. This can cause a systematic underestimation of fluxes from localized plumes. To avoid biases in the inversion results, we suggest to treat strong plumes separately, with their own flux categories. This allows us to quantify the problem (see Sect. 4.2) and to limit the plume penalty influence on other flux categories.

## 2.3 Far-field correction

For cases where the model predicts almost no influence from our categorized emissions (i.e., clean air cases), deviations between model and observations point to the need for correcting the $CH_4$ advected across the lateral boundaries – here referred to as "far field".[2] For our regional inversion problem, it is essential to separate the $CH_4$ emitted within the domain from the far field, in order to avoid model biases which would confound the aspired flux scaling (see, e.g., Chen et al., 2019, for $CO_2$). To minimize potential biases arising from imperfect boundary conditions, we construct a correction field which is added to the modeled far-field concentration in the whole domain after the transport simulation. We require this correction field to be smooth on spatial and temporal scales $320\,\mathrm{km}$ (horizontal), $1\,\mathrm{km}$ (vertical), and $16\,\mathrm{h}$ (time). We construct this far-field correction using a Kalman smoother as described in detail in Appendix A. This construction uses only clean-air observations with a cumulated signal of all flux categories of $\leq 20\,\mathrm{ppb}$ and a total signal from emissions within our domain of $\leq 50\,\mathrm{ppb}$.

Figure 2 shows a statistical overview of the far-field correction when using real observations (red line) or pseudo-observations (shaded area). The considered pseudo-observations are generated from the ensemble members of the transport simulation and represent the case where simulated emissions and boundary conditions are perfect, i.e., equal to the truth. The far-field correction range is usually limited to $\pm 10\,\mathrm{ppb}$ when using real observation data and $\pm 5\,\mathrm{ppb}$ in the synthetic observation experiments (Fig. 2 a) with variations of a few ppb per day (Fig. 2 b). The broad distribution of the root mean square (RMS) for different observation sites and months in Fig. 2 (c) indicates significant differences among the stations when using real observations.

Figure 2 (d) shows that the correction has a small bias towards positive corrections even when using synthetic observations with unbiased fluxes and boundary conditions. This is partially due to the pseudo-observations, which are biased by $+0.5\,\mathrm{ppb}$ compared to the simulated concentrations due to details of the transport model configuration. The other part of the bias hints to a more general problem. We construct the far-field correction using observations for which the model predicts clean air, i.e., a low signal from the emissions. Since the transport model is not perfect, this introduces a sampling bias: We select more

---

[2]Technically, the far field also includes the initial $CH_4$ concentration. But this is hardly relevant due to our generous spin-up period of 17 days.

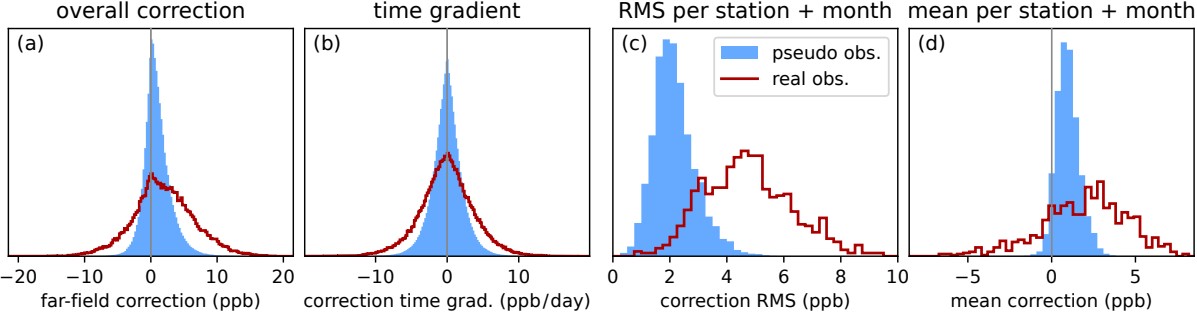

**Figure 2.** Statistical evaluation of the far-field correction at the observation coordinates when using synthetic observations (light blue area) or real observations (dark red line). Considering all data points used in the inversion, histograms of the far-field correction show (a) the range of the correction and (b) its temporal variation. For each station, month, and realization of pseudo-observations, we compute the root mean square (RMS) and the mean (or bias). Histograms combining these values for all stations and months are shown in (c) and (d).

observations for which the model underestimates the concentrations and thereby increase the bias to $1.2\,\mathrm{ppb}$. In response to this bias, the far-field correction increases the simulated concentrations by $1.0\,\mathrm{ppb}$.

The sampling bias will likely also occur when working with real observations. But the estimated correction bias of $0.6\,\mathrm{ppb}$ due to the sampling is small compared to the accuracy of the Copernicus Atmosphere Monitoring Service (CAMS) inversion-optimized data product used for our boundary conditions (Segers et al., 2023) (see Sect. 3.1). We therefore do not expect a significant impact on the emission estimates.

## 2.4 General approach of the inversion framework

We use a Bayesian inversion to optimize the agreement of model and observations. We define a vector of scaling factors – in our application $s \in \mathbb{R}^{46}$ – consisting of one prefactor for each flux category. This low-dimensional parametrization of the fluxes leads to the optimization problem

$$s^{\mathrm{post}} = \underset{s}{\arg\min}\left\{\tfrac{1}{2}(y - Hs - x^{\mathrm{ff}})^{\top} R^{-1}(y - Hs - x^{\mathrm{ff}}) + \tfrac{1}{2}(s - s^{\mathrm{prior}})^{\top} B^{-1}(s - s^{\mathrm{prior}})\right\} \tag{1}$$

for the posterior scaling factors $s^{\mathrm{post}}$. Here, the first term penalizes the deviation from the observed concentrations, and the second term penalizes the deviation from the prior fluxes. In the first term, the vector $y$ of observed concentrations is compared to the model prediction, which consists of the contribution $Hs$ of fluxes within the model domain and the modeled far field $x^{\mathrm{ff}}$ including the far-field correction. All model predictions ($x^{\mathrm{ff}}$ and $Hs$) are already projected to the observation space. The contribution of fluxes $Hs$ depends linearly on the vector $s$. The difference between modeled and observed values is weighted by the error covariance matrix $R$ describing the combined uncertainty of the transport model and the observations. With the second term we constrain the deviation of $s$ from a priori scaling factors $s^{\mathrm{prior}}$ ($s_k^{\mathrm{prior}} = 1$ for all $k$) with an error covariance matrix $B$ characterizing the a priori uncertainty (see Sect. 2.8).

In Eq. (1), the model observation operator $H$ connects the space of scaling factors (vectors $s^{\text{prior}}$, $s^{\text{post}}$) to the observation space (vectors $y$, $x^{\text{ff}}$). Computing $H$ requires the transport model which distinguishes the flux categories. The setup is designed for optimizing a low-dimensional vector $s^{\text{post}}$ of scaling factors ($\sim 10^2$ degrees of freedom) using a large number of observations ($\sim 10^4$), but an extension to more degrees of freedom and/or more observations is possible.

## 2.5 Approximations for the error covariance matrix $R'$

The definition of the error covariance matrix $R$ in Eq. (1) is crucial for the inversion. $R$ describes the combined uncertainties and correlations of observations and model predictions. In our case, the observation uncertainty (usually $\lesssim 1\,\text{ppb}$, ICOS RI (2020)) is small compared to the ensemble-estimated transport uncertainty (typically $5\,\text{ppb}$ to $10\,\text{ppb}$). We therefore focus on the model uncertainty.

Many works have used diagonal $R$ matrices (e.g. Bergamaschi et al., 2010; Petrescu et al., 2023; Steiner et al., 2024b) and others found non-diagonal approximations for $R$ (Ghosh et al., 2021; Steiner et al., 2024a). Here, we use the diagonal $R$ for comparison to two different ways of constructing a non-diagonal $R$ matrix from our transport ensemble. We therefore compare three ways of constructing $R$:

**Diagonal $R$:** This baseline scenario considers a diagonal $R$ matrix and discards all information from the transport ensemble.

**Prior $R$:** In a standard ensemble approach, we construct $R$ using the transport ensemble with a priori fluxes.

**Posterior $R$:** We extend the standard approach by estimating $R$ using the posterior fluxes in the transport ensemble.

The construction of the different $R$ matrices consists of two steps that are described below. First, we construct a matrix $R'$ that estimates the dominant uncertainties and correlations using one of the three methods. Second, we obtain $R$ from $R'$ by inflating and adding additional uncertainties to mitigating some known issues of the inversion (Sect. 2.6).

### 2.5.1 Diagonal $R$

In the baseline scenario of a diagonal $R$ matrix, all observation and model uncertainties are assumed to be uncorrelated. However, it is known that model predictions for observations separated by only one hour usually have correlated errors. To avoid underestimating the overall uncertainty without introducing correlations in $R$, we assume high uncertainties of each observation. Following Steiner et al. (2024b), we assume that the signal from $CH_4$ emissions within our domain will generally increase the model uncertainty in the predicted $CH_4$ concentration. This motivates defining $R'_{ii} = \sigma_{\text{const}}^2 + (\beta H s^{\text{prior}})_i^2$ where $\sigma_{\text{const}} = 10\,\text{ppb}$ and $\beta = 0.5$ are scalar tuning factors. Index $i$ labels observation data points that are typically distinguished by location, time, and sampling height. The diagonal $R$ scenario uses crude approximations because the selection of observations is designed for an inversion that can handle correlations. However, we will obtain qualitative insights from the comparison to the other approximations for $R$.

### 2.5.2 Prior $R$

This approximation of $R$ is based on an ensemble of $M = 12$ different transport realizations. The potential of using a small transport ensemble for estimating model uncertainties was demonstrated by Steiner et al. (2024a). We can use the covariance of the ensemble members to estimate the transport uncertainty. We define

$$R'_{ij} = C_{ij} \frac{1}{M-1} \sum_{m=1}^{M} (x_i^m - \bar{x}_i)(x_j^m - \bar{x}_j) + \delta_{ij} \sigma_{\text{const}}^2, \tag{2}$$

where $x_i^m$ is the prediction of ensemble member $m$ for observation $y_i$ assuming a priori fluxes, $\bar{x}_i = \frac{1}{M} \sum_m x_i^m$ is the ensemble mean, and $\sigma_{\text{const}} = 10\,\text{ppb}$ is a constant uncertainty added to each observation. With this uncorrelated uncertainty $\sigma_{\text{const}}$, we account for additional uncertainties, such as representativity errors inherent to a simulation at finite resolution. Indices $i, j$ label observation data points. By $C_{ij}$ we denote a localization in space and time such that $C_{ii} = 1$ and $C_{ij} = 0$ for any observations $i$ and $j$ that we expect to be uncorrelated because of their temporal or spatial separation. In the application to Germany, we choose $C_{ij}$ to be a Gaussian localization matrix with standard deviations $6\,\text{h}$ (time), $319\,\text{km}$ (horizontal), and $400\,\text{m}$ (vertical). We use the notation $\delta_{ij} = 1$ if $i = j$ and $\delta_{ij} = 0$ if $i \neq j$.

### 2.5.3 Posterior $R$

The posterior $R$ approximation is a variation of the prior $R$ approximation. In Eq. (2), we use model predictions for the concentrations $x_i^m$. Instead of using the prior concentrations as in the prior $R$ construction, we can define $x_i^m$ as the posterior concentrations and thereby allow $x_i^m$ to change as the inversion changes the fluxes. This leads to a self-consistent estimate of $R'$ in the inversion. Consequently, Eq. (2) remains valid but $x_i^m$, $R'$, and $R$ become functions of the scaling factors $s$. Since $R$ is estimated using posterior scaling factors, we call this method the posterior $R$ inversion as opposed to the prior $R$ estimate. To compute the posterior concentration $x_i^m(s)$ for each ensemble member without prohibitive computational effort, we use an approximation described in Appendix B.

As opposed to the diagonal $R$ and prior $R$ inversion with fixed $R$, the posterior $R$ inversion does not allow for a closed form solution of Eq. (1). To solve the minimization problem in Eq. (1) numerically, we used SciPy's "trust-exact" implementation of a trust-region method (Virtanen et al., 2020; Moré and Sorensen, 1983; Conn et al., 2000). Within each iteration, the incomplete LU decomposition (Li et al., 1999; Li and Shao, 2011) of the sparse matrix $R(s)$ is the most computationally expensive task when the number of observations is large.

### 2.6 Additional uncertainties and final error covariance matrix $R$

The previously derived approximations for the error covariance matrices $R'$ describe our knowledge of the transport uncertainty and the observation uncertainty. In the next four steps, we increase uncertainties and include other possible sources of uncertainty to obtain approximations for $R$ that are suitable for the inversion.

### 2.6.1 Mitigating the plume localization problem

To reduce the bias which we predicted for strong plumes in Sect. 2.2, we increase the uncertainty for all observations that are likely affected by a plume. The transport ensemble will already lead to an increased uncertainty when the model cannot predict reliably whether a plume hits an observation site. But with an ensemble of only 12 members, this will not cover all cases where model and observations deviate. We therefore introduce an auxiliary tracer that contains emissions from the Upper Silesian Coal Basin, spatially smoothened on a length scale of $0.4°$ (one standard deviation of a Gaussian filter). Denoting the concentration of this tracer at observation $i$ by $\rho_i$, we increase the uncertainties to $R^{\text{step 1}} = R'_{ij} + 0.25\rho_i^2\delta_{ij}$.

### 2.6.2 Dynamic uncertainty inflation

To avoid potential biases through site-specific small-scale features not captured in the model, we aim to base our inversion on many observations. To this end, we limit the influence of individual data points on the inversion result by inflating the uncertainty further in the case of a very large disagreement between model and observation. This is achieved by an uncertainty inflation of individual observations until the deviation $\mu = y - Hs^{\text{prior}} - x^{\text{ff}}$ between model and observations is at most three standard deviations of the resulting error covariance matrix $R_{ij}^{\text{step 2}} = g_i g_j R_{ij}^{\text{step 1}}$, i.e., $g_i = \max\{1, \frac{|\mu_i|}{3\sqrt{R_{ii}^{\text{step 1}}}}\}$. This is justified because large deviations between model and observations, $|\mu_i| > 3\sqrt{R_{ii}^{\text{step 1}}}$, are likely caused by local pollution or modeling problems that are not captured appropriately in our uncertainty estimate. This correction makes sure that inversion results will be based on many observations and no single measurement can have an extreme impact. At the same time, this method it is less sensitive to tuning parameters than discarding outliers completely.

### 2.6.3 Static uncertainty inflation

The transport ensemble in the prior $R$ and posterior $R$ construction may not necessarily include the full uncertainty of the transport model, and the localization $C_{ij}$ further reduces the simulated uncertainty by suppressing correlations. This motivates another inflation of the uncertainty to avoid overconfidence in the model prediction. We inflate the uncertainty by a factor $f_i > 1$ depending on the observation site of observation $i$, leading to $R_{ij}^{\text{step 3}} = f_i f_j R_{ij}^{\text{step 2}}$. We choose $f_i = 2$ except for some stations with known difficulties, for which $f_i = 3$ (see Table C1). To keep the methods for constructing $R$ comparable, we apply this inflation also to the diagonal $R$ matrix.

### 2.6.4 Far-field uncertainty

We furthermore account for the uncertainty in the far-field correction, although the effect of this additional uncertainty is small. We define $R_{ij} = R_{ij}^{\text{step 4}} = R_{ij}^{\text{step 3}} + 0.5|c_i c_j|\tilde{C}_{ij}$ where $c_i$ denotes the smooth correction field introduced in Sect. 2.3 at observation $i$ and $\tilde{C}_{ij}$ is the Gaussian localization matrix constructed by the length and time scales of the far-field correction (see Appendix A).

**Table 2.** Median of $\chi^2/N_{\text{dof}}$ for different configurations. $\chi^2/N_{\text{dof}}$ for the prior $R$ inversion also serves as an approximation for the posterior $R$ inversion. Synthetic observations are generated using the ensemble simulation, assuming that the a priori fluxes and the $CH_4$ concentration on lateral boundaries are known exactly.

| Observations | Far-field correction | $\chi^2/N_{\text{dof}}$, diagonal $R$ | $\chi^2/N_{\text{dof}}$, prior $R$ |
|:---:|:---:|:---:|:---:|
| real | yes | 0.18 | 0.16 |
| real | no | 0.21 | 0.18 |
| synthetic | yes | 0.05 | 0.03 |
| synthetic | no | 0.06 | 0.03 |

### 2.6.5 $\chi^2$ analysis

To assess whether the estimated uncertainties are reasonable, one can compute the $\chi^2/N_{\text{dof}}$ value (Pearson, 1900). This value compares the a priori model–observation mismatch to the uncertainty assumed for this mismatch (see Appendix D for details). A value of $\chi^2/N_{\text{dof}} > 1$ indicates that uncertainties are underestimated, whereas values smaller than one indicate the opposite. When comparing the observations to the far-field-corrected model, we find $\chi^2/N_{\text{dof}} \approx 0.16$ for the prior $R$ inversion when using real observations (see Table 2). In an idealized setup, this indicates that the uncertainties of the model-data mismatch are overestimated by a factor 2.5. This implies that our uncertainty inflation by a factor $f_i = 2$ for most observations seems unnecessary in the idealized setup. However, our data can contain unknown biases in transport and boundary conditions, and simplifying assumptions about the representativity of the low-dimensional state space of the inversion. We contain these potential issues of unknown error components by inflating the uncertainties.

In the synthetic experiments, the idealized transport uncertainty and perfect a priori emissions lead to even lower $\chi^2$, which is expected because not all uncertainties are contained in the pseudo-observations of these synthetic experiments. Computing $\chi^2/N_{\text{dof}}$ for the posterior $R$ inversion is more difficult, but the result is expected to be similar to the prior $R$ inversion. The tuning parameters of the diagonal $R$ matrix were chosen such that the posterior uncertainties are similar to the prior $R$ inversion, which also leads to similar $\chi^2/N_{\text{dof}}$ (see Table 2).

### 2.7 Inversion time window and temporal aggregation

We simulate the transport for the whole year 2021 without any interruption. The inversion is then applied to each month separately by selecting only observations within one month. The scaling factors of the months are treated as independent, each month starting with the same a priori scaling factors ($s_k^{\text{prior}} = 1$ for all $k$) and the same a priori scaling uncertainties ($B$ matrix). The continuous transport simulation over the whole year implies that the initial $CH_4$ concentration is hardly relevant after the spin-up. At the beginning of each month, the modeled $CH_4$ concentration already consists of the far field – the contribution of the lateral boundaries – and the contribution of the fluxes, which will be adjusted by the inversion.

In summary, we correct the contribution of the lateral boundaries on the time scale of $16\,\mathrm{h}$ by the far-field correction, and the fluxes on the time scale of one month defined by the inversion time window. The inversion results consist of one vector $s^{\mathrm{post}} \in \mathbb{R}^{46}$ of scaling factors and the corresponding error covariance matrix for each month. When aggregating results for the whole year, we treat the uncertainties of the prior or posterior fluxes of different months as correlated because these likely include systematic uncertainties and biases which we cannot fully separate from the statistical uncertainty. We therefore aggregate by adding up absolute emissions and their uncertainties linearly.

## 2.8 Prior uncertainties

In each inversion time window, we consider a priori scaling factors with a two standard deviation ($2\sigma$) uncertainty of $0.8$ for most flux categories, corresponding to a $95\,\%$ confidence interval of $\pm 0.8$. Throughout this paper, uncertainties will denote two standard deviations or $95\,\%$ confidence intervals. Categories resolving emission sectors have a higher prior $2\sigma$ uncertainty of $1.0$, and within Germany categories describing the same sector have an a priori uncertainty correlation of $0.5$ (e.g., uncertainties of agriculture emissions in the German states of Bavaria and Baden-Wuerttemberg are assumed to be correlated). All other categories are treated as uncorrelated in the a priori. For the Upper Silesian Coal Basin as well as regions with low observation density outside of our primary focus in Central Europe (marked "remote" in Table 1), the $2\sigma$ uncertainty is set to $0.5$.

## 3 Input data and processing

We apply the method to estimate $CH_4$ fluxes in the year 2021 in Germany and in the surrounding European domain, relying on input data for the transport simulation, $CH_4$ concentration on the lateral boundary (Sect. 3.1), a priori fluxes (Sect. 3.2), and observations (Sect. 3.3).

### 3.1 Initial and lateral boundary conditions

The meteorological initial and lateral boundary conditions used to drive our transport model are taken from the archive of DWD's operational NWP, which also employs the ICON model. As we do not assimilate meteorological data in our application, we re-initialize the meteorological fields every night at 0 UTC, using the analysis fields from the operational NWP data assimilation. Lateral boundary conditions for the meteorological fields are taken from the NWP short term forecasts with hourly resolution.

For the $CH_4$ concentrations, we use initial and lateral boundary concentrations from the CAMS global inversion-optimized dataset (Segers and Houweling, 2020), version v22r2, in the variant based on surface air-sample data for the inversion. The CAMS data have a resolution of $1° \times 1°$ and are interpolated onto our model grid. In contrast to the meteorological fields, the $CH_4$ concentrations are only transported and never re-initialized. Each transport ensemble member uses slightly different initial and lateral boundary conditions for meteorological fields (see Sect. 2.1.2), but equal $CH_4$ concentrations on the lateral boundaries.

**Table 3.** Input data for a priori $CH_4$ fluxes. The second column lists where these fluxes were considered. Here, "Germany" refers to all model grid cells that lie fully within the German borders. The national reporting distinguishes emissions by GNFR sectors of which A–M include all anthropogenic emissions excluding land use, land use change and forestry (LULUCF).

| Data provider | Domain | Fluxes | Original grid | Time profile | Remarks |
|---|---|---|---|---|---|
| Umweltbundesamt (UBA) | Germany | GNFR sectors A–M | native (ICON) | constant | Based on reporting to the UNFCCC (UBA, 2023), spatially distributed using the Gridding Emission Tool for ArcGIS (GRETA 1.2.01) (Feigenspan et al., 2024) |
| Thünen Institute | Germany | organic and mineral soils (part of LULUCF) | 100 m × 100 m | constant | Emissions from organic and mineral soils, including wetlands but excluding artificial ponds (approx. 160 kt $CH_4$ per year) (Fuß and Akubia, 2024) |
| CAMS-REG-ANT, v7.0 | model domain excl. Germany | GNFR sectors A–M | $0.05° \times 0.1°$ | constant | Based on data reported to the UNFCCC for countries in Western and Central Europe (incl. Finland and the Baltic states) (Kuenen et al., 2021, 2022) |
| CAMS inversion optimized, v22r2 | model domain excl. Germany, excl. oceans | wetlands | $1° \times 1°$ | monthly averages | Variant using surface air-sample data for the inversion (Segers and Houweling, 2020); Fluxes in model grid cells located over the ocean are set to zero. |
| Rocher-Ros et al. (2023), version 1.1 | full model domain | rivers and streams | $0.25° \times 0.25°$ | monthly averages | |
| Weber et al. (2019) | oceans (full model domain) | oceans | $0.25° \times 0.25°$ | constant | |

## 3.2 A priori $CH_4$ fluxes

For the inversion, we employ a priori $CH_4$ fluxes that were compiled from six datasets of anthropogenic and natural fluxes, as detailed in Table 3. We ensured mass conservation when interpolating to our model grid. We generally distinguish between anthropogenic emissions excluding LULUCF, and natural fluxes plus LULUCF. Since the input datasets for anthropogenic emissions are based on reporting to the UNFCCC, these distinguish between GNFR sectors following the reporting conventions (Veldeman et al., 2013). For the inversion, we combine these sectors and only distinguish between agriculture and the sum of all other sectors as described in Sect. 2.1.3. Natural plus LULUCF fluxes of $CH_4$ are mostly dominated by wetland emissions, for which we do not distinguish between natural and anthropogenic origin.

For Germany, we obtained a priori fluxes directly from the national inventory agencies. The a priori LULUCF fluxes obtained from the Thünen Institute cover the emissions from mineral and organic soils. Notably, this excludes emissions from artificial water bodies in Germany – such as ponds – amounting to $160\,kt$ or $8.5\%$ of the total German emissions in the national reporting, though these numbers are associated with large uncertainties (UBA, 2024, Table 399). These emissions are missing in our a priori estimate, leading to a low bias in the a priori.

### 3.3 Observations and pre-processing

We compare our model predictions to the high quality ground-based in situ observations of $CH_4$ concentrations collected in the European Obspack (ICOS RI et al., 2024), which includes the ICOS stations among others. These observations are assumed to be representative for a larger area (Storm et al., 2023). Table C1 lists all 53 available stations and Fig. 1 shows 50 stations that were used for the inversion. For tower observations, we use up to three sampling heights per station, preferring the highest three sampling heights and discarding observations below $50\,m$ above ground level to reduce the influence of very local emissions. Due to significant model–observation mismatch, we exclude the IPR, FKL and LMP stations. For LUT, BIR and HUN we only consider some seasons, specified in Table C1.

The model data are interpolated horizontally and vertically to the station sampling locations. The vertical sampling locations in model coordinates are derived from the station sampling heights and the modeled station elevations, depending on the station characteristics (column "mountain" in Table C1). For high mountain stations, the modeled station elevation is given by the real station elevation above mean sea level. For stations on smaller mountains, we consider the arithmetic mean between real station elevation and model topography as proposed by Brunner et al. (2012) and Henne et al. (2016), and for all other stations the modeled station elevation is set to the model topography.

To make use of observations which are likely well represented by the model, we filter the observations based on the local time of day, wind speed, and model–data mismatch. Table 4 lists how the root mean square error (RMSE) of the model output changes during these pre-processing steps. We start by smoothing both observations and modeled concentrations in a time window of approximately $\pm 1.5\,h$ around each observation time as depicted in Fig. 3. This allows for some uncertainty in the timing of modeled tracer transport. The resulting correlation of neighboring time steps is automatically considered in the ensemble-based uncertainty estimate.

In the next steps, we filter the data by time in order to keep only observations expected to be representative for large regions. Observations within the planetary boundary layer are most representative in the afternoon hours whereas measurements at high mountains are less influenced by very local fluxes at night time. Inversions therefore commonly use afternoon observations for flat land stations and night times at mountain sites (Bergamaschi et al., 2015; Steiner et al., 2024b). We use the time windows $23\,h$ to $5\,h$ (local mean time) for stations on high mountains and $11\,h$ to $17\,h$ for all other stations.

We furthermore exclude times with no wind to avoid a strong influence of local emissions that are not resolved in the model, motivated by Ganesan et al. (2015). All data points for which the model predicts a wind speed of $< 2\,ms^{-1}$ are excluded, which improves the overall agreement of model and observations as shown in Table 4 (step 4). Figure 4 shows that the RMSE

**Table 4.** Average root mean square error (RMSE in ppb), mean absolute bias of the model prediction minus observation (in ppb), and number of available data points after each processing step (1–6) for synthetic (left) and real observations (right). Each row adds a processing step to all previous steps and improves the RMSE. Three numbers for steps 7 and 8 distinguish diagonal $R$, prior $R$, and posterior $R$ inversion. Step 7 (uncertainty weighting) is not a processing step in the inversion since it uses only the diagonal of the uncertainty matrix $R$, but it underscores the importance of accurate uncertainty estimation. Step 8 refers to the result of the inversion. RMSE and absolute bias are computed separately for each station, sampling height and month. The obtained values are weighted by the number of data points and averaged. By taking the mean of multiple RMSEs for different stations, sampling heights and months, we obtain lower numbers than for the RMSE of the combined dataset, which would average squared values and thereby would give higher weight to large deviations between model and observations.

| | Step | Synthetic observations (ppb) | | | Real observations (ppb) | | |
|---|---|---|---|---|---|---|---|
| | | RMSE | Absolute bias | Data points | RMSE | Absolute bias | Data points |
| 1 | horizontal and vertical interpolation | – | – | – | 27.6 | 9.6 | $6.02 \cdot 10^5$ |
| 2 | time average (3 h) | 11.1 | 0.9 | $6.02 \cdot 10^5$ | 25.8 | 9.6 | $6.02 \cdot 10^5$ |
| 3 | time window $11\,\mathrm{h}-17\,\mathrm{h} / 23\,\mathrm{h}-5\,\mathrm{h}$ | 10.2 | 1.1 | $1.48 \cdot 10^5$ | 23.5 | 9.8 | $1.48 \cdot 10^5$ |
| 4 | minimal wind speed $2\,\mathrm{m\,s^{-1}}$ | 9.6 | 1.0 | $1.30 \cdot 10^5$ | 22.4 | 9.7 | $1.30 \cdot 10^5$ |
| 5 | exclude extreme deviations | 9.6 | 1.0 | $1.30 \cdot 10^5$ | 21.5 | 9.4 | $1.29 \cdot 10^5$ |
| 6 | far-field correction | 9.0 | 0.9 | $1.30 \cdot 10^5$ | 19.4 | 7.2 | $1.29 \cdot 10^5$ |
| 7 | weight by inverse uncertainty | 7.1, 6.9, 6.9 | 0.7, 0.8, 0.8 | $1.30 \cdot 10^5$ | 14.4, 16.6, 16.6 | 5.7, 6.6, 6.6 | $1.29 \cdot 10^5$ |
| 8 | inversion (posterior) | 6.9, 6.8, 6.8 | 0.6, 0.8, 0.6 | $1.30 \cdot 10^5$ | 12.4, 14.2, 14.0 | 2.5, 3.4, 3.0 | $1.29 \cdot 10^5$ |

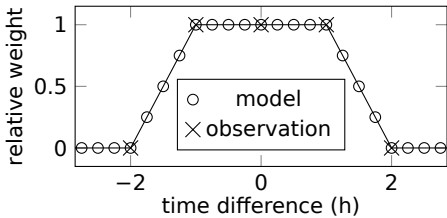

**Figure 3.** Weighting function for time interpolation of model and observations. For example, an interpolated model point at 16:30 UTC averages over all model output between 15:30 UTC and 17:30 UTC with full weight and another 1 h with linearly decreasing relative weight. The model yields instantaneous values every 15 min, whereas observations are provided as hourly averages, three of which contribute to the observational time average. Reference times are those times for which observations are available.

indeed increases significantly at low wind speeds. This increase is partially captured by an increase of the ensemble spread, supporting the idea of an uncertainty estimate depending on wind speed as proposed by Bergamaschi et al. (2022).

In the last filtering step – step 5 in Table 4 – we exclude data points with extreme mismatch between far-field corrected a priori and observations, where $|y - Hs - x^{\mathrm{ff}}| > 200\,\mathrm{ppb}$. Data points where $y - x^{\mathrm{ff}} < -20\,\mathrm{ppb}$ are also discarded. Since no strong sinks of $CH_4$ are expected, the contribution of $CH_4$ from the lateral boundaries should not exceed the observations.

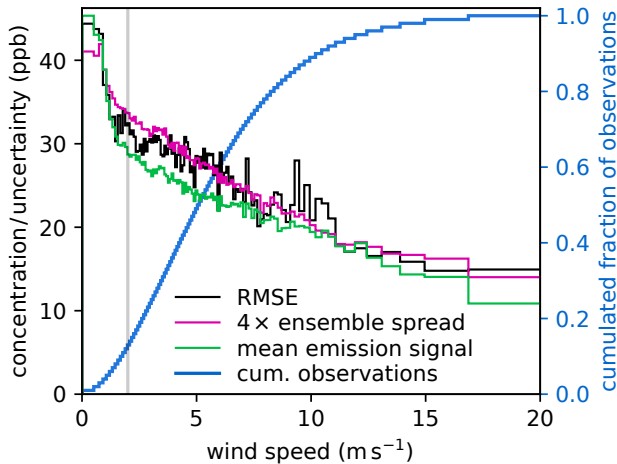

**Figure 4.** Dependency of RMSE and proxies for the model uncertainty on wind speed (left axis). All data points from step 3 in Table 4 were ordered by the model-predicted wind speed and split into 100 bins, each containing approximately 1500 data points. The blue line indicates the cumulative fraction of observations (right axis). The figure shows the RMSE difference of model and observation (black line), the mean ensemble spread multiplied by factor 4 (red line), and the mean a priori concentration due to categorized emissions (green line) for each of these bins. The ensemble spread is the standard deviation of the model prediction in the 12 ensemble members. It is a main contribution to our uncertainty estimate for the model–data mismatch in the prior $R$ and posterior $R$ inversion. The signal of categorized emissions is used to estimate the uncertainty for the diagonal $R$ matrix. Much of the larger RMSE at low wind speed is well captured by the ensemble spread inflated by factor 4 and by the mean a priori emission signal. In the inversion, we discard data points with wind speeds below $2\,\mathrm{m\,s^{-1}}$ (gray vertical line).

Thus, an observation below the model-predicted far field indicates an error in this far field. Steps 6–8 in Table 4 complete our processing chain by applying the far-field correction (Sect. 2.3), indicating the relevance of the model uncertainty (Sections 2.5 and 2.6), and finally using the inversion results.

### 3.4 Synthetic observation experiments

To test our setup and analyze biases, we use synthetic experiments in which observation data are replaced by model-generated pseudo-observations. These synthetic experiments use exactly the same setup and the same observation coordinates. Only the observation values are replaced by the simulation result of one of our 12 ensemble members. We thus obtain 12 separate datasets of pseudo-observations, in which a transport error is simulated by using the transport ensemble members. The true fluxes assumed for these synthetic experiments are identical to the prior fluxes. This allows us to estimate a bias and a random error in the posterior scaling factor. We will repeat this procedure with modified true fluxes in Sect. 4.3. An analysis of the sensitivity to random changes in the true fluxes is included in Part 2 (Bruch et al., 2025a).

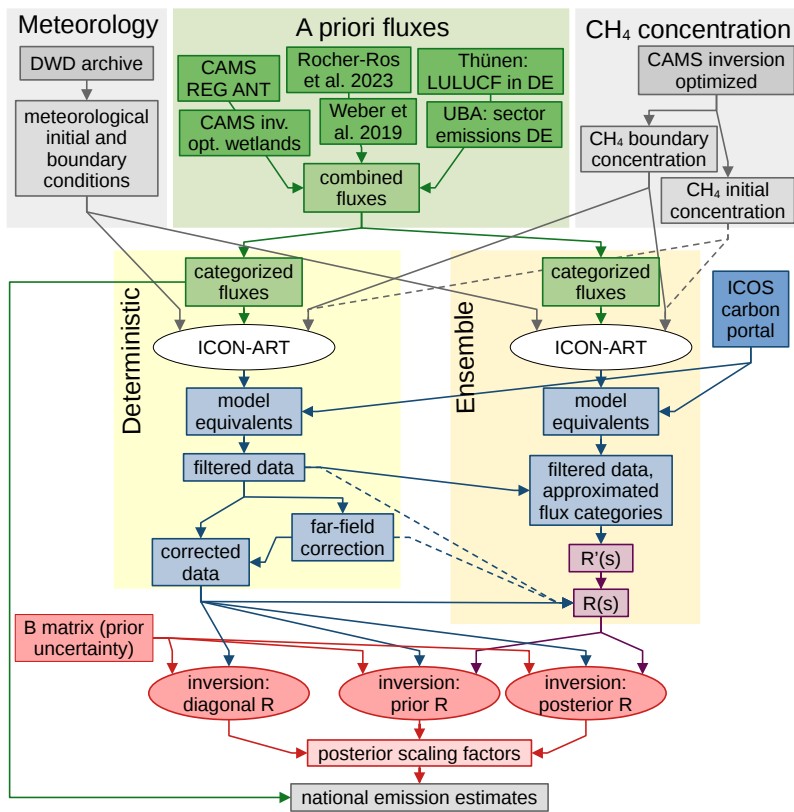

**Figure 5.** Overview of the inversion system including input data sources. Arrows indicate data streams. Dashed lines indicate data streams with small or negligible impact on the inversion results. Colored areas group the input data (top), the deterministic model run and data processing (left), and the ensemble model run including processing of the resulting data (right). Colored text boxes distinguish gridded fluxes (green), data in observation space (blue, matrices in purple), and data in the space of scaling factors (red). Observation data are included when working in observation space (not explicitly marked). At the end of the processing chain (bottom), the three methods for estimating $R$ lead to different scaling factors from which we can compute national emission estimates.

## 3.5 Summary and overview

We can now summarize the inversion method following the required data streams in Fig. 5. After collecting the input data for the transport simulation (Sections 3.1 and 3.2, top of Fig. 5), we prepare the inversion by categorizing the fluxes (Sect. 2.1.3). The transport is simulated separately for the deterministic and ensemble run (Sect. 2.1.1, white ellipses in Fig. 5). Using observation data from the ICOS carbon portal and the simulation output, we compute model equivalents and filter these to ensure a high quality of the model predictions (Sect. 3.3). The data from the deterministic run are used to construct a far-field correction to mitigate uncertainties in the boundary conditions (Sect. 2.3). The ensemble data are used to construct the uncertainty matrix $R(s)$ as required for the prior $R$ and posterior $R$ inversion (Sect. 2.5.2). The far-field corrected data and the $R$ matrix serve as

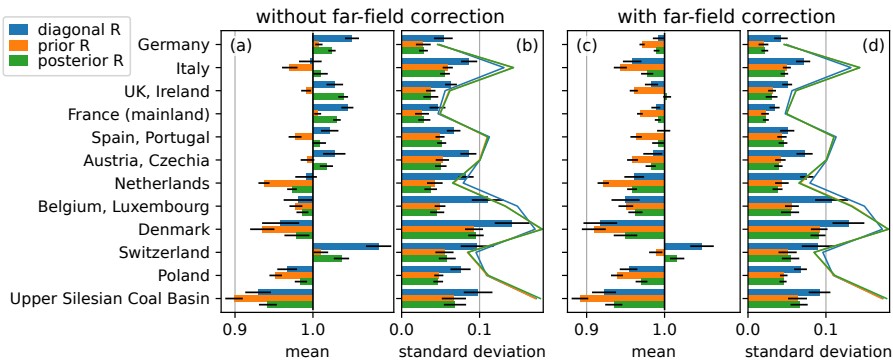

**Figure 6.** Mean (a, c) and standard deviation (b, d) of monthly flux estimates relative to the prior in synthetic experiments for diagonal $R$ (blue), prior $R$ (orange), and posterior $R$ inversion (green). Each bar represents the posterior fluxes for 144 inversions, obtained from 12 datasets of pseudo-observations, each covering 12 monthly time windows. Black horizontal lines indicate the $2\sigma$ statistical uncertainty estimate. Panels (a, c) show the bias as the relative deviation of the mean posterior from the prior, which is equal to the synthetic truth. The standard deviation (b, d) among the 144 emission estimates indicates the random error expected in each monthly inversion. Colored lines in (b, d) show the mean posterior $1\sigma$ uncertainty, which is similar for all three methods.

input for the Bayesian inversion (Sect. 2.4). By combining the resulting posterior scaling factors with the categorized fluxes, we obtain posterior flux estimates.

# 4  Results and discussion

In this section, we examine the presented inversion system using synthetic experiments and sensitivity tests. We start by considering synthetic observation experiments in which the synthetic truth is equal to the a priori fluxes. Figure 6 shows a statistical evaluation of inversion results for this case, which we analyze for multiple aspects.

## 4.1  Random error

In Fig. 6, we see the bias (panels a, c) and random error (b, d) of the inversion results for selected countries or emission sources relative to the a priori emissions, distinguishing the three methods for constructing $R$. The random error is estimated by the standard deviation obtained from 144 inversions and indicates the precision or reliability of these results for a single month. The comparison of the three methods shows that the prior $R$ and posterior $R$ method lead to a very similar random error, which is considerably lower than for the diagonal $R$ in all considered regions. This leads to the conclusion that using a transport ensemble to estimate uncertainties and their correlations can significantly reduce the random error in emission estimates, independent of the far-field correction.

Since the diagonal $R$ construction uses different tuning parameters than the prior $R$ and posterior $R$ inversion, we need to make sure that the chosen configurations are comparable. This is achieved by aiming for a similar posterior uncertainty in all methods for constructing $R$. Thin lines in Fig. 6 (b, d) show the posterior $1\sigma$ uncertainties to validate the similarity.

By comparing emission estimates without (panels a, b) and with the far-field correction (c, d), one can identify that the far-field correction changes the bias and slightly reduces the random error. Both effects are very similar for all three choices of $R$. Since the far-field correction pulls the simulated prior concentrations towards the observations, we can expect that it brings the emission estimates closer to the prior. But we can see in Fig. 6 (b, d) that the resulting reduction in random error is only weak.

## 4.2    Inversion bias

The bias shown in Fig. 6 (a, c) clearly depends on the far-field correction. The pseudo-observations without far-field correction have a bias of $+0.5\,\mathrm{ppb}$. The far-field correction reverts this to a negative bias of $-0.5\,\mathrm{ppb}$ due to a sampling bias as explained in Sect. 2.3. Ideally, we would therefore expect a small positive bias in Fig. 6 (a) and an equally strong negative bias in panel (c). But the bias differs depending on how $R$ is constructed.

For the diagonal $R$ inversion, we see overall a positive bias for most regions. This approximation for $R$ assumes a large uncertainty if the model predicts a strong signal from emissions. For an imperfect transport model, this implies that the model will tend to have a higher uncertainty when it overestimates the concentration and a lower uncertainty when it underestimates the real emission signal. As the model is more confident when observations are higher than the model prediction, it will tend to overestimate the emissions.

For the prior $R$ approximation, we find a negative bias in the emission estimates in many regions. This may be due do the plume bias problem introduced in Sect. 2.2. For the Upper Silesian Coal Basin as a very strong and localized source, all methods show the expected negative bias. Notably, a considerable negative bias is also found for the Netherlands as a small country with high emission rates.

    In the posterior $R$ approximation, the negative bias for plumes is reduced, but also all other emission estimates are higher
compared to the prior $R$ inversion. To understand this, we recall that a transport error in our model only leads to an error in the predicted $CH_4$ concentration if the concentration field contains spatial gradients. Such gradients are caused by emissions. Stronger emissions directly cause higher uncertainty estimates in the meteorological ensemble. In the posterior $R$ inversion, the inversion can adjust the emissions of the transport ensemble and thereby change the uncertainties. As we optimize the agreement of model and observations relative to the uncertainties, the system will prefer larger uncertainties. Thus, the inversion
will tend to overestimate emissions to reach higher uncertainties. This counteracts the negative plume bias, but it may also lead to a positive bias.

    By combining bias and random error, we obtain the RMSE. For Germany, the monthly results with far-field correction show an RMSE between $2.4\,\%$ (posterior $R$) and $4.3\,\%$ (diagonal $R$). For yearly totals, this reduces to $1.2\,\%$ for posterior $R$ and $1.8\,\%$ for diagonal $R$, while the prior $R$ inversion is dominated by the bias and has an RMSE of $2.9\,\%$. This indicates that
the simulated transport error in our synthetic experiments leads to an error of approximately $2\,\%$ on the German yearly total

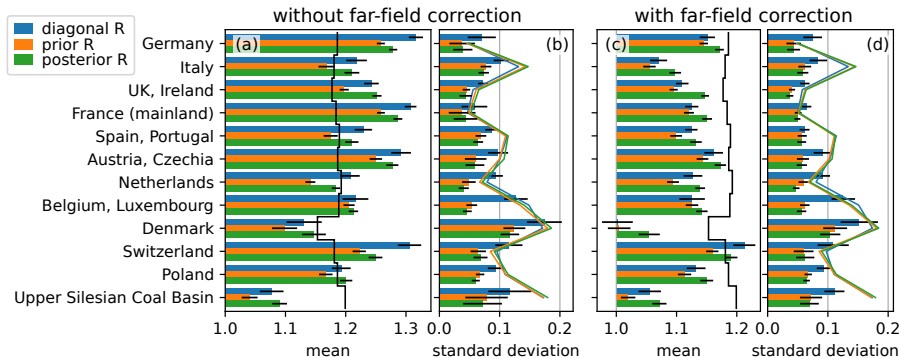

**Figure 7.** Mean (a, c) and standard deviation (b, d) of monthly flux estimates relative to the prior in synthetic experiments with $20\%$ increased anthropogenic emissions in the synthetic truth for diagonal $R$ (blue), prior $R$ (orange), and posterior $R$ inversion (green). In (a, c), the a priori has value $1.0$ and a black vertical line shows the synthetic truth. Bars connect the prior to the posterior. Like in Fig. 6, each bar represents the posterior fluxes for 144 inversions, combining 12 months with 12 datasets of pseudo-observations. Horizontal lines show $2\sigma$ statistical uncertainties and colored lines in (b, d) indicate the posterior $1\sigma$ uncertainty.

emission estimate. Overall, the posterior $R$ inversion shows the best performance as it has a lower random error and only a small bias.

## 4.3 Sensitivity to increased true emissions

To test the sensitivity of the inversion to true fluxes, we repeat the synthetic experiments with an identical setup but different
pseudo-observations. For these new pseudo-observations, we increase all anthropogenic emissions by $20\%$. The a priori emissions remain unchanged and are thus lower than the synthetic truth. The inversion results are summarized in Fig. 7, which is analogous to Fig. 6.

Figure 7 (a) and (c) show the mean posterior (bars) compared to the synthetic truth (black vertical line). Without the far-field correction, the inversion is too sensitive in many regions, as it increases the emissions beyond the synthetic truth. This leads to an overestimation, which is likely due to the artificial lifetime of the flux category tracers (see Sect. 2.1.4). With the far-field correction (panel c), the deviation of the posterior from the prior is damped and we obtain a low bias compared to the truth, as expected when the a priori emissions are underestimated. The random error (b, d) remains similar to the case with perfect prior emissions, albeit a small increase can be seen (compare Fig. 6). Like for the perfect prior emissions, the best performance with the lowest RMSE is found for the posterior $R$ inversion.

## 4.4 Sensitivity to bias and noise in observations

We now turn from the focus on the transport error to uncertainties in the observations. To this end, we consider different pseudo-observations without any transport error that follow scenarios defined in Fig. 8. To avoid the transport error, we generate these

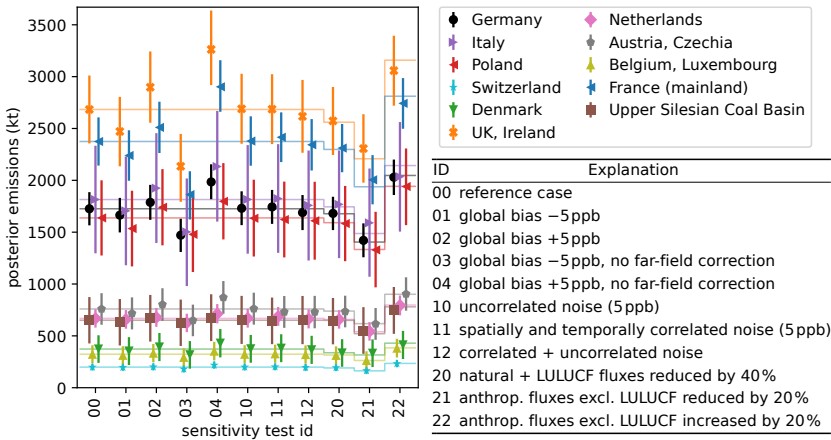

**Figure 8.** Total posterior emissions in 2021 of selected countries and German sectors for synthetic experiments with perfect transport. Markers show the average of the emission estimates obtained from the prior $R$ and posterior $R$ inversion. Thin horizontal lines indicate the synthetic truth. Vertical lines show uncertainties (95 % confidence intervals).

pseudo-observations based on the deterministic model run. For simplicity, we only consider the average of prior $R$ and posterior $R$ inversion.

In the first scenarios, we shift all pseudo-observations by $-5$ ppb (case 01 in Fig. 8) and $+5$ ppb (case 02). This bias is mostly compensated by the far-field correction with monthly averages of $\pm 2.75$ ppb to $\pm 3.8$ ppb, the sign depends on the scenario. Because of this correction, the effect on the estimated German total emissions remains well within the posterior uncertainty. This is in stark contrast to the same scenarios without the far-field correction (cases 03 and 04) and demonstrates the benefits of the far-field correction.

We furthermore test the effect of correlated and uncorrelated Gaussian noise added to the observations (cases 10–12), finding that the effect on the posterior emissions is small compared to the posterior uncertainties. The correlated Gaussian noise is a three-dimensional Gaussian random field in flat (longitude, latitude, time) coordinates with a lower cutoff for fluctuations on scales $\lesssim 2.5°$ (horizontal) and $\lesssim 12$ days (time) such that it acts as a slowly varying random bias. The RMS of the noise is normalized to $5$ ppb. For the last three test cases (20–22), we scale either the natural and LULUCF fluxes or all other emissions

in the synthetic truth while leaving the a priori emissions unchanged. Overall, the emission estimates follow the change in the synthetic truth well as already found in Sect. 4.3.

### 4.5   Sensitivity to inversion parameters

Our inversion method has various tuning parameters. Above we have described the inversion and synthetic experiments for one choice of these parameters. We analyze the sensitivity to these parameters by repeating the inversion 50 times with real

observations and modified parameters. Table E1 lists these test cases with their ID, parameters, and influence on the inversion results. An overview of the national emission estimates for each test case is provided in Fig. E1. Here, we summarize the main

results and refer to Table E1 for details. We use the average of the prior $R$ and posterior $R$ inversion results and focus on the influence of the parameters on the emission estimates, leaving the discussion of the inversion results for Part 2 (Bruch et al., 2025a).

### 4.5.1 Comparison to observations

Before comparing model and observations, we apply multiple filtering steps that influence the inversion results considerably. Most prominently, selecting nighttime observations for high mountain stations and afternoon hours for other stations strongly affects the inversion and improves the model representativeness (case 201 in Table E1). This is one of only five sensitivity tests with posterior fluxes deviating from the reference case by $\gtrsim 30\,\%$ of the posterior uncertainty, which we call a strong change in inversion results. Other filtering parameters such as the number of sampling heights used per station (case 202) and the minimal wind speed (cases 203–205) affect the inversion results noticeably, although changes are small compared to the uncertainties. Limiting the influence of outliers with model–observation mismatch $|\mu_i| > 3\sqrt{R'_{ii}}$ by increasing their uncertainty (see Sect. 2.6.2) has a considerable impact (cases 208, 209). Completely neglecting extreme outliers – defined by $|y - Hs - x^{\mathrm{ff}}| > 200\,\mathrm{ppb}$ or $y - x^{\mathrm{ff}} < -20\,\mathrm{ppb}$ – has only a small effect (cases 206, 207).

The choice of observation sites is analyzed in cases 601 and 602, which select subsets of stations with good observation coverage over the full year. When using only 27 stations (case 602), the results change strongly compared to the reference case with 50 stations, also because some regions are hardly observed in case 602. Varying the elevation of high mountain stations has only little impact on the inversion results (case 100). The effect of time-averaging over $3\,\mathrm{h}$ (as chosen in step 2 of Sect. 3.3) is noticeable in the results, but small compared to the uncertainties (case 101).

### 4.5.2 Uncertainty

The diagonal $R$ inversion deviates from the reference case by one third of the posterior uncertainty (case 311). Also the construction of the error covariance matrix $R$ following Sections 2.5 and 2.6 contains numerous tuning parameters. Key parameters are the overall uncertainty inflation factors $f_i$ (Sect. 2.6.3, cases 302 and 303 in Table E1) and the uncorrelated additive uncertainty $\sigma_{\mathrm{const}}$ (see Eq. (2)) of each data point (cases 309, 310). Variations of these parameters change the inversion results considerably. The tuning parameter $\sigma_{\mathrm{const}}$ illustrates the importance of hidden patterns in the considered data. Increasing to $\sigma_{\mathrm{const}} = 20\,\mathrm{ppb}$ effectively reduces the weight of observations with a small ensemble-estimated transport uncertainty. As observations with strong emission signals and high transport uncertainty become more relevant, the emission estimate for Germany is increased by $5\,\%$ (case 310 in Fig. E1).

Other important parameters are the correlation scales in the localization matrix $C$ for the ensemble-based uncertainty estimate (see Sect. 2.5.2). The overall effect of these scales on the posterior scaling factors is small (cases 304–308), but these parameters also influence the posterior uncertainties. The sensitivity tests indicate that 12 ensemble members are sufficient to estimate the uncertainties and correlations even without a strong localization. In general, we expect that a larger transport ensemble will yield better statistical estimates for uncertainties and their correlations. This reduces the need for a localization which suppresses spurious correlations. The considered additional plume localization uncertainty (see Sect. 2.6.1, cases

300 and 301) arising from the Upper Silesian Coal Basin seems negligible when considering the full domain. However, the additional plume localization uncertainty reduces the negative bias for the plume emissions that was discussed in Sect. 2.2.

### 4.5.3  Far-field correction

The synthetic experiments already showed that the far-field correction introduced in Sect. 2.3 influences the results considerably (see Figs. 6 and 7). When using real observations, removing the correction field leads to strong changes in the inversion results

(case 400), albeit the results remain within the posterior uncertainty bounds. Without the correction, the scaling factors for some natural fluxes in Scandinavia even become negative for some months – a clearly unrealistic result that underlines the importance of the far-field correction. However, changing various tuning parameters of the far-field correction within a reasonable range has much smaller effects. The selection of data points used for the far-field correction (cases 409, 410) and the overall correction strength (cases 401, 402) have modest influence, whereas correlation scales in the correction play a minor role (cases 403–408).

The additional uncertainty added to $R$ due to the far-field correction (see Sect. 2.6.4) has little influence on the inversion results (cases 412–414).

### 4.5.4  A priori error covariance matrix

Modifying the a priori uncertainty or correlations of the scaling factors ($B$ in Eq. (1)) changes the results quantitatively, but not qualitatively (cases 500–502). A coarser spatial resolution in Germany (case 504) and different choices of sectors (cases 503,

506) yield aggregated German sector emissions that agree well with the reference case.

### 4.5.5  Inversion time windows

In the reference case, we considered each month independently. Increasing the inversion time windows to three months has a considerable influence on the results (case 702). As the inversion time window increases, the overall weight of the observations in the inversion also increases. Thus, posterior uncertainties are reduced and the deviations between posterior and prior are

amplified.

## 5  Conclusions

This study introduced a new flux inversion system that explores the potential of a transport ensemble from NWP for observation-based regional estimation of methane emissions. In experiments with pseudo-observations and simulated transport error, we found that using a transport ensemble can substantially reduce the random error of the flux estimates compared to a simple

baseline scenario ("diagonal $R$"). This is in line with findings by Ghosh et al. (2021) and by Steiner et al. (2024a), who estimated $CH_4$ emissions in Europe using an ensemble Kalman smoother. But in contrast to Ghosh et al. (2021), who studied $CO_2$ at urban scale using an ensemble transform Kalman filter, we identified no significant improvement in the bias of the emission estimates. Instead, our results indicate systematic biases depending on the emissions characteristics. Most notably, localized sources causing strong plumes can be underestimation by $10\%$ by our synthesis inversion. To benefit from the transport en-

semble and to reduce such biases, we proposed to use the posterior concentrations in the ensemble when constructing $R$. This posterior $R$ inversion showed the best performance in the synthetic experiments. Overall, we expect an error of $2\%$ for the total German $CH_4$ emissions in 2021 in our inversion system due to random transport errors.

When applying our regional inversion system to real observations, we face the challenge of uncertain $CH_4$ concentrations at the lateral boundaries. Different approaches exist to correct biased boundary conditions. In some cases, selected measurements
can provide a baseline (Lauvaux et al., 2013). At national or continental scale, a coarse discretization of the boundaries allows optimization along with the emissions (Ganesan et al., 2015; Steiner et al., 2024b). Here, we followed a different path by adding a smooth correction field for the simulated concentrations. This allowed us to use different time scales for the inversion and the far-field correction. The far-field correction causes a small bias towards the prior fluxes, but without the correction we expect errors from wrongly projecting any boundary bias onto the fluxes. We demonstrated the potential of the far-field correction
using biased pseudo-observations and analyzed its importance in sensitivity tests, for which we repeated the inversion with different tuning parameters. These tests with real observations show that switch on the far-field correction changes the results considerably within the uncertainty ranges, but the specific choices made in constructing the correction field have only minor or moderate effects. Also other tested changes in tuning parameters only lead to variations of the full-year flux estimates well within the uncertainty ranges, indicating that we found robust settings for our application. This establishes a basis for applying
our system to validate the German emission inventory in Part 2 (Bruch et al., 2025a).

The presented novel inversion system leverages the potential of the ICON–ART model and the ensemble modeling capabilities from operational NWP for national scale estimation of $CH_4$ fluxes. It is tailored to the validation of national inventories by using high-resolution a priori emission estimates from national reporting and allowing for distinguishing emission sectors, as will be discussed in detail in Part 2. With synthetic experiments and sensitivity tests we demonstrated the suitability for
estimating national $CH_4$ emissions.

*Data availability.* A collection of model data, inversion results, and data for reproducing most figures in this work is available at https://doi.org/10.5281/zenodo.17414768 (Bruch et al., 2025b).

## Appendix A:  Formal definition of far-field correction

This appendix provides details for the far-field correction introduced in Sect. 2.3. We correct the computed far field by a
575 smooth field that is determined using all data points where the cumulated signal of all flux categories is at most $20\,\text{ppb}$, the total concentration due to all fluxes in the domain – including natural and uncategorized fluxes – is at most $50\,\text{ppb}$, and natural plus LULUCF fluxes contribute at most $20\,\text{ppb}$. These criteria aim to select only measurements of sufficiently clean air for the far-field correction.

The far-field correction is realized as a Kalman smoother on the selected data points. For simplicity, we only provide the
580 definition of the correction at the observation coordinates. Consider the vector of all model predictions $x$, which is aligned with

the observation vector $y$. By $P$ we denote the projector selecting those data points that shall be used to determine the far-field correction. We aim to find a correction vector $c$ aligned with $x$ and $y$ that minimizes

$$\underset{c}{\arg\min} \left\{ \tfrac{1}{2}(x+c-y)^\top P^\top \left(P\tilde{R}P^\top\right)^{-1} P(x+c-y) + \tfrac{1}{2} c^\top P^\top \left(P\tilde{C}P^\top\right)^{-1} Pc \right\}, \tag{A1}$$

where $\tilde{R} = 16I$ is a diagonal matrix and $\tilde{C}$ a Gaussian localization matrix with standard deviations $16\,\mathrm{h}$ (time), $319\,\mathrm{km}$ (horizontal) and $1\,\mathrm{km}$ (vertical), normalize to $\tilde{C}_{ii} = 1$ for all $i$. The matrix $\tilde{C}$ ensures that the correction field $c$ is smooth on these scales. For the under-determined Eq. (A1) we use the solution

$$c = \tilde{C}P^\top \left[ P(\tilde{C}+\tilde{R})P^\top \right]^{-1} P(y-x). \tag{A2}$$

This only defines $c$ at the observations, but we can generalize Eq. (A2) to arbitrary locations and times by including these coordinates in $\tilde{C}$. Formally, this then defines a smooth field.

To prove that Eq. (A2) is one possible – albeit not unique – solution of Eq. (A1), we use that Eq. (A1) is a quadratic form and compute its gradient with respect to $c$:

$$0 \stackrel{!}{=} P^\top \left(P\tilde{R}P^\top\right)^{-1} P(x+c-y) + P^\top \left(P\tilde{C}P^\top\right)^{-1} Pc. \tag{A3}$$

Since $PP^\top$ has full rank, this implies that

$$0 \stackrel{!}{=} \left[ \left(P\tilde{R}P^\top\right)^{-1} + \left(P\tilde{C}P^\top\right)^{-1} \right] Pc + \left(P\tilde{R}P^\top\right)^{-1} P(x-y) \tag{A4}$$

$$\implies Pc = \left[ 1 + P\tilde{R}P^\top \left(P\tilde{C}P^\top\right)^{-1} \right]^{-1} P(y-x) \tag{A5}$$

$$= P\tilde{C}P^\top \left[ P(\tilde{C}+\tilde{R})P^\top \right]^{-1} P(y-x). \tag{A6}$$

It follows that Eq. (A2) is a solution of Eq. (A1) that is independent of the non-selected data points. One can furthermore show that Eq. (A2) is optimal in the sense that it minimizes $c^\top \tilde{C}^{-1} c$ under constraint that $c$ is a solution of Eq. (A1). Thus, this solution is as close as possible to zero under the constraint of smoothness (quantified by $\tilde{C}$). By defining $\xi = \left[ P(\tilde{C}+\tilde{R})P^\top \right]^{-1} P(y-x)$ and introducing Lagrange multipliers $\lambda$, we obtain

$$f(c,\lambda) = c^\top \tilde{C}^{-1} c + \lambda^\top (Pc - P\tilde{C}P^\top \xi), \quad \frac{\partial f}{\partial c_i} = 0, \frac{\partial f}{\partial \lambda_j} = 0, \tag{A7}$$

$$c = -\tilde{C}P^\top \lambda \quad \text{from } \partial_{c_i} f(c,\lambda) = 0, \tag{A8}$$

$$Pc = P\tilde{C}P^\top \xi \quad \text{from } \partial_{\lambda_j} f(c,\lambda) = 0. \tag{A9}$$

Since $P\tilde{C}P^\top$ has full rank, combining Eqs. (A8) and (A9) implies that $\lambda = -\xi$ and thereby $c = \tilde{C}P^\top \xi$ is the unique solution of the optimization problem $\arg\min_c f(c,0)$ under the constraint that $Pc = P\tilde{C}P^\top \xi$.

## Appendix B: Posterior $R$ with reduced ensemble

When using a priori scaling factors to estimate the model uncertainty in $R$, we need only the total concentration $x_i^m(s^{\mathrm{prior}})$ for each ensemble member $m$ and each observation $i$, where $s^{\mathrm{prior}}$ is known. Thus, only a single tracer field is required in

the ensemble transport simulation. To compute $x_i^m(s)$ for arbitrary $s \in \mathbb{R}^{46}$, the flux categories need to be distinguished for each ensemble member, resulting in $> 40$ tracer fields in the ensemble simulation. To avoid wasting numerical resources, we chose to approximate $x_i^m(s)$ by only a few tracer fields, using additional information from the deterministic model run which distinguishes all tracer fields.

From the deterministic model run, we know the operator $H$ mapping scaling factors $s$ to a model prediction $Hs + x^{\text{ff}}$ for the concentrations. For ensemble member $m$, we would ideally know $H^m$ and $x^{\text{ff},m}$ to compute a model prediction $H^m s + x^{\text{ff},m}$. In lack of computational resources to compute $H^m$ for every ensemble member, we combine information from the deterministic run ($H$) and selected tracers for the ensemble run to approximate $H^m$. We group the flux categories into groups $\{g\}$ and denote by $P_g$ the projector of scaling vectors $s$ on the subspace spanned by the flux categories in group $g$. In the ensemble members, we compute the total concentration from group $g$, $x_i^{mg} = H^m P_g s^{\text{prior}}$. We distribute the 46 flux categories to only three groups and thereby reduce the computational effort considerably. To estimate the full dependence on the scaling factors in the ensemble, we approximate:

$$x_i^m(s) \approx \sum_g \frac{(HP_g s)_i}{(HP_g s^{\text{prior}})_i} x_i^{mg} + x_i^{\text{ff},m}. \tag{B1}$$

Thus, we compute the transport ensemble for a few tracer groups and estimate $x^m(s)$ for arbitrary $s$ by using the ratios of tracer fields within the tracer groups from the deterministic run. Using the approximation in Eq. (B1), we estimate the posterior model uncertainties with only five tracer fields in an ensemble of 12 transport simulations:

1. far field (initial and lateral boundary conditions)
2. total anthropogenic fluxes
3. total natural fluxes
4. total anthropogenic fluxes from Germany with lifetime five days
5. total anthropogenic fluxes from outside Germany with lifetime five days

**Appendix C: Observation sites**

**Table C1.** Observation stations from the European Obspack (ICOS RI et al., 2024). Column 6 ("mountain") characterizes the stations as high mountains, small mountains, and other stations. This serves as a reference for computing the station height in the model and for the daily time window. We indicate the sampling heights used in the inversion (column 7) and mark those sampling heights with an asterisk that have good observation coverage in each month (used in sensitivity test 602). Column 8 indicates times in which the station was excluded due to modeling problems. Column 9 ("inflation") defines the factor $f_i$ of the static uncertainty inflation (see Sect. 2.6.3).

| Code | Name | Country | ICOS class | Elevation (m) | Mountain | Sampling heights (m) | Limitations | Inflation |
|------|------|---------|-----------|---------------|----------|----------------------|-------------|-----------|
| BIK | Białystok | PL | – | 183 | no | 90, 180, 300 | | 2 |
| BIR | Birkenes | NO | 2 | 219 | no | 75 | excl. Apr–Aug | 3 |
| BIS | Biscarrosse | FR | – | 73 | small | 47$^*$ | | 2 |
| BRM | Beromunster | CH | – | 797 | no | 72, 132, 212 | | 2 |
| BSD | Bilsdale | UK | – | 382 | no | 108, 248 | | 2 |
| CBW | Cabauw | NL | 1 | 0 | no | 67, 127$^*$, 207$^*$ | | 2 |
| CMN | Monte Cimone | IT | 2 | 2165 | high | 8 | | 2 |
| CRA | Centre de Recherches Atmosphériques | FR | – | 600 | no | 60$^*$ | | 2 |
| CRP | Carnsore Point | IE | – | 9 | no | 14 | | 2 |
| ERS | Ersa | FR | – | 533 | small | 40 | | 3 |
| FKL | Finokalia | GR | – | 250 | small | – | excluded | – |
| GAT | Gartow | DE | 1 | 70 | no | 132$^*$, 216$^*$, 341$^*$ | | 2 |
| HEI | Heidelberg | DE | – | 113 | no | 30$^*$ | | 3 |
| HEL | Helgoland | DE | 2 | 43 | no | 110$^*$ | | 2 |
| HPB | Hohenpeissenberg | DE | 1 | 934 | small | 50, 93$^*$, 131$^*$ | | 2 |
| HTM | Hyltemossa | SE | 1 | 115 | no | 70, 150 | | 2 |
| HUN | Hegyhátsál | HU | 2 | 248 | no | 82, 115 | incl. Mar–Oct | 3 |
| IPR | Ispra | IT | 2 | 210 | no | – | excluded | – |
| JFJ | Jungfraujoch | CH | 1 | 3571.8 | high | 13.9 | | 2 |
| JUE | Jülich | DE | 2 | 98 | no | 120$^*$ | | 3 |
| KAS | Kasprowy Wierch | PL | – | 1987 | high | 7$^*$ | | 2 |
| KIT | Karlsruhe | DE | 1 | 110 | no | 60$^*$, 100$^*$, 200$^*$ | | 2 |
| KRE | Křešín u Pacova | CZ | 1 | 534 | no | 50, 125, 250 | | 2 |
| LHW | Laegern-Hochwacht | CH | – | 840 | small | 32 | | 3 |
| LIN | Lindenberg | DE | 1 | 73 | no | 98 | | 2 |
| LMP | Lampedusa | IT | 2 | 45 | no | – | excluded | – |

| Code | Name | Country | ICOS class | Elevation (m) | Mountain | Sampling heights (m) | Limitations | Inflation |
|------|------|---------|-----------|---------------|----------|----------------------|-------------|-----------|
| LMU | La Muela | ES | – | 571 | no | 79 | | 2 |
| LUT | Lutjewad | NL | 2 | 1 | no | 60 | excl. Nov–Dec | 2 |
| MHD | Mace Head | IE | – | 5 | no | 24$^*$ | | 2 |
| MLH | Malin Head | IE | – | 22 | no | 47 | | 2 |
| NOR | Norunda | SE | 1 | 46 | no | 58$^*$, 100$^*$ | | 2 |
| OHP | Observatoire de Haute Provence | FR | – | 650 | no | 50, 100 | | 2 |
| OPE | Observatoire pérenne de l'environnement | FR | 1 | 390 | no | 50$^*$, 120$^*$ | | 2 |
| OXK | Ochsenkopf | DE | 1 | 1022 | small | 90, 163 | | 2 |
| PAL | Pallas | FI | 1 | 565 | no | 12$^*$ | | 2 |
| PDM | Pic du Midi | FR | – | 2877 | high | 28 | | 2 |
| PRS | Plateau Rosa | IT | 2 | 3480 | high | 10 | | 2 |
| PUI | Puijo | FI | 2 | 232 | small | 84$^*$ | | 2 |
| PUY | Puy de Dôme | FR | 2 | 1465 | small | 10$^*$ | | 2 |
| RGL | Ridge Hill | UK | 2 | 207 | no | 90$^*$ | | 2 |
| ROC | Roc'h Trédudon | FR | – | 362 | no | 25, 80, 140 | | 2 |
| SAC | Saclay | FR | 1 | 160 | no | 60$^*$, 100$^*$ | | 2 |
| SMR | Hyytiälä | FI | 1 | 181 | no | 67.2$^*$, 125$^*$ | | 2 |
| SSL | Schauinsland | DE | 2 | 1205 | small | 12, 35 | | 2 |
| STE | Steinkimmen | DE | 1 | 29 | no | 127$^*$, 187$^*$, 252$^*$ | | 2 |
| SVB | Svartberget | SE | 1 | 269 | no | 85$^*$, 150$^*$ | | 2 |
| TAC | Tacolneston | UK | – | 64 | no | 54$^*$, 100$^*$, 185$^*$ | | 2 |
| TOH | Torfhaus | DE | 2 | 801 | small | 76$^*$, 110$^*$, 147$^*$ | | 2 |
| TRN | Trainou | FR | 2 | 131 | no | 50$^*$, 100$^*$, 180$^*$ | | 2 |
| UTO | Utö - Baltic sea | FI | 2 | 8 | no | 57$^*$ | | 2 |
| WAO | Weybourne | UK | 2 | 17 | no | 10$^*$ | | 2 |
| WES | Westerland | DE | 2 | 12 | no | 14 | | 2 |
| ZSF | Zugspitze | DE | 2 | 2666 | high | 3$^*$ | | 2 |

# Appendix D: $\chi^2$ analysis

In this appendix, we provide the mathematical details for the $\chi^2/N_\mathrm{dof}$ analysis (see, e.g., Greenwood and Nikulin, 1996) used in Sect. 2.6.5. The aim of this analysis is to quantify whether the data used in the inversion agree with the assumed uncertainties. We restrict this analysis to the prior $R$ and diagonal $R$ inversion, for which the matrix $R$ is constant. These inversions formally

rely on the assumption of Gaussian probability distributions of the a priori scaling factors (error covariance matrix $B$) and the model–observation mismatch ($R$).

We start from the probability density of observations $y$ under the assumption that $s$ describes the true emissions:

$$P(y|s) \propto \exp\left[-\tfrac{1}{2}(y - Hs - x^{\mathrm{ff}})^\top R^{-1}(y - Hs - x^{\mathrm{ff}})\right]. \tag{D1}$$

Like in the inversion, $R$ describes uncertainties in the transport, in the corrected far-field contribution $x^{\mathrm{ff}}$, and in the observations $y$. By a change of variables we obtain the probability for the a priori model–observation mismatch $\mu^{\mathrm{prior}} = y - Hs^{\mathrm{prior}} - x^{\mathrm{ff}}$:

$$P(\mu^{\mathrm{prior}}|s)d\mu = P(y|s)|_{y = Hs^{\mathrm{prior}} + x^{\mathrm{ff}} + \mu^{\mathrm{prior}}}dy.$$

To estimate whether a given $\mu^{\mathrm{prior}}$ is realistic, we need to integrate out the scaling factors $s$ to obtain $P(\mu^{\mathrm{prior}})$. We denote the integral over the vector space of scaling factors $s$ with probability measure $dP_s$ by $\int_s \bullet dP_s = \int_s P(s) \bullet d^n s$ for $s \in \mathbb{R}^n$. Using the above definitions in Eq. (D1), we obtain[3] (Berchet et al., 2015)

$$P(\mu^{\mathrm{prior}})$$

$$= \int_s P(\mu^{\mathrm{prior}}|s)\,dP_s \tag{D2}$$

$$\propto \int_s \exp\left[-\tfrac{1}{2}(y - Hs - x^{\mathrm{ff}})^\top R^{-1}(y - Hs - x^{\mathrm{ff}}) - \tfrac{1}{2}(s - s^{\mathrm{prior}})^\top B^{-1}(s - s^{\mathrm{prior}})\right]_{y = Hs^{\mathrm{prior}} + x^{\mathrm{ff}} + \mu^{\mathrm{prior}}}\,d^n s \tag{D3}$$

$$\overset{\tau = s - s^{\mathrm{prior}}}{=} \int_\tau \exp\left[-\tfrac{1}{2}(\mu^{\mathrm{prior}} - H\tau)^\top R^{-1}(\mu^{\mathrm{prior}} - H\tau) - \tfrac{1}{2}\tau^\top B^{-1}\tau\right]\,d^n\tau \tag{D4}$$

$$\propto \exp\left[-\tfrac{1}{2}\mu^{\mathrm{prior}\top}\left(R + HBH^\top\right)^{-1}\mu^{\mathrm{prior}}\right] \tag{D5}$$

$$=: \exp\left(-\tfrac{1}{2}\mu^{\mathrm{prior}\top}Q\mu^{\mathrm{prior}}\right). \tag{D6}$$

This result is a high-dimensional Gaussian probability distribution, $\mu^{\mathrm{prior}} \sim \mathcal{N}(0, Q^{-1})$. When drawing a random vector $\mu$ from
655 a probability distribution $P(\mu)$ as in Eq. (D6), it is very likely to find $\mu$ such that $\chi^2 \equiv \mu^\top Q\mu \approx N_{\mathrm{dof}}$ where $N_{\mathrm{dof}}$ denotes the number of degrees of freedom, which is the dimension of vector $\mu$. In our case, $N_{\mathrm{dof}} \sim 10^4$ is the number of observation data points used per one-month time window. In the limit of large $N_{\mathrm{dof}}$, one can approximate the probability distribution for $\chi^2$ by $\chi^2 \sim \mathcal{N}(N_{\mathrm{dof}}, 2N_{\mathrm{dof}})$ (Gaussian distribution with mean $N_{\mathrm{dof}}$ and variance $2N_{\mathrm{dof}}$) (Abramowitz and Stegun, 1964, Sect. 26.4). Thus, in an idealized setup we expect that $\chi^2/N_{\mathrm{dof}} = 1 \pm 0.03$ (95 % confidence interval). Values $\gtrsim 1.05$ hint at underestimated
uncertainties and $\chi^2/N_{\mathrm{dof}} \lesssim 0.95$ indicates that uncertainties were too high. However, in reality we may have biases and not fully described errors such that the assumption of a Gaussian uncertainty in the model–observation mismatch becomes invalid and $\chi^2/N_{\mathrm{dof}} < 1$ does not necessarily imply that uncertainties can simply be reduced.

---

[3]In Eq. (D5), we first solve the Gaussian integral to obtain $\exp\left\{-\tfrac{1}{2}\mu^{\mathrm{prior}\top}\left[R^{-1} - R^{-1}H(B^{-1} + H^\top R^{-1}H)^{-1}H^\top R^{-1}\right]\mu^{\mathrm{prior}}\right\}$ and then use that $(R + HBH^\top)\left[R^{-1} - R^{-1}H(B^{-1} + H^\top R^{-1}H)^{-1}H^\top R^{-1}\right] = I$.

## Appendix E: Sensitivity tests

Table E1 provides an overview of the sensitivity tests. For this table, we quantify the impact of a parameter variation on the inversion results by the following, heuristic metric: Consider a fixed region, sector and inversion time window with posterior fluxes $F$, defined as the average of the prior $R$ and posterior $R$ inversion result. The normalized deviation from the reference inversion is defined as $\Delta = \frac{2|F - F^{\text{ref.}}|}{F^{\text{ref. upper}} - F^{\text{ref. lower}}}$, where $F^{\text{ref. upper}}$ and $F^{\text{ref. lower}}$ denote the bounds of the posterior uncertainty range. The overall impact is computed as the arithmetic mean of $\Delta$ over the (usually monthly) time windows and a selection of regions and sectors. In the regions UK+Ireland, France, Italy, Poland, Austria+Czechia, Netherlands, Belgium+Luxembourg, Switzerland, and Denmark we consider only total fluxes without distinguishing sectors. In Germany we include $\Delta$ for the total fluxes in four different regions (north, east, south, west) and additionally for national total fluxes distinguishing the three sectors agriculture, natural plus LULUCF, and other sectors. Effectively, this counts all fluxes in Germany twice and gives them more weight in the impact metric for Table E1.

**Table E1.** Sensitivity tests for estimating the robustness of the inversion results with respect to tuning parameters. Modified numbers are marked in bold font. The impact column quantifies the deviation of the inversion results relative to the uncertainties and shall qualitatively indicate the relevance of the modified parameters (see explanation in the text). An impact of $100\%$ means that the average deviation from the reference case is as large as the posterior uncertainty. Overall, we see that most tests have an impact of $\lesssim 15\%$, implying that the effect on the inversion results is small compared to the uncertainty in the reference case. See also Fig. E1 for the posterior emissions in the sensitivity tests.

| ID | Test case | Explanation | Impact |
|---|---|---|---|
| 0 | reference | reference case as explained in Sections 2 and 3 and discussed in Part 2 (Bruch et al., 2025a), uses 129 117 observations in 2021 | |
| | **Model equivalent calculation** (see Sect. 3.3) | | |
| 100 | station elevation for mountain stations | treat all mountain stations like small mountains when computing model heights, as proposed by Brunner et al. (2012); Henne et al. (2016); Bergamaschi et al. (2022), uses 127 087 observations | 5.3 % |
| 101 | no additional time averaging | average over 1 h like in the observations, instead of averaging 3 h | 13 % |
| | **Filtering observations** (see Sect. 3.3) | | |
| 200 | fewer hours of day | use time window 12 h–16 h (0 h–4 h for high mountains), 85 674 observations (reference case uses 11 h–17 h / 23 h–5 h) | 11 % |
| 201 | all hours of day | no filtering by time of day, increase uncertainty inflation (factors $f_i$ in Sect. 2.6.3) by factor 1.5, uses 508 594 observations | 38 % |
| 202 | one sampling height per station | use only highest sampling height of each station instead of up to 3 highest levels, 80 132 observations | 16 % |
| 203 | no filtering based on wind | include data points with low wind speed, 147 019 observations | 12 % |
| 204 | low min. wind speed | minimum wind speed: $1.11\,\mathrm{m\,s^{-1}}$ (reference: $2\,\mathrm{m\,s^{-1}}$), 140 650 obs. | 9.4 % |
| 205 | high min. wind speed | minimum wind speed: $3.0\,\mathrm{m\,s^{-1}}$ (reference: $2\,\mathrm{m\,s^{-1}}$), 112 275 obs. | 11 % |
| 206 | low max. model-obs. mismatch | discard when $|y-Hs-x^{\mathrm{ff}}| > \mathbf{120}\,\mathrm{ppb}$ or $y-x^{\mathrm{ff}} < \mathbf{-12}\,\mathrm{ppb}$, 127 055 obs. (reference case: $200\,\mathrm{ppb}$ / $-20\,\mathrm{ppb}$) | 3.5 % |
| 207 | high max. model-obs. mismatch | discard when $|y-Hs-x^{\mathrm{ff}}| > \mathbf{300}\,\mathrm{ppb}$ or $y-x^{\mathrm{ff}} < \mathbf{-30}\,\mathrm{ppb}$, 129 706 obs. | 1.3 % |
| 208 | low max. data point influence | increase uncertainty if $|\mu_i| > \mathbf{2.5}\sqrt{R_{ii}^{\mathrm{step\,1}}}$ in Sect. 2.6.2 (reference value: 3) | 11 % |
| 209 | high max. data point influence | increase uncertainty if $|\mu_i| > \mathbf{4}\sqrt{R_{ii}^{\mathrm{step\,1}}}$ in Sect. 2.6.2 (reference value: 3) | 15 % |
| | **Uncertainty / error covariance matrix** $R$ (see Sections 2.5 and 2.6) | | |
| 300 | no plume uncertainty | no extra uncertainty due to localized emissions (Sect. 2.6.1) | 0.27 % |
| 301 | high plume uncertainty | extra uncertainty: $R_{ij}^{\mathrm{step\,1}} = R'_{ij} + \mathbf{0.5}\rho_i^2\delta_{ij}$ in Sect. 2.6.1 (reference: 0.25) | 0.56 % |
| 302 | low uncertainty inflation | uncertainty inflation by $f_i = 1.5$ or $2.25$ instead of 2 or 3 in Sect. 2.6.3 | 8.6 % |
| 303 | high uncertainty inflation | uncertainty inflation by $f_i = 3$ or $4.5$ instead of 2 or 3 in Sect. 2.6.3 | 13 % |
| 304 | small horizontal error correlation scale | scale $191\,\mathrm{km}$ instead of $319\,\mathrm{km}$ in localization matrix $C_{ij}$ (Sect. 2.5.2) | 6.0 % |

| ID | Test case | Explanation | Impact |
|----|-----------|-------------|--------|
| 305 | large horizontal error correlation scale | scale $510\,\mathrm{km}$ instead of $319\,\mathrm{km}$ in localization matrix $C_{ij}$ (Sect. 2.5.2) | 8.3 % |
| 306 | small vertical error correlation scale | scale $400\,\mathrm{m}$ instead of $1\,\mathrm{km}$ in localization matrix $C_{ij}$ (Sect. 2.5.2) | 2.3 % |
| 307 | short error correlation time scale | scale $4\,\mathrm{h}$ instead of $6\,\mathrm{h}$ in localization matrix $C_{ij}$ (Sect. 2.5.2) | 2.5 % |
| 308 | long error correlation time scale | scale $10\,\mathrm{h}$ instead of $6\,\mathrm{h}$ in localization matrix $C_{ij}$ (Sect. 2.5.2) | 2.8 % |
| 309 | low uncorrelated uncertainty | $\sigma_{\mathrm{const}} = 5\,\mathrm{ppb}$ instead of $10\,\mathrm{ppb}$ in Eq. (2) | 21 % |
| 310 | high uncorrelated uncertainty | $\sigma_{\mathrm{const}} = 20\,\mathrm{ppb}$ instead of $10\,\mathrm{ppb}$ in Eq. (2) | 22 % |
| 311 | diagonal $R$ without ensemble | see Sect. 2.5.1 | 33 % |

**Far-field correction** (see Sect. 2.3 and Appendix A)

| ID | Test case | Explanation | Impact |
|----|-----------|-------------|--------|
| 400 | no far-field correction | | 35 % |
| 401 | weak far-field correction | $\tilde{R} = 100I$ instead of $16I$ in Eq. (A1) | 16 % |
| 402 | strong far-field correction | $\tilde{R} = 2.78I$ instead of $16I$ in Eq. (A1) | 9.2 % |
| 403 | small horiz. far-field correction scale | scale $191\,\mathrm{km}$ instead of $319\,\mathrm{km}$ in localization matrix $\tilde{C}_{ij}$ in Appendix A | 6.8 % |
| 404 | large horiz. far-field correction scale | scale $510\,\mathrm{km}$ instead of $319\,\mathrm{km}$ in localization matrix $\tilde{C}_{ij}$ in Appendix A | 4.5 % |
| 405 | short far-field correction time scale | time scale $10\,\mathrm{h}$ instead of $16\,\mathrm{h}$ in localization matrix $\tilde{C}_{ij}$ in Appendix A | 3.7 % |
| 406 | long far-field correction time scale | time scale $28\,\mathrm{h}$ instead of $16\,\mathrm{h}$ in localization matrix $\tilde{C}_{ij}$ in Appendix A | 3.8 % |
| 407 | extra-long far-field correction time | time scale $48\,\mathrm{h}$ instead of $16\,\mathrm{h}$ in localization matrix $\tilde{C}_{ij}$ in Appendix A | 7.1 % |
| 408 | low vertical far-field correction scale | scale $400\,\mathrm{m}$ instead of $1\,\mathrm{km}$ in localization matrix $\tilde{C}_{ij}$ in Appendix A | 0.92 % |
| 409 | strict far-field observation selection | construct far-field correction based on observations with cumulated signal from categorized fluxes $\leq 10\,\mathrm{ppb}$ (reference: $20\,\mathrm{ppb}$) and from natural fluxes $\leq 10\,\mathrm{ppb}$ (reference: $20\,\mathrm{ppb}$) | 20 % |
| 410 | loose far-field observation selection | far-field correction uses observations with cumulated signal from categorized fluxes $\leq 30\,\mathrm{ppb}$ (ref.: $\leq 20\,\mathrm{ppb}$), from natural fluxes $\leq 30\,\mathrm{ppb}$ (ref.: $20\,\mathrm{ppb}$), and from all emissions within the domain $\leq 80\,\mathrm{ppb}$ (ref.: $50\,\mathrm{ppb}$) | 14 % |
| 411 | unrestricted iterative far-field correction | far-field correction uses all observations with cumulated signal from categorized fluxes $\leq 50\,\mathrm{ppb}$; $\tilde{C}_{ij}$ uses localization scales $10\,\mathrm{h}$, $191\,\mathrm{km}$; far-field correction and inversion are iterated 3 times, the correction always uses the posterior concentrations from the previous iteration. This aggressively suppresses large scale signals (biases) in the observations. | 30 % |
| 412 | low correction uncertainty | use $R_{ij}^{\mathrm{step}\,4} = R_{ij}^{\mathrm{step}\,3} + \mathbf{0.25}|c_i c_j|\tilde{C}_{ij}$ in Sect. 2.6.4 (reference value: 0.5) | 2.5 % |
| 413 | high correction uncertainty | use $R_{ij}^{\mathrm{step}\,4} = R_{ij}^{\mathrm{step}\,3} + \mathbf{1.0}|c_i c_j|\tilde{C}_{ij}$ in Sect. 2.6.4 (reference value: 0.5) | 4.2 % |
| 414 | uncorrelated correction uncertainty | use $R_{ij}^{\mathrm{step}\,4} = R_{ij}^{\mathrm{step}\,3} + 2c_i^2 \delta_{ij}$ in Sect. 2.6.4 | 3.6 % |

**A priori scaling factor error covariance matrix** $B$ (see Sect. 2.8)

| ID | Test case | Explanation | Impact |
|----|-----------|-------------|--------|
| 500 | low prior uncertainty | $1\sigma$ prior uncertainty set to 0.25 (ref.: 0.4) for well-observed areas, 0.2 (ref.: 0.25) for remote and plume categories, 0.33 (ref.: 0.5) for sector-resolving categories | 14 % |

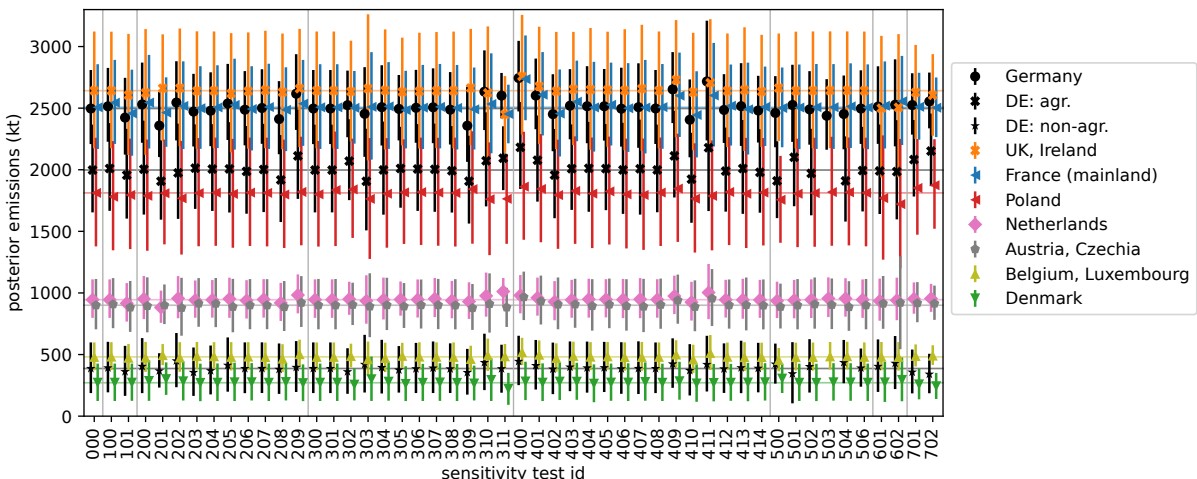

**Figure E1.** Posterior emissions and uncertainties of selected countries and German sectors for all sensitivity tests. Thin horizontal lines indicate the posterior of the reference case 0. Markers show the average of prior $R$ and posterior $R$ inversion. Vertical lines show uncertainties (95 % confidence intervals) and cover the uncertainty range of prior $R$ and posterior $R$ inversion. The individual tests are listed in Table E1. For all test cases, the emission estimates for the shown countries remain within the uncertainty range of the reference case.

| ID | Test case | Explanation | Impact |
|----|-----------|-------------|--------|
| 501 | high prior uncertainty in Germany | prior uncertainty such that national total sector emissions in Germany have $1\sigma$ uncertainty 60 % for each distinguished sector (reference: approx. 40%) | 8.6 % |
| 502 | uncorrelated prior, $B$ is diagonal | $1\sigma$ prior uncertainty in sector categories in Germany: 0.75; uncertainty on national total: 35 % for agriculture, 39 % for other anthropogenic) | 5.6 % |
| 503 | no sector distinction in prior | four regions in Germany with uncorrelated $1\sigma$ prior uncertainty of 0.4 | 7.7 % |
| 504 | low spatial resolution in Germany | two initially uncorrelated regions in Germany (south-west and north-east), each distinguishing sectors like in the reference case | 15 % |
| 506 | distinguish 5 sectors in Germany | split "non-agr." into sectors waste, public power, and other emissions | 2.1 % |
| | **Station selection** | | |
| 601 | require full-year coverage | require $\geq 10$ days coverage each month: 35 of 50 stations, 105 701 obs. | 13 % |
| 602 | require good full-year coverage | require $\geq 20$ days coverage each month: 27 of 50 stations, 82 912 observations (discussed in Fig. A2 of Part 2) | 33 % |
| | **Inversion time windows** (see Sect. 2.7) | | |
| 701 | 2 month inversion window | uncertainties are not adjusted to the longer window | 12 % |
| 702 | 3 month inversion window | uncertainties are not adjusted to the longer window | 18 % |

*Author contributions.* VB and TR conceptualized the inversion method. VB implemented the inversion method and wrote the original draft together with AKW. TR configured the transport model. TR and BE interpolated the a priori flux data which BE collected. DJCO organized data streams of $CH_4$ concentrations and observations. JF, BM, AMB, DJCO, TR and VB contributed to testing and tuning the transport model. NB contributed to the model–observation comparison. AKW supervised and coordinated the project. All authors reviewed and edited the manuscript.

*Competing interests.* The authors declare that they have no conflict of interest.

*Acknowledgements.* In our simulations we use modified Copernicus Atmosphere Monitoring Service information and ECCAD products for initial and lateral boundary conditions, and for a priori fluxes. We thank Stefan Feigenspan, Christian Mielke, Theo Wernicke, John Akubia and Roland Fuß for helpful discussions and providing a priori emission fields. We thank Roland Potthast, Frank-Thomas Koch, Christoph Gerbig, Dominik Brunner, Michael Steiner, David Ho, Thomas Kaminski, Hannes Imhof and our partners in the ITMS project for very helpful and inspiring discussions. We also wish to thank Peter Bergamaschi, Aurélie Colomb, Martine De Mazière, Lukas Emmenegger, Dagmar Kubistin, Irene Lehner, Kari Lehtinen, Markus Leuenberger, Cathrine Lund Myhre, Michal V. Marek, Simon O'Doherty, Stephen M. Platt, Christian Plaß-Dülmer, Francesco Apadula, Sabrina Arnold, Pierre-Eric Blanc, Dominik Brunner, Huilin Chen, Lukasz Chmura, Łukasz Chmura, Sébastien Conil, Cédric Couret, Paolo Cristofanelli, Grant Forster, Arnoud Frumau, Christoph Gerbig, François Gheusi, Samuel Hammer, Laszlo Haszpra, Juha Hatakka, Michal Heliasz, Stephan Henne, Arjan Hensen, Antje Hoheisel, Tobias Kneuer, Eric Larmanou, Tuomas Laurila, Ari Leskinen, Ingeborg Levin, Matthias Lindauer, Morgan Lopez, Ivan Mammarella, Giovanni Manca, Andrew Manning, Damien Martin, Frank Meinhardt, Meelis Mölder, Jennifer Müller-Williams, Steffen Manfred Noe, Jarosław Nęcki, Mikaell Ottosson-Löfvenius, Carole Philippon, Joseph Pitt, Michel Ramonet, Pedro Rivas-Soriano, Bert Scheeren, Marcus Schumacher, Mahesh Kumar Sha, Gerard Spain, Martin Steinbacher, Lise Lotte Sørensen, Alex Vermeulen, Gabriela Vítková, Irène Xueref-Remy, Alcide di Sarra, Franz Conen, Victor Kazan, Yves-Alain Roulet, Tobias Biermann, Marc Delmotte, Daniela Heltai, Ove Hermansen, Kateřina Komínková, Olivier Laurent, Janne Levula, Chris Lunder, Per Marklund, Josep-Anton Morguí, Jean-Marc Pichon, Martina Schmidt, Damiano Sferlazzo, Paul Smith, Kieran Stanley, Pamela Trisolino and Giulia Zazzeri for providing the atmospheric observations for the stations listed in Table C1. VB, DJCO, NB and AMB acknowledge funding by the German Federal Ministry for Education and Research (BMBF) in the ITMS project (grant 01LK2102B) as well as BE (grant 01LK2104A). Map plots were made with Natural Earth.

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

# German methane fluxes estimated top-down using ICON–ART – Part 2: Inversion results for 2021

Valentin Bruch[1], Thomas Rösch[1], Diego Jiménez de la Cuesta Otero[1], Beatrice Ellerhoff[1], Buhalqem Mamtimin[1], Niklas Becker[1], Anne-Marlene Blechschmidt[1], Jochen Förstner[1], and Andrea K. Kaiser-Weiss[1]

[1]Deutscher Wetterdienst, Frankfurter Str. 135, 63067 Offenbach

**Correspondence:** Valentin Bruch (valentin.bruch@dwd.de) and Andrea K. Kaiser-Weiss (andrea.kaiser-weiss@dwd.de)

**Abstract.** A reliable quantification of greenhouse gas emissions is important for climate change mitigation strategies. Inverse methods based on observations and atmospheric transport simulations can support emission quantification at the national scale, yet, they are often limited by the observing systems, transport model uncertainties, and inversion methodologies. This two-part study introduces a system for observation-based, regional methane flux estimation. In the present Part 2, we apply this system to estimate German methane emissions in 2021. The numerical weather prediction model ICON with its ART module for trace gases is used to simulate the atmospheric transport while estimating uncertainties using a transport ensemble. We use a priori fluxes from national reporting to facilitate the validation of reported fluxes. Posterior fluxes are estimated with a modified synthesis inversion method introduced in Part 1, relying on in-situ observations. Compared to the a priori, we find a significant increase in methane emissions in Germany and in the Benelux. We estimate German methane emissions $(32 \pm 19)\,\%$ higher than the anthropogenic emissions in the national inventory, and our inversion method attributes this difference mainly to the agricultural sector, although separation from Land Use, Land Use Change and Forestry (LULUCF) as well as natural fluxes requires further research. The combination of an ensemble-enhanced numerical weather prediction model for atmospheric transport and good observation coverage paves the way to sector-specific, observation-based national emission estimates.

## 1 Introduction

Reducing greenhouse gas (GHG) emissions is crucial for mitigating current anthropogenic global warming. UNFCCC (United Nations Framework Convention on Climate Change) compliant national inventories and/or process models quantify anthropogenic GHG emissions for the purpose of monitoring the effectiveness of mitigation as planned, e.g., in the Paris Agreement. In addition to so-called "bottom-up" methods, atmospheric GHG concentration observations are used in "top-down" flux estimations. The latter are complementary, as they are sensitive to the total fluxes (i.e., anthropogenic and natural) and provide options for independent validation of a priori fluxes provided by inventories (IPCC et al., 2019). The usefulness of top-down estimates has been demonstrated, e.g., for the United Kingdom (Manning et al., 2011), Switzerland (Henne et al., 2016), Europe (Petrescu et al., 2023) and globally (Deng et al., 2022; Petrescu et al., 2024).

Although research foundations for top-down methods have been developed in recent decades (see Janssens-Maenhout et al. (2020) and references therein), applications remain limited due to sparse observation coverage and representativeness, and most critically, due to transport model uncertainties (Engelen et al., 2002; Gerbig et al., 2008). The latter is a well-known issue not solved yet (Munassar et al., 2023). Inversions using satellite observations (e.g. Estrada et al., 2024) benefit from larger spatial observation coverage, but the uncertainties of the observations are larger compared to in situ data and the influence on the inversion results was found smaller where in situ coverage is good (Thompson et al., 2025). The benefits of increased model resolution (Agustí-Panareda et al., 2019; Bergamaschi et al., 2022) can be reaped with regional high resolution modeling and ensembles can cover parts of the meteorological uncertainty (Steiner et al., 2024a). At short time scales, the regional model uncertainties will constitute the main uncertainty, while at longer time scales, the boundary conditions become critical for tracer transport (Chen et al., 2019).

In this work, we present first results of a modular system for regional top-down estimates of $CH_4$ fluxes designed to validate national inventories, including the discrimination of economic sectors such as agriculture and industry. We apply this method focusing on German inventories (provided by Umweltbundesamt and Thünen Institute) for the year 2021 using in situ observations collected by ICOS (ICOS RI, 2024). Atmospheric transport is simulated using the numerical weather prediction model ICON (Zängl et al., 2015) extended with the module for Aerosol and Reactive Trace gases (ART) (Rieger et al., 2015; Schröter et al., 2018) with a spatial resolution of $6.5\,\mathrm{km}$. The model is combined with a synthesis inversion approach (Kaminski et al., 2001) which is developed further to make use of the ensemble-estimated transport uncertainty. For minimizing transport errors, we rely on the operational numerical weather prediction at Germany's Meteorological Service (DWD) for meteorological initial conditions, lateral boundaries and transport ensemble calculations. Further, we use the Copernicus Atmosphere Monitoring Service (CAMS) for boundary conditions of methane, and compensate possible biases on the boundaries by deriving a correction field. Benefiting from the numerical weather prediction model and spatially highly resolved a priori fluxes from the inventory agencies, we explore the basis for future operational top-down validation of national emission reporting, with special emphasis on further use in Germany.

In Sect. 2, we summarize the methodology which is introduced in detail in Part 1 of this work (Bruch et al., 2025a). Section 3 contains the results for 2021, together with validation tests and an analysis of the ability to distinguish emission sectors. In Sect. 4 we discuss limitations and capabilities of the method and compare to other studies, followed by a conclusion in Sect. 5.

## 2 Method

This section is a non-technical summary of the detailed method description in Part 1 (Bruch et al., 2025a).

### 2.1 Parametrization of fluxes

We aim to validate the national reporting of German $CH_4$ emissions to the UNFCCC. A simple way to address this validation problem is the following question: By which single number should we multiply all reported German $CH_4$ emissions based on the information from observed $CH_4$ concentrations? We can extend this question and estimate different scaling factors for

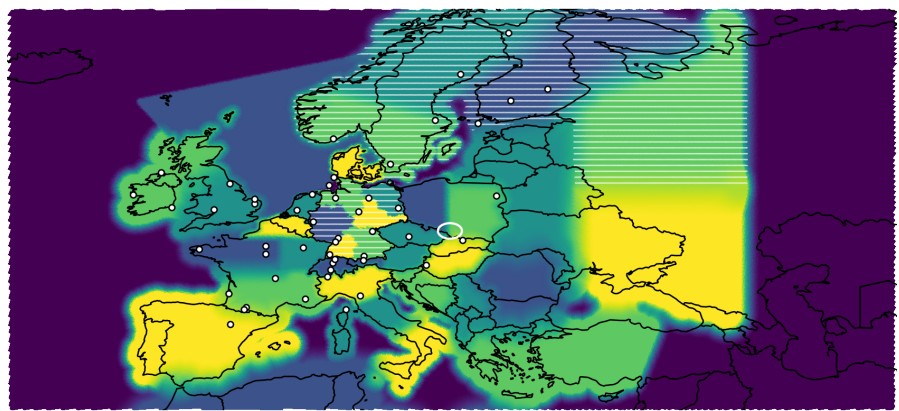

**Figure 1.** Overview of the model domain indicating flux categories (colored areas) and observation sites (white dots), modified from Part 1 (Bruch et al., 2025a). Each connected area of equal color defines one flux category for anthropogenic emissions, except in Germany and the Netherlands, where the categories are split up further to distinguish agriculture emissions from other sectors. In white hatched regions, natural fluxes form additional flux categories because large natural fluxes are expected. Close to the eastern and western domain boundary (dark blue), emissions are not adjusted by the inversion. Fugitive emissions from the Upper Silesian Coal Basin (white ellipse) define their own flux category.

different regions and different emission sectors. In this work, we estimate scaling factors for 46 categories of $CH_4$ fluxes for each month in 2021. The spatial definition of these flux categories is shown in Fig. 1. In Germany, we distinguish 11 flux categories, consisting of six regions for the agriculture sector, one flux category for land use, land use change and forestry (LULUCF) plus natural fluxes, and four regions for the sum of all remaining emissions. In summary, the state space of our inversion is defined by the flux categories and consists of only 46 numbers.

## 2.2 A priori fluxes

For the a priori fluxes outside Germany, we combine CAMS-REG (Kuenen et al., 2021, 2022) for anthropogenic emissions with wetland emissions from the CAMS global inversion-optimized dataset (Segers and Houweling, 2020), version v22r2. For Germany, we use emissions obtained from the inventory agencies, that is, the Umweltbundesamt (German Environmental Agency, Feigenspan et al., 2024) and the Thünen Institute (Fuß and Akubia, 2024). Moreover, we consider emissions from rivers and streams (Rocher-Ros et al., 2023), as well as oceans (Weber et al., 2019).

## 2.3 Transport simulation

To connect surface fluxes and observations, we need to simulate atmospheric transport. This simulation is done using the numerical weather prediction model ICON (Zängl et al., 2015) with the module for Aerosol and Reactive Trace gases (ART) (Rieger et al., 2015; Schröter et al., 2018) at a horizontal resolution of $6.5\,km$. Initial and lateral boundary conditions for the $CH_4$ concentrations are taken from the CAMS global inversion-optimized dataset (Segers and Houweling, 2020), version

v22r2. To mitigate a possible bias in the lateral boundary conditions, we construct a smooth correction field that is added to all model predictions of the boundary contributions. This far-field correction is constructed based on observations for which the model predicts clean air with small influence of emissions from within our domain. We estimate transport uncertainties and their correlations using an ensemble of 12 members with slightly different meteorology, derived from the operational numerical weather prediction at DWD (Schraff et al., 2016).

## 2.4 Observations

We use $CH_4$ concentration observations from the European Obspack (ICOS RI et al., 2024) as provided on the Integrated Carbon Observation System (ICOS) carbon portal. The hourly observations are filtered by time of day and wind speed to use only observations that can be predicted well by the transport model. We use night time observations (23 h to 5 h local mean time) for high mountain stations and afternoon hours (11 h to 17 h local mean time) for all other sites, discarding observations at wind speeds below $2\,\mathrm{m\,s^{-1}}$.

## 2.5 Bayesian Inversion

To estimate the scaling factors of the flux categories, we use a Bayesian inversion. Denoting the scaling factors as a vector $s \in \mathbb{R}^{46}$, the inversion is formulated as the optimization problem

$$s^{\mathrm{post}} = \arg\min_s \left\{ \tfrac{1}{2}[y - H'(s)]^\top R^{-1}[y - H'(s)] + \tfrac{1}{2}(s - s^{\mathrm{prior}})^\top B^{-1}(s - s^{\mathrm{prior}}) \right\}. \tag{1}$$

Here, $y$ denotes a vector of all observations and $H'(s)$ is the model prediction for these observations, which includes the previously mentioned far-field correction. $R$ is the error covariance matrix of the model–observation mismatch and $B$ is the error covariance matrix of the a priori scaling factors $s^{\mathrm{prior}}$. Since $s$ describes prefactors to the a priori emissions, we initially set $s_k^{\mathrm{prior}} = 1$ for all $k$. In $B$ we assume an a priori uncertainty of $2\sigma = 0.8$ (two standard deviations) for the scaling factors of most regions. This gives the inversion enough freedom to adjust the scaling factors. In large distance from Germany, the a priori uncertainty is reduced to $2\sigma = 0.5$ (see Fig. 2 b), and for emission sectors in Germany and the Netherlands we use $2\sigma = 1.0$.

The construction of $R$ based on the transport ensemble is discussed in detail in Part 1 (Bruch et al., 2025a). In Eq. (1), $R$ can be estimated using either a priori or a posteriori fluxes. This defines two slightly different methods that are introduced in Sect. 2.5 of Part 1 as "prior $R$" and "posterior $R$" inversion. Here, we only consider the average of the two results and the union of the two posterior uncertainty ranges.

## 2.6 Posterior uncertainties

To estimate the uncertainties of posterior fluxes conservatively, we repeat the inversion $50 \times 2$ times with each of the 50 observation sites excluded once for each of the two approximations for $R$. The lower and upper bounds of the resulting hundred $2\sigma$ uncertainty ranges form our posterior $95\%$ confidence interval. This ensures that a result that is only based on a single observation site will not be considered significant.

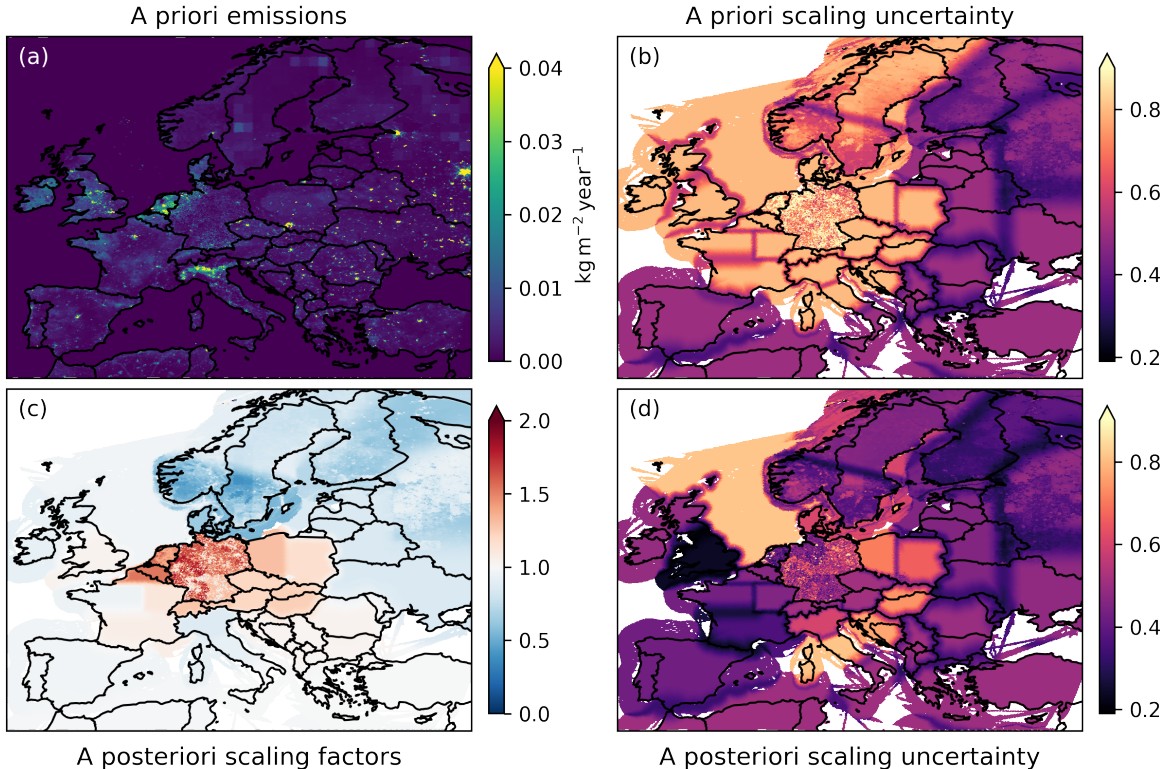

**A priori emissions**

**A priori scaling uncertainty**

**A posteriori scaling factors**

**A posteriori scaling uncertainty**

**Figure 2.** Full-year averages of (a) a priori fluxes, (b) a priori uncertainty on scaling factors, (c) a posteriori scaling factors, and (d) a posteriori uncertainty on scaling factors. Multiplying the a priori emissions (a) with the scaling factors (c) yields the a posteriori emissions. (b) and (d) show half of the $95\%$ confidence interval of the fluxes relative to the a priori fluxes, i.e., a $2\sigma$ uncertainty of 0.5 on the a priori appears as 0.5 on the color scale. The direct comparison indicates the uncertainty reduction. The smooth boundaries between two regions with separate scaling factors appear as darker lines because these scaling factors are assumed to be initially uncorrelated.

## 2.7 Inversion time window

The scaling factors are estimated separately for each month in 2021 by using only observations from the selected month. The results for different months are thus independent. But since the posterior uncertainty estimates include systematic uncertainties, we assume that uncertainties from different months are correlated.

## 3 Results

### 3.1 Resulting scaling factors

Figure 2 presents an overview of (a) the a priori $CH_4$ fluxes accumulated over the year 2021, (c) the resulting scaling factors averaged over 2021, and the respective uncertainties (b, d). The a posteriori scaling factors (Fig. 2 c) show the correction to

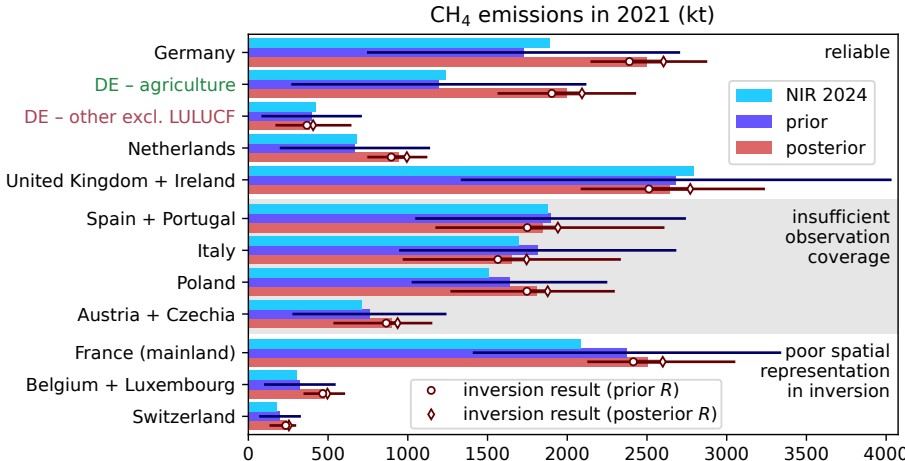

**Figure 3.** National CH$_4$ emission estimates comparing reported (NIR), prior, and posterior fluxes for 2021 with horizontal lines indicating 95 % confidence intervals. Countries are grouped by the expected robustness of their inversion results. Some neighboring countries are combined to obtain more accurate results. For Germany, the inversions can resolve the agricultural sector, though the separation against natural and LULUCF fluxes is difficult. All other anthropogenic sectors are combined in the category "other excl. LULUCF". The inclusion of two inversion methods ("prior $R$" and "posterior $R$", markers) provides an estimate of the methodological uncertainty. Accumulated emissions from national inventory reports (NIR) to the UNFCCC submitted 2024 (including LULUCF emissions) are shown for reference (light blue bars, UNFCCC, 2024). For France (Citepa, 2024) and the United Kingdom (Department for Energy Security and Net Zero, 2024), the light blue bars show emission data from the respective inventory agencies excluding overseas territories and crown dependencies. Posterior uncertainties that are asymmetric with respect to flux estimates such as in Switzerland indicate the strong influence of a single observation site.

the a priori emissions obtained in the inversion. A considerable increase in emissions is found for Germany and the Benelux. Lower emissions compared to the a priori are predicted for Scandinavia (see discussion in Sect. 4.3). The scaling factors should be considered jointly with their uncertainties. The comparison of Fig. 2 (b) and (d) shows a substantial uncertainty reduction for Germany and most of the surrounding countries, for which we chose a high a priori uncertainty.

For a more detailed comparison of a priori and a posteriori emissions and uncertainties, we consider selected national emis-
115 sion estimates in Fig. 3. Reliable inversion results are expected for countries or regions with sufficient observation coverage, strong emission signals, representation in the respective flux categories, and only moderate issues due to complex topography. These criteria are met for Germany, the Netherlands and the United Kingdom plus Ireland as grouped in Fig. 3. For Germany (first entry in Fig. 3), the total posterior CH$_4$ emissions (red bar) are $(32 \pm 19)\%$ higher than the anthropogenic emissions including LULUCF reported to the UNFCCC in 2024 (light blue bar). The direct comparison to the reporting neglects the
120 unreported natural fluxes, but for Germany these are expected to be small because all relevant soil emissions are included in the LULUCF sector. The inversion significantly increases emission estimates from the agriculture sector while the combined

other sectors remain nearly unchanged. Note, however, that the uncertainty in the sector attribution is large (horizontal lines, see further discussion in Sections 3.4.2 and 4.3).

For the Netherlands, we also find significantly higher emissions than in the inventory. Compared to Germany, the attribution to sectors has an even larger uncertainty, associated with fewer observations that could distinguish the sectors. Nevertheless, the total emissions from the Netherlands are comparably well constrained by the observations. For the United Kingdom and Ireland – which we combine to obtain more accurate results – the inversion yields a strong uncertainty reduction while hardly changing the total emissions, indicating a good agreement of observations and national inventory.

In most countries, the observations do not cover the whole country, or the inversion results rely on few observations. In Fig. 3 (gray-shaded part) we provide emission estimates also for countries or regions affected by this issue, though these have a large posterior uncertainty. Another issue arises from the definition of the flux categories, which do not necessarily follow country borders (see Fig. 1). In France, Belgium, and Switzerland, the inversion scales flux categories that overlap multiple countries[1]. This implies that national emission estimates derived for these countries have an additional uncertainty and artificial correlations with neighboring countries. However, this is of no concern for our application for Germany. The national emission estimates are computed from the gridded posterior fluxes and precisely follow the country borders as shown in Fig. 2. The scaling factors and uncertainties of all flux categories are listed in Fig. A1 for completeness.

## 3.2 Seasonal cycle

Although the national emission estimates are given for the full year, a closer examination of the seasonal cycle provides additional insights. Figure 4 shows the monthly emission rates for the countries considered in Fig. 3. While the seasonal cycle is strikingly different depending on the region, we find some recurrent features. For Germany, Poland, the Netherlands, and Austria plus Czechia (panel (a) in Fig. 4), the posterior emission rates have their minimum in May. A local minimum between April and June is also found for northern France and Belgium plus Luxembourg, see panel (b). In most countries, this minimum is followed by a local maximum in July or August, which is most prominent in the Netherlands and Austria plus Czechia (panel (a)).

The differences between the regions become larger in autumn and winter. In September, posterior emission rates reach their maximum in Germany and Italy, and their minimum in (northern) France. France and Belgium plus Luxembourg have their highest emission rates in winter, when Switzerland and Spain plus Portugal have their minimum. For some regions – most notably Italy and the United Kingdom plus Ireland – no clear pattern is found in the seasonal cycle for 2021 (panel (c) in Fig. 4).

The seasonal cycle in the inversion results may be partially influenced by the observation coverage because many stations lack data covering the whole year. To avoid this effect, we repeated the inversion using only stations which provide data for at least 20 days of each month. The seasonal cycle in these results does not change significantly, see supplementary Fig. A2. We further note that there is a seasonal cycle in the observations (East et al., 2024), which is captured well by the far field in

---

[1]Technically, the issue also affects Italy because Corsica is combined with parts of Italy in one flux category. But the a priori emissions from Corsica are so low that the effect on the national emission estimate is negligible.

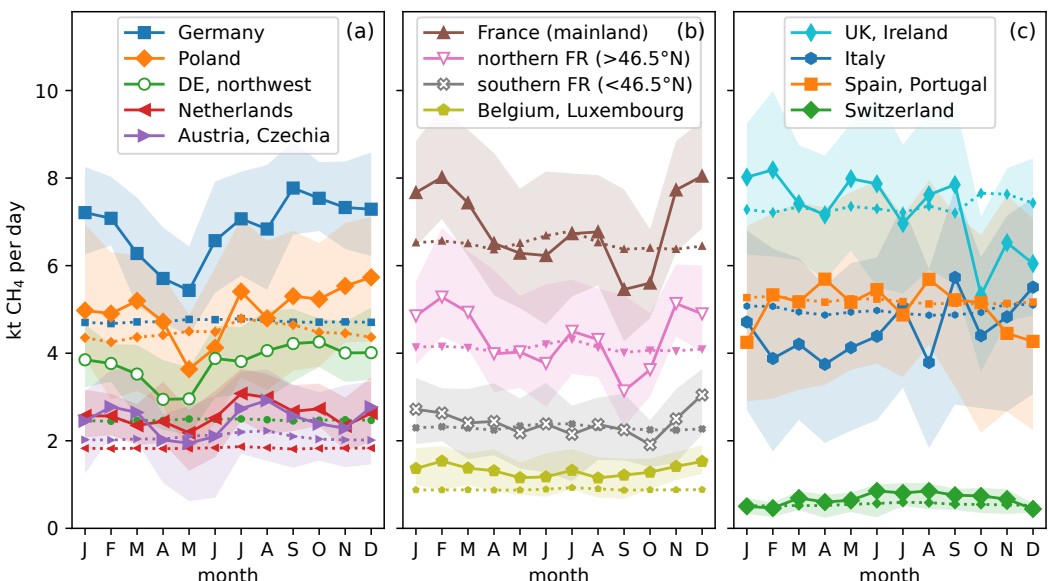

**Figure 4.** Monthly posterior emission rates for selected countries or regions. Colored areas show the posterior uncertainties, and dotted lines with small markers indicate prior emission rates. In the prior, only the natural and LULUCF fluxes are time-dependent. The panels show (a) countries with minimum in May, (b) countries with a maximum in winter, and (c) other countries and regions. For France and Germany, selected regions are shown additionally (white markers). "DE, northwest" includes Rhineland-Palatinate, Saarland, Hesse, North Rhine-Westphalia, Lower Saxony, Schleswig-Holstein, Bremen and Hamburg.

the model though (see Fig. A3). This "far field" is defined as $CH_4$ transported into our domain from the lateral boundaries.

A possible bias in the lateral boundary conditions could influence the seasonal cycle in the estimated fluxes. Moreover, the different meteorology in summer and winter – especially influencing the planetary boundary layer and vertical mixing (Seidel et al., 2012) – can lead to a seasonal bias in our transport model (Bessagnet et al., 2016; Canepa and Builtjes, 2017). This highlights the need for careful interpretation of the seasonal cycle, as meteorological differences could introduce biases that mask true emission patterns. Another potential contribution to the seasonal cycle could arise from neglecting the OH sink of

$CH_4$ in our limited domain (Logan et al., 1981).

### 3.3 Validation

A straightforward validation of the inversion results is possible using independent validation stations. Having excluded each station once in separate inversion runs, we can use every station as an independent validation site in the respective inversion run. Figure 5 shows histograms of the root mean square error (RMSE) statistics obtained from the model–data mismatch before

165 and after the inversion. The validation stations agree on average significantly better with observations when using a posteriori emissions compared to the a priori. A comparison of the same histograms for the different methods of estimating uncertainties introduced in Part 1 (Bruch et al., 2025a) shows no significant differences (see supplementary Fig. A4).

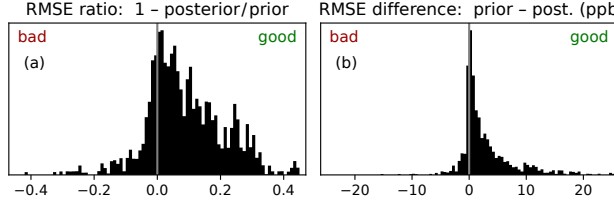

**Figure 5.** Statistics of the relative (a) and absolute (b) improvement of the model–observation mismatch by the inversion at independent validation stations. Each station and month is considered separately in its own inversion, with the validation station excluded from the inversion to remain independent. The histograms show (a) $1 - r^{\mathrm{post}}/r^{\mathrm{prior}}$ and (b) $r^{\mathrm{prior}} - r^{\mathrm{post}}$ where $r^{\mathrm{post}}$ and $r^{\mathrm{prior}}$ refer to the RMSE of the model–observation comparison in the case of posterior scaling and prior scaling, respectively. Each time series contributing to the histogram is weighted by the number of its data points. We consider all data points within the daily time window without filtering for wind speed or model–observation mismatch and without the far-field correction introduced in Part 1 (Bruch et al., 2025a) to keep the comparison as close as possible to the original data. Positive values indicate an improvement in the model prediction due to the inversion.

## 3.4 Potential for detecting emissions

In this section, we complement the uncertainty estimates of our inversion results by separate measures for the sensitivity of the posterior to true emissions. The potential for detecting emissions from different sources can be identified using the posterior error covariance matrix $B_{\mathrm{post}}$. However, the real error reduction is also influenced by the far-field correction and the filtering of observations as detailed in Part 1 (Bruch et al., 2025a). These aspects are not fully captured in $B_{\mathrm{post}}$. We therefore use experiments with a "synthetic", i.e., defined truth and pseudo-observations to test the full inversion system.

### 3.4.1 National emission estimates

We first aim to verify that the inversion yields meaningful posterior emission estimates and uncertainties given a perfect transport model. To this end, we generate 100 random vectors of scaling factors following the probability distribution assumed in the a priori uncertainty. Each vector of scaling factors defines a synthetic truth, and the model prediction for the observations obtained using these scaling factors defines our pseudo-observations. We further add uncorrelated Gaussian noise of standard deviation $2\,\mathrm{ppb}$ to these pseudo-observations. Since the pseudo-observations are inferred from the model data, there is no transport error in these synthetic experiments. This construction of pseudo-observations clearly underestimates the true error in the model–observation comparison, but it allows us to test the interplay of far-field correction and inversion in a controlled setup. Synthetic experiments with a simulated transport uncertainty are discussed in Part 1 (Bruch et al., 2025a).

The quality of the model prediction is shown in Fig. 6 for selected countries and German sectors. By comparing to the synthetic truth, we find the prior and posterior error. Their ratio (vertical axis in Fig. 6) shows a significant improvement by the inversion for all considered regions and German sectors, with the exception of German natural and LULUCF fluxes. The uncertainty reduction of the inversion (horizontal axis) provides a realistic estimate of the real error reduction (vertical axis) for the case of high quality observations, ideal transport modeling, and perfect lateral boundary conditions. In some cases

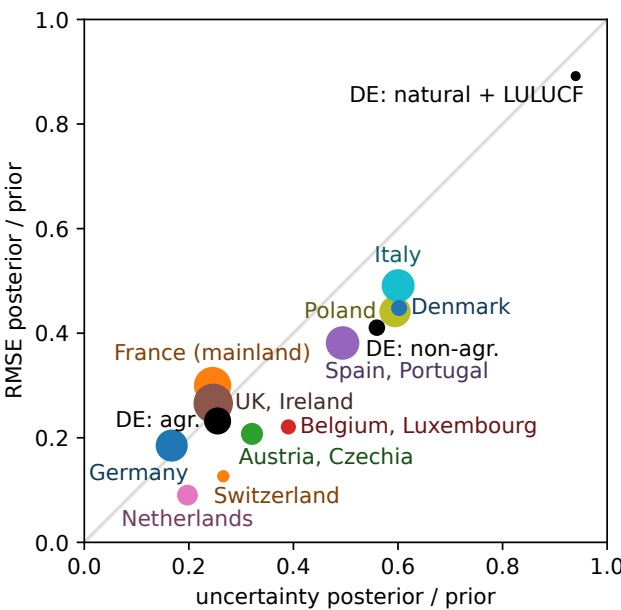

**Figure 6.** RMSE and mean uncertainty of CH$_4$ emission estimates in synthetic experiments for selected countries, regions, and German emission sectors. Each of the 100 synthetic experiments uses random true emissions. The vertical axis shows the root mean square (RMS) deviation of the posterior from these true emissions, relative to the RMS deviation of the prior from the truth. Lower values indicate that the inversion improves the emission estimate. The horizontal axis shows the posterior uncertainty relative to the prior uncertainty. Therefore, the bottom left indicates best performance. The disk size indicates the magnitude of the prior emissions.

(Netherlands, Switzerland, Belgium, and Luxembourg), the real error reduction is significantly better than the uncertainty reduction suggests. This is no surprise because in this synthetic setup the transport error as the main source of uncertainty is switched off. Overall, the synthetic experiments confirm the potential for a strong uncertainty reduction in Central Europe.

### 3.4.2 Distinguishing sectors in Germany

Within Germany, we distinguish agriculture from other emissions. The discrimination of emission sectors works in the same way as we distinguish emissions from different areas. Each sector has a specific spatial distribution of emissions, which we assume to be correct in the a priori. The predicted CH$_4$ concentration at the observation sites will therefore depend on how the individual sectors are scaled. In the inversion, the sector emissions are scaled to find optimal agreement of model prediction and observations.

The ability to distinguish sectors can be described by averaging kernel matrices which estimate the dependence of the posterior on the true emissions, $A_{ij}^{\text{emis}} = \partial e_i^{\text{post}} / \partial e_j^{\text{truth}}$ where $e_i$ denotes emissions from sector $i$. Since the true emissions $e^{\text{truth}}$ are generally unknown, the averaging kernels $A^{\text{emis}}$ can only be estimated. Figure 7 shows such estimates for $A^{\text{emis}}$ (panels a, c) and the averaging kernel for scaling factors, $A_{ij}^{\text{scaling factors}} = \partial s_i^{\text{post}} / \partial s_j^{\text{truth}}$ (panels b, d). Assuming a perfect transport model

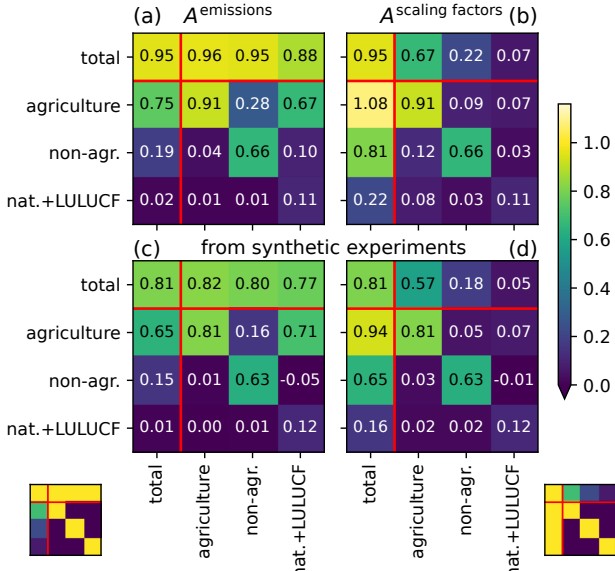

**Figure 7.** Averaging kernel matrices of German sector emissions (a, c) and scaling factors (b, d). The kernel is estimated using either the posterior covariance matrix (a, b) or 100 synthetic experiments with random truth (c, d). The small matrices on the bottom indicate what we aim for (posterior equals truth). The value 0.96 in the first row ("total"), second column ("agriculture") of panel (a) means that if in reality all German agriculture emissions were $1\,\mathrm{kt}$ higher than in our prior, then we would expect an increase in the posterior total German emissions by $0.96\,\mathrm{kt}$. Similarly, the value 0.67 in the same cell of panel (b) means that increasing real agriculture emissions by $10\,\%$ should increase our posterior total emissions by $6.7\,\%$. All matrices are averaged over the whole year. Red lines separate the individual sectors from their sum ("total"). By "non-agr." we denote anthropogenic emissions excluding agriculture and LULUCF.

and perfect far field, the averaging kernel matrix can be estimated by $A^{\mathrm{emis}} \approx I - B_{\mathrm{post.\ emis}} B_{\mathrm{prior\ emis}}^{-1}$ (Rodgers, 2000) using the prior and posterior covariance matrices of the emissions from the "prior $R$" inversion (see Appendix B1). $I$ denotes the identity matrix. Figure 7 (a) shows this averaging kernel estimate for German sector emissions, extended by a row and column for the total German emissions.

The first row of Fig. 7 (a) indicates that the total German posterior emissions follow changes in every sector with high accuracy ($88\,\%$ to $96\,\%$). The diagonal of Fig. 7 (a) signifies that changes in the agriculture will be detected very well and also the attribution to the sum of all other anthropogenic sectors excluding LULUCF ("non-agr.") will be mostly correct. However, LULUCF plus natural fluxes will in large parts be falsely attributed to the agriculture (second row, last column). Note that ideally, the first row and the diagonal elements would be close to $100\,\%$ (color-coded in the small matrix bottom left). The

averaging kernel $A^{\mathrm{scaling\ factors}}$ in Fig. 7(b) shows that the influence of LULUCF and natural emissions on the posterior scaling factor for agriculture emissions remains low (second row, last column). But if all emissions are scaled by the same factor (first column), the changes will be mostly attributed to the agriculture sector. This effect is expected because the agriculture sector

has the highest absolute a priori uncertainty, which makes changes in agriculture more likely than changes in any other sector. A formal derivation of this argument is presented in Appendix C.

The averaging kernel matrices in Fig. 7 (a) and (b) are estimated based on the "prior $R$" inversion while neglecting the far-field correction. We complement these by a statistical estimate of the averaging kernels using 100 synthetic experiments with random truth (see Appendix B2), shown in Fig. 7 (c) and (d). Here, the far-field correction is applied as implemented in our processing chain. While these statistical estimates reproduce all qualitative features in the averaging kernels, the matrix entries estimated using synthetic experiments are generally lower. This is likely due to the far-field correction and indicates

that deviations from the prior emissions may be underestimated by our inversion. Importantly, both presented strategies for estimating the averaging kernels assume a perfect transport model. The real sensitivity of the posterior to the true emissions is therefore expected to be lower.

## 4   Discussion

Our inversion system combines precise in situ observations, accurate a priori fluxes from national reporting, the ICON–ART

transport model at $6.5\,\mathrm{km}$ resolution, and an ensemble-estimated transport uncertainty. We further rely on CAMS boundary conditions and high-resolution meteorological fields from operational numerical weather prediction. This yields in general a good agreement between the model prediction and filtered observations, allowing us robust emission estimates for well-observed countries, such as Germany. We compare top-down $CH_4$ emission estimates to the reported German inventory and its agriculture sector with enough accuracy to lay the technical foundations for a future long-term observation-based national

inventory verification. This section discusses our main results (Sect. 4.1), including a comparison with other studies (Sect. 4.2). We elaborate the limitations of our approach (Sect. 4.3) and its potential for the development of observation-based national inventory verification to inform climate policy (Sect. 4.4).

### 4.1   Key findings

Firstly, we find that our top-down $CH_4$ emission estimates are significantly higher than reported for Germany. Secondly, we

identify the agriculture sector and possibly LULUCF and natural fluxes as the likely main source of this discrepancy. Thirdly, we recall from Part 1 (Bruch et al., 2025a) that the transport error simulated in the meteorological ensemble leads to an uncertainty of $2\,\%$ on the total German $CH_4$ emissions.

### 4.2   Comparison to other methods

Our Eulerian approach with sectoral segregation differs from other studies on $CH_4$ inversions for single countries, e.g., Henne

et al. (2016) for Switzerland and Ganesan et al. (2015) for the United Kingdom that use Lagrangian transport models. The latter both qualitatively attribute deviations from the inventory reporting to the agriculture sector by comparing the spatial and/or temporal patterns in the posterior fluxes to sectoral a priori fluxes. A similar strategy for sectoral segregation based on a known spatial distribution of fluxes is followed by Varon et al. (2022) and analyzed by Cusworth et al. (2021). For deriving

sector estimates, some inversions assume a spatial correlation of gridded emissions within each sector (Rödenbeck et al., 2003; Meirink et al., 2008b; Bergamaschi et al., 2010). Based on the same assumption, Steiner et al. (2024b) and Tenkanen et al. (2025) construct ensembles of perturbed a priori fluxes to distinguish natural and anthropogenic fluxes utilizing the CarbonTracker Data Assimilation Shell (van der Laan-Luijkx et al., 2017). Notably, Tenkanen et al. (2025) avoid the lateral boundary problem by simulating transport globally with nested zoom in Europe to estimate Finnish $CH_4$ emissions on a coarse resolution of $1° \times 1°$. In the present work, we take the next step by validating sectoral emissions reported to UNFCCC and analyzing possible false attributions, making use of a significantly higher model resolution.

Our results are qualitatively in line with the discrepancy of top-down estimates and UNFCCC reporting for Germany and the Benelux found in different regional inversions for the years 2018 and earlier (Petrescu et al., 2023; Bergamaschi et al., 2022, 2018; Steiner et al., 2024b). Furthermore, it appears as a robust feature in our results that emissions from the UK plus Ireland agree well with reported emissions, in line with Bergamaschi et al. (2022) for the year 2018. For the French emissions, our inversion shows a tendency towards slightly higher emissions similar to Steiner et al. (2024b), whereas other inversions suggest significantly higher emissions (Petrescu et al., 2023; Bergamaschi et al., 2022).

## 4.3 Limitations

Although we simulate emissions and transport in a large domain, we can only provide reliable emission estimates for selected countries (compare Fig. 3). Regions without notable uncertainty reduction and regions with known modeling difficulties do not benefit from our model setup. In Scandinavia, we find strong wetland emissions with insufficiently modeled fine-scale spatial and temporal variability. Combined with only small signals from non-LULUCF anthropogenic emissions, this leads to a low signal-to-noise ratio, which prevents conclusive results for Scandinavia. Furthermore, the synthesis inversion may be prone to underestimating large localized sources due to transport errors – an issue we address in Part 1 (Bruch et al., 2025a).

Another limitation comes from the challenges for the regional flux inversion caused by biases in the lateral boundary conditions. The uncertainty in lateral boundary concentrations motivates the far-field correction that is discussed in Part 1 (Bruch et al., 2025a). We expect that the far-field correction leads to more robust estimates for well-observed emissions, but it may also cause a bias towards the prior and towards lower emission estimates.

In our highly resolved transport simulation, every flux category is numerically expensive. Aiming to validate reported German emissions, we could reduce the state space of the inversion to only 46 scaling factors with monthly time resolution. This substantially limits the spatial and temporal variations that can be represented in the inversion. This approach is justified if the a priori fluxes already provide a realistic spatial distribution of all major $CH_4$ sources within each flux category. While this may be the case in Germany and neighboring countries, the constant scaling factors for large flux categories in more distant regions may be oversimplified and could lead to less accurate results in these regions. Moreover, adjusting only a few degrees of freedom may not be sufficient to obtain realistic flux estimates in regions with limited or highly uncertain information on a priori fluxes, such as Scandinavia.

When constructing the state space, we unevenly distributed the 46 degrees of freedom on our model domain – using 11 degrees of freedom for Germany and only four for mainland France plus Belgium and Luxembourg. But the choice of flux

categories affects the results and can lead to biases depending on the location of the observations (Kaminski et al., 2001). In our application, this effect is small because of the good observation coverage in Germany. This is checked in Part 1 (Bruch et al., 2025a) using sensitivity tests.

We exploit the sectoral discrimination of emission in a well-observed region as a key feature of our inversion method. This relies heavily on an accurate spatial distribution and completeness of the a priori fluxes, which appears to be sufficient for the major emitting sectors in Germany. Furthermore, the sector discrimination relies on resolving comparably small spatial scales, which poses a challenge to the transport modeling. A general problem in sector attribution is that sectors with large absolute uncertainty – such as agriculture – may be falsely blamed for any change in total emissions when the observations do not clearly distinguish the sectors (see Appendix C). By quantifying this effect in the averaging kernels (see Fig. 7), we confirmed that in Germany agriculture can be distinguished from other anthropogenic emissions excluding LULUCF. Small sectors like natural plus LULUCF fluxes could not be reliably distinguished from large sectors such as agriculture, and we therefore combined smaller sectors like waste and public power into the larger category "non-agr.".

## 4.4 Implications for future research

We chose the synthesis inversion for the first application of our modular inversion system, but designed this framework to be expandable to other inversion methods. For instance, most of the steps in the inversion can be applied with only minor adjustments when replacing the flux categories by an ensemble of randomly perturbed surface fluxes, similar to Steiner et al. (2024b), or by grid cell clusters as used by Estrada et al. (2024). Such applications with a larger state space are limited by the computational effort of the transport simulation, which is much higher than the computational effort of the inversion itself. Similar to the inversion method, the far-field correction can be replaced by a different strategy for mitigating a boundary bias. For example, one could construct the far field based on an ensemble of boundary concentrations.

Further possibilities of extension involve other observation types, including satellite data. Our Eulerian system allows in principle the handling of large observation datasets without prohibitive computational effort, albeit changes in the construction and handling of $R$ may be required when reaching $\gtrsim 10^5$ observations per time window. This potential is leveraged by many inversion systems that use Eulerian transport simulations (e.g., Varon et al., 2022; Meirink et al., 2008a; Bergamaschi et al., 2013). The increasing availability of satellite data is especially interesting for constraining concentrations and emissions in regions with few or no ground-based observations, such as near the boundaries of our domain, which is an aspect to be addressed in future studies.

We identified potentials and risks in separating sectors based on the spatially highly resolved distribution of fluxes. Extending this by temporal profiles for a priori fluxes offers a yet untapped potential for future improvement of our system. Moreover, our inversion could benefit from an a priori emission ensemble reflecting the uncertainty in the spatial and temporal distribution of the fluxes. It remains to be explored whether improvements in distinguishing sectors can be achieved in our system using co-tracers such as ethane for fossil $CH_4$ emissions (Ramsden et al., 2022; Mead et al., 2024) or by distinguishing carbon isotopes (Basu et al., 2022; Thanwerdas et al., 2024; Chandra et al., 2024).

# 5    Conclusions

We presented first results from a novel system for regional flux inversion designed to validate national $CH_4$ emission reporting. Applying this method to Central Europe in 2021 with a focus on Germany, we found significantly higher emissions from Germany and the Benelux compared to the reporting. Careful estimation of posterior uncertainties revealed for the investigated year that the total German posterior emissions are $(32 \pm 19)\,\%$ higher than the respective anthropogenic emissions reported to the UNFCCC (submission 2024). With our inversion method the difference is attributed to emissions from the agriculture sector, possibly with contributions from the LULUCF sector and natural sources. Our results were confirmed by validation with independent observation sites and by an exhaustive range of sensitivity tests presented in Part 1 (Bruch et al., 2025a). Synthetic experiments with known truth revealed the method's ability to distinguish the agricultural from the non-agricultural sectors in Germany, whereas disentangling possible influences from natural and LULUCF sources requires further work and possibly more observations.

A methodological comparison to other regional inversion systems highlights the advantages of our method for the purpose of distinguishing emission sectors and its suitability for validating national emission estimates. The qualitative gap between UNFCCC reporting and our estimates for Germany and the Benelux is consistent with earlier works (Petrescu et al., 2023; Bergamaschi et al., 2022, 2018; Steiner et al., 2024b). We complement these studies by providing an emission estimate for the German agriculture sector that can be directly compared to the national reporting, revealing a significant mismatch.

In this study we presented the first application of an extensible, novel inversion system. Future developments may include the integration of satellite data, the incorporation of temporal profiles, a more comprehensive treatment of boundary conditions and flux uncertainties using ensemble methods, and an extension of the state space. The close connections to operational numerical weather prediction – especially in the underlying transport simulation – and the modular design establish the potential for long-term operational support of national emissions reporting.

*Data availability.*  A collection of model data, inversion results, and data for reproducing most figures in this work is available at https://doi.org/10.5281/zenodo.17414768 (Bruch et al., 2025b).

## Appendix A:  Supplementary figures

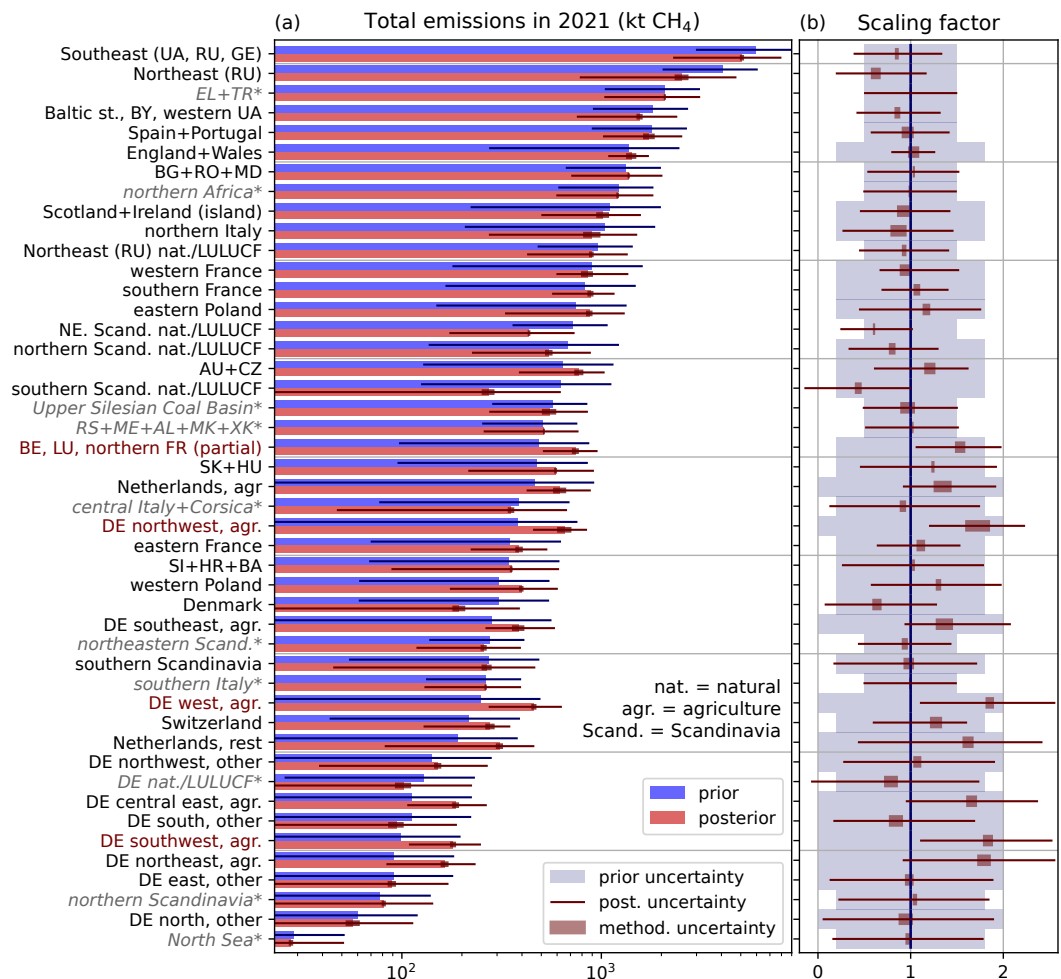

**Figure A1.** Prior and posterior emissions (a) and scaling factors (b) for all flux categories, ordered by prior emissions. Horizontal lines indicate 95 % confidence intervals. See Fig. 1 for the geographical definition of the flux categories and Fig. 2 for the resulting map of scaling factors. (a) If no sector is explicitly specified, the flux categories contain all anthropogenic fluxes excluding LULUCF. For flux categories marked with an asterisk, the inversion does not reduce the absolute uncertainty. Thus, reliable information is only gained by our inversion for flux categories without asterisk (see Sect. 2.6). Red color of the category names indicates a statistically significant increase of emissions. (b) Scaling factors are the raw results of our inversion, though here they are already combined for the whole year. The posterior scaling factor is defined as the center of the methodological uncertainty range indicated by brown boxes.

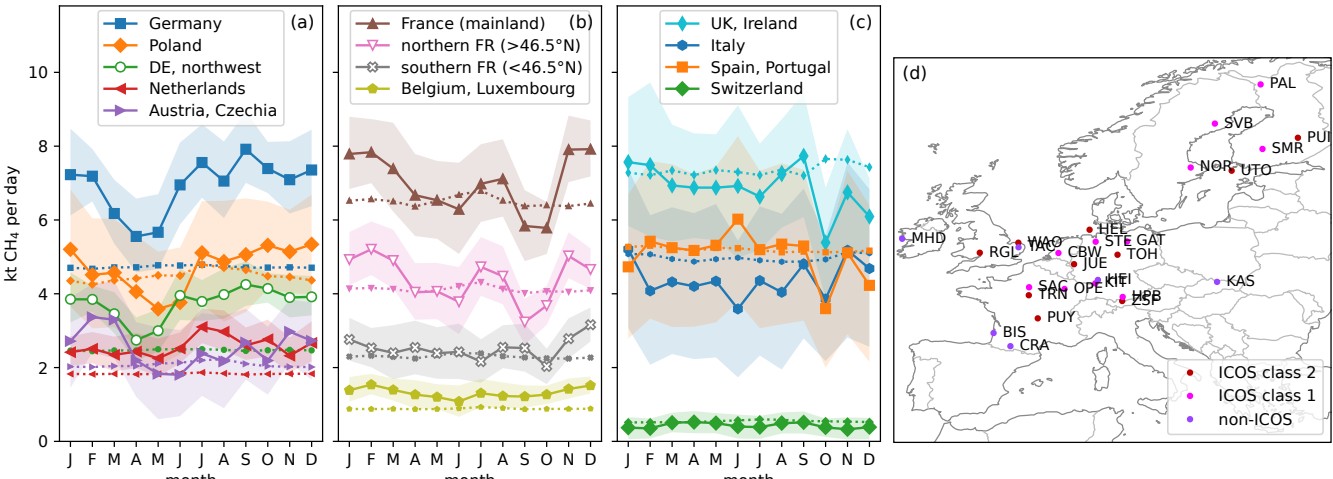

**Figure A2.** (a–c) Seasonal cycle when using only observations from stations that were active during the whole year. We select those stations and sampling heights, for which we used at least two data points per day on at least 20 days of each month in 2021 in our main inversion. This selects 27 stations shown in (d) with $8.3 \cdot 10^4$ data points for the inversion, compared to 50 stations with $1.29 \cdot 10^5$ data points in the reference case (compare Fig. 4). Colored areas show the posterior uncertainties (95 % confidence intervals), which were computed without excluding individual stations from the inversion and are therefore smaller than in Fig. 4. Prior emission rates are shown as dotted lines with small markers.

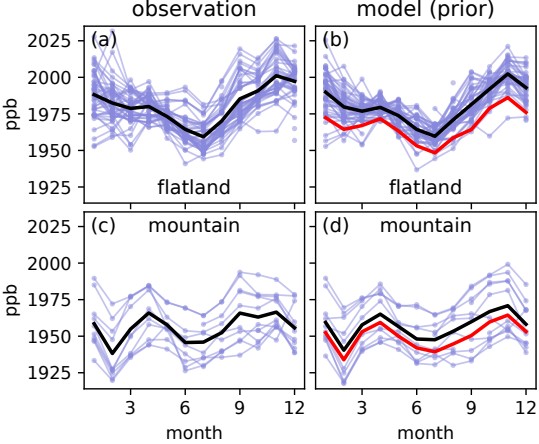

**Figure A3.** Seasonal cycle in observations at stations with elevation below 500 m above sea level (a, b) and above 1000 m (c, d), supplementary to the discussion in Sect. 3.2. Thin blue lines represent the 10 % quantile of each month, station, and sampling height for (a, c) observations and (b, d) model predictions (prior). The 10 % quantile is chosen to minimize the effect of local pollution. Thick black lines indicate the mean of all selected stations and sampling heights. Thick red lines in (b) and (d) show the 10 % quantile of the modeled far-field concentration. The flatland stations show a pronounced seasonal cycle with minimum in summer for both model and observations. This cycle is dominated by the contribution of the far field. The mountain stations have a weaker seasonal cycle.

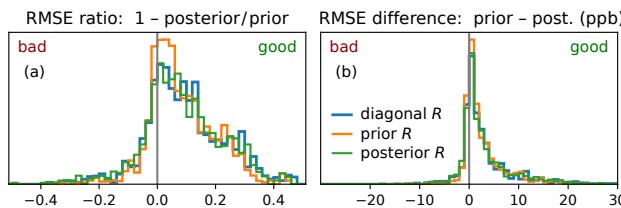

**Figure A4.** Statistics of the relative (a) and absolute (b) improvement of the model–observation mismatch at independent validation stations for different choices of the error covariance matrix $R$ discussed in Part 1 (Bruch et al., 2025a). The figure is analogous to Fig. 5, where the visualization and the data selection is explained. Here, we distinguish three inversion methods that differ in how $R$ is constructed, as introduced in Sect. 2.5 of Part 1. No clear advantage of one method over the others can be seen. The diagonal $R$ inversion has the lowest posterior RMSE at validation sites, followed by the posterior $R$ and prior $R$ inversion, but the differences are not statistically significant.

## Appendix B: Averaging kernel matrices

As introduced in Sect. 3.4.2, the averaging kernel matrices $A^{\text{emis}}$ and $A^{\text{scaling factors}}$ estimate the change in the posterior when changing the truth, $A^{\text{emis}} = \partial e^{\text{post}} / \partial e^{\text{truth}}$ where $e$ denotes the vector of emissions. Here, we summarize how these matrices are estimated using either the error covariance matrices $B$ and $B_{\text{post}}$ or the statistics from inversion runs with synthetic truth.

### B1 Analytic estimate using error covariance matrices

We first estimate the sensitivity of the posterior scaling factor to the true emissions under the assumption that the transport model, far field, observations, and the a priori spatial distribution within each flux category are perfect. Under these idealized assumptions, the model–observation mismatch for given scaling factors $s$ is $\mu(s) = y - Hs - x^{\text{ff}} = H(s^{\text{truth}} - s)$ where $s^{\text{truth}}$ denotes the true scaling factors. Our "prior $R$" inversion will now maximize

$$P(s) \propto \exp\left[ -\tfrac{1}{2}(s - s^{\text{truth}})^{\top} H^{\top} R^{-1} H (s - s^{\text{truth}}) - \tfrac{1}{2}(s - s^{\text{prior}})^{\top} B^{-1}(s - s^{\text{prior}}) \right] \tag{B1}$$

$$\propto \exp\left[ -\tfrac{1}{2}(s - s^{\text{post}})^{\top} B_{\text{post}}^{-1}(s - s^{\text{post}}) \right]. \tag{B2}$$

This yields $s^{\text{post}} = s^{\text{prior}} + A(s^{\text{truth}} - s^{\text{prior}})$ with the averaging kernel $A = I - B_{\text{post}} B^{-1}$ and the posterior error covariance matrix $B_{\text{post}}^{-1} = H^{\top} R^{-1} H + B^{-1}$ (Rodgers, 2000). Knowing $B$ and $B_{\text{post}}$, we can compute the averaging kernel $A$ to estimate how the posterior scaling factors depend on the true scaling factors.

### B2 Statistical estimate using synthetic experiments

In the statistical approach, we estimate the sensitivity of posterior scaling factors $\xi := s^{\text{post}} - s^{\text{prior}}$ to changes in the synthetic truth $\zeta := s^{\text{truth}} - s^{\text{prior}}$ using 100 synthetic experiments with random synthetic truth $s^{\text{truth}}$. Given a sample of $N$ realizations $\{\xi^n\}_n$ and $\{\zeta^n\}_n$, we aim to find the scaling factor averaging kernel matrix $A$ that solves

$$A = \underset{A'}{\arg\min} \sum_{n=1}^{N} \| \xi^n - A' \zeta^n \|^2. \tag{B3}$$

For $\|x\|^2 = \sum_i x_i^2$, differentiation by $A'_{ij}$ yields $0 = \sum_{n=1}^{N} \zeta_j^n (\xi^n - A\zeta^n)_i$ for all $i, j$ and thereby

$$A = \Xi Z^{-1}, \quad \Xi_{ij} = \sum_{n=1}^{N} \xi_i^n \zeta_j^n, \quad Z_{ij} = \sum_{n=1}^{N} \zeta_i^n \zeta_j^n. \tag{B4}$$

Equation (B4) was used to produce panels (c) and (d) of Fig. 7.

## Appendix C: Relevance of absolute prior uncertainty in sector attribution

When observations can detect a change in total emissions but cannot distinguish between different emission sectors, the sector-resolving inversion will change the sectoral distribution based on the prior uncertainties. To understand this problem qualitatively, we consider the worst case: We assume that fluxes from all sectors are uncorrelated in the prior but $100\%$ spatially

correlated such that they cannot be distinguished in the inversion. The a priori probability density for an emission vector $e$ of sector emissions $e_i$ is

$$P(e) \propto \exp\left[-\tfrac{1}{2}\sum_i (e_i - e_i^{\text{prior}})^2 \sigma_i^{-2}\right], \tag{C1}$$

where $\sigma_i$ denotes the a priori standard deviation of $e_i$. The inversion will estimate the total emissions $e_{\text{tot}}^{\text{post}}$ such that the a posteriori probability density $P(e|y)$ is maximized. But by assumption, these observations do not distinguish between sectors such that the a posteriori probability density fulfills $P(e|y) \propto P(e)$ as long as $\sum_i e_i$ is fixed. We thus obtain the posterior emissions of the sectors by maximizing Eq. (C1) with the constraint $\sum_i e_i = e_{\text{tot}}^{\text{post}}$. By introducing a Lagrange multiplier, one can show[2] that this implies

$$e_i - e_i^{\text{prior}} = \alpha \sigma_i^2, \qquad \alpha = \frac{e_{\text{tot}}^{\text{post}} - e_{\text{tot}}^{\text{prior}}}{\sum_i \sigma_i^2}. \tag{C2}$$

This shows that sectors with larger absolute a priori uncertainty are disproportionally stronger corrected. Applied to our emission estimates for Germany, this implies that if the observations were unsuitable for distinguishing sectors, the inversion would attribute up to 95 % of the changes in total fluxes to the agriculture sector, which is responsible for 69 % of the total a priori emissions. Fortunately, this worst case scenario is not realistic because the observations do contain information on the different sectors as indicated e.g. by Figs. 6 and 7. But a tendency remains to correct the agriculture stronger than the other sectors.

**Appendix D: Attempt to distinguish five sectors in Germany**

Our setup for the transport simulation was designed to separate five sectors in Germany: agriculture, natural plus LULUCF, waste, public power, and the sum of all other sectors ("other"). We try to distinguish these sectors in a separate inversion run, in which each of these sectors is scaled separately (sensitivity tests 506 in Part 1 (Bruch et al., 2025a)). This inversion uses 19 separate scaling factors in Germany instead of 11. We find no notable changes in the posterior emissions compared to our reference setup, in which we combined waste, public power, and other into one larger sector "non-agr." However, the uncertainties and the averaging kernels change considerably. We assume an a priori $2\sigma$ uncertainty of $\pm 100\,\%$ for each sector-resolving flux category. Thus, splitting the total fluxes in more uncorrelated flux categories reduces the a priori uncertainty of the total fluxes.

Figure D1 shows the averaging kernel matrices (introduced in Sect. 3.4.2 and Appendix B) for the inversion when separating five sectors. These matrices indicate that waste, public power, and "other" cannot be distinguished: The corresponding columns Fig. D1(a) are approximately equal. Thus, trying to distinguish these sectors does not provide any additional information. By comparing the row and column for "non-agr." to Fig. 7, we identify drawbacks of the attempt to distinguish smaller sectors. When trying to distinguish five sectors, the false attribution of emissions to the agriculture sectors is more severe than when distinguishing only three sectors (48 % compared to 28 %). Consequently, the expected error reduction in the combined non-

---

[2]We define $L(e, \lambda) = -\tfrac{1}{2}\sum_i (e_i - e_i^{\text{prior}})^2 \sigma_i^{-2} + \lambda(e_{\text{tot}}^{\text{post}} - \sum_i e_i)$ and require $\frac{\partial L}{\partial e_i} = 0, \frac{\partial L}{\partial \lambda} = 0$.

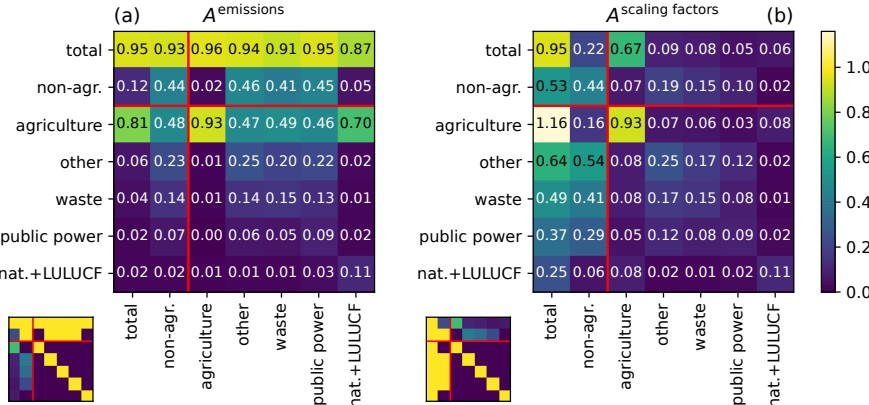

**Figure D1.** Averaging kernel matrices of German sector emissions (a) and the corresponding scaling factors (b) when trying to distinguish sectors waste, public power and other, estimated using the posterior error covariance matrix. Small matrices at the bottom indicate the ideal result. See Fig. 7 for an explanation of the representation. Panel (a), third row, shows that increasing true emissions in any sector is expected to cause higher posterior agriculture emissions with a false attribution of 46 % to 70 %. The same row in panel (b) shows that when looking at relative changes in the emissions, the influence of the false attribution on the agriculture sector is not very large.

agriculture sectors (excluding natural plus LULUCF) is better when considering only three sectors. Qualitatively, this is what we expect from Appendix C for cases where the observations are insufficient to distinguish the considered sectors.

*Author contributions.* VB and TR conceptualized the inversion method. VB implemented the inversion method and wrote the original draft together with AKW. TR configured the transport model. TR and BE interpolated the a priori flux data which BE collected. DJCO organized data streams of $CH_4$ concentrations and observations. JF, BM, AMB, DJCO, TR and VB contributed to testing and tuning the transport model. NB contributed to the model–observation comparison. AKW supervised and coordinated the project. All authors reviewed and edited the manuscript.

*Competing interests.* The authors declare that they have no conflict of interest.

*Acknowledgements.* In our simulations we use modified Copernicus Atmosphere Monitoring Service information and ECCAD products for initial and lateral boundary conditions, and for a priori fluxes. We thank Stefan Feigenspan, Christian Mielke, Theo Wernicke, John Akubia and Roland Fuß for helpful discussions and providing a priori emission fields. We thank Roland Potthast, Frank-Thomas Koch, Christoph Gerbig, Dominik Brunner, Michael Steiner, David Ho, Thomas Kaminski, Hannes Imhof and our partners in the ITMS project for very helpful and inspiring discussions. We also wish to thank Peter Bergamaschi, Aurélie Colomb, Martine De Mazière, Lukas Emmenegger, Dagmar Kubistin, Irene Lehner, Kari Lehtinen, Markus Leuenberger, Cathrine Lund Myhre, Michal V. Marek, Simon O'Doherty, Stephen

M. Platt, Christian Plaß-Dülmer, Francesco Apadula, Sabrina Arnold, Pierre-Eric Blanc, Dominik Brunner, Huilin Chen, Lukasz Chmura, Łukasz Chmura, Sébastien Conil, Cédric Couret, Paolo Cristofanelli, Grant Forster, Arnoud Frumau, Christoph Gerbig, François Gheusi, Samuel Hammer, Laszlo Haszpra, Juha Hatakka, Michal Heliasz, Stephan Henne, Arjan Hensen, Antje Hoheisel, Tobias Kneuer, Eric Larmanou, Tuomas Laurila, Ari Leskinen, Ingeborg Levin, Matthias Lindauer, Morgan Lopez, Ivan Mammarella, Giovanni Manca, Andrew Manning, Damien Martin, Frank Meinhardt, Meelis Mölder, Jennifer Müller-Williams, Steffen Manfred Noe, Jarosław Nęcki, Mikaell Ottosson-Löfvenius, Carole Philippon, Joseph Pitt, Michel Ramonet, Pedro Rivas-Soriano, Bert Scheeren, Marcus Schumacher, Mahesh Kumar Sha, Gerard Spain, Martin Steinbacher, Lise Lotte Sørensen, Alex Vermeulen, Gabriela Vítková, Irène Xueref-Remy, Alcide di Sarra, Franz Conen, Victor Kazan, Yves-Alain Roulet, Tobias Biermann, Marc Delmotte, Daniela Heltai, Ove Hermansen, Kateřina Komínková, Olivier Laurent, Janne Levula, Chris Lunder, Per Marklund, Josep-Anton Morguí, Jean-Marc Pichon, Martina Schmidt, Damiano Sferlazzo, Paul Smith, Kieran Stanley, Pamela Trisolino and Giulia Zazzeri for providing the atmospheric observations. VB, DJCO, NB and AMB acknowledge funding by the German Federal Ministry for Education and Research (BMBF) in the ITMS project (grant 01LK2102B) as well as BE (grant 01LK2104A). Map plots were made with Natural Earth.

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
