# Peer review of "German methane fluxes estimated top-down using ICON-ART – Part 1: Ensemble-enhanced scaling inversion"

_EGUsphere, 2025_

## Referee Comment (RC1)

Review of

**German methane fluxes in 2021 estimated with an ensemble-enhanced scaling inversion based on the ICON–ART model (2025-1464)**

Valentin Bruch, Thomas Rösch, Diego Jiménez de la Cuesta Otero, Beatrice Ellerhoff, Buhalqem Mamtimin, Niklas Becker, Anne-Marlene Blechschmidt, Jochen Förstner, and Andrea K. Kaiser-Weiss

For Atmospheric Chemistry and Physics

**1 Overview**

This paper describes the regional methane inversion system developed by the DWD. It covers multi-sectors of methane emissions over Europe, focusing on Germany. I am more familiar with inversion systems that try to estimate the initial conditions and the fluxes simultaneously, while this system estimates a so-called "far-field" first and followed by the fluxes, the latter of which appear to be represented as "scaling factors". Hence, I had to try hard to understand how this system works.

The system is very complex and the study is clearly very comprehensive, examining how observations can be pre-processed and filtered, how the uncertainties (R-matrix) can be estimated, how the posterior errors are examined. It studies the sensitivities of the posteriors to the truth (using the averaging kernels), and sensitivities to a very wide range of configurations of the system.

The system appears to work and is clearly very useful, but there are some weaknesses of the paper, mainly in how it is presented. I would very much like to see the work published, but I feel there needs to be some major re-writing. In particular, after reading the paper, I feel the following overall points deserve attention.

- The system description is not in the best order. The paper would benefit from a system flow chart of the steps and a more logical order of descriptions.

- There is arguably too much information given in the appendices (there are nine appendices). The work shown in some of the appendices is important, such as additional experiments and sensitivity tests. In fact I think that the paper is too long. I would suggest having a two-part paper, part I for the system description, and part II for the results. I feel that the appendices should just contain things like derivations and supplementary tables. The mixture of results between the main part of the paper and the appendices caused me to lose track of the flow of the paper.

- There appears to be some contradictory statements made in the paper. This may be just my misunderstanding, but an example is the description of how the model uncertainties are found. There seems to be one explanation in Section 2.4, another one in Sect. 4.2, and another in Sect. 5.4.2. See also detailed points 10, 20, and 27. More examples are given in my detailed comments.

- I'm not completely sure how the ensemble comes into the main inversion. I can see how an ensemble is used to help define the model uncertainties in Sect. 2.4, and in some of the supplementary statistics, but I am not sure how it is used in the main inversion (Eq. (1)), given the paper's title suggests that the ensemble is used for that. See also detailed points 8 and 10.

- I am unclear about the structure of the main control variable, $s$, especially as these are called scaling factors. I cannot find what these factors actually multiply in order to lead to predictions of the model's observations via $Hs$. See also detailed points 2, 7e, 7f, and 41a.

- I am unclear how individual flux categories can be distinguished from the observations (apart from using information in the ratio of the background error statistics, which divides the posterior fluxes according to how these are prescribed). See also detailed points 29 and 40a.

- It is not clear to me how the ICON model is used in the inversion itself, and whether any chemical processes are simulated as part of the forward model, $H$. See also detailed points 1, 1b, 12, and 13.

As I mentioned above, the work should be published, as there are many positives of the work, but after some major restructuring and rewriting. I give a detailed list of scientific issues in Sect. 2 and minor points in Sect. 3. Note that although there are a lot of points in my report, I believe that each can be addressed fairly quickly. I don't think any more experiments necessarily need to be run.

In my report the text of the paper is referenced by Lx (line x, where x uses the guide down each page), or by section/figure/equation/table number, and I often quote from the paper to help refer to the part that the comment refers to. Text that I suggest to be added is underlined, and text that I suggest to be removed is scored out.

**2   Scientific points**

1. L49-51: "Atmospheric transport is simulated using the numerical weather prediction model ICON (Zängl et al., 2015) extended with the module for Aerosol and Reactive Trace gases (ART) (Rieger et al., 2015; Schröter et al., 2018) with a spatial resolution of 6.5 km."

    (a) I'm just wondering how the ICON model is used to provide the transport winds. As well as providing tracer transport winds, is the ICON model actually simulating the winds (on-line) at the same time as advecting the methane, or is the ICON model used to pre-determine winds, which are then used by the inversion system to do the transport? The former would provide very high-temporal resolution winds (and include sub-grid-scale transport), without the need to store them, which would be ideal, but very expensive. If ICON is simulating the winds on-line, then it must be run with a data assimilation scheme to keep them realistic. I think the authors could say a little about how this is done (although see also point 12 below).

    (b) Are any chemical processes (such as the reaction with OH and methane oxidation included? L398-399: "Another potential contribution to the seasonal cycle could arise from neglecting the OH sink of $CH_4$ in our limited domain." suggests not. Could this be an issue over the timescale of the inversions? The authors should at least say that this is an assumption made by their system

2. L72-74: "The categorized fluxes are scaled to minimize the mismatch between model prediction and observed concentrations. Thus, the inversion result consists of one scaling factor for each flux category. The a priori fluxes multiplied by the scaling factors yield the a posteriori fluxes." By "... one scaling factor for each flux category.", do the authors mean one for all positions and time, or one per flux category per position and per time (obviously with relevant correlation scales)?

3. L91-92: "Each ensemble member uses slightly different but equally likely parametrizations and meteorological initial and boundary conditions." Does each member use slightly different driving winds too?

4. Sect. 2.2 and Appendix B: I'm not really sure what is meant by the far field. I assume it's a smooth correction to the methane field, determined before the inversion, and is a function of space and time. Is that right? Looking at Eqs (1) and (B1), it looks like it exists only at observation locations. See also point 35 below.

5. L100-102: "To minimize potential biases arising from imperfect boundary conditions, we construct a correction field which is added to the modeled far-field concentration in the whole domain after the transport simulation." The terms "far-field" and "whole domain" sound contradictory to me.

6. L102-103: "We require this correction field to be smooth on large length and time scales, chosen in our case as ..." This sounds like a tautology to me: surely "smooth" means "large scale"?

7. Sect. 2.3:

    (a) How is $B$ determined as used in Eq. (1)?

    (b) What are the observations of? Appendix A lists the stations, but I cannot find anywhere whether the observations are of methane or of the flux itself. What are the instrument types and instrument precisions? This would be good to know even though they are claimed to be negligible to the model error (L125). I think it should be stated as early as possible in the paper that the observations are of methane.

    (c) What is the time window and how is information propagated from one time window to the next?

    (d) As the initial conditions of the methane field are not mentioned in the cost function, Eq. (1), it looks like the initial conditions are not adjusted as part of the inversion. Is that right? Perhaps this is the purpose of the far-field correction step? If so, please make this clear. If the initial conditions are sufficiently wrong, then this would have an impact on the quality of the inversion.

    (e) Given that the control variable in Eq. (1) is $s$ (a scaling factor), this suggests that it should multiply something. What is the field that it multiplies? See also point 41a below.

    (f) What is the structure of the field $s$? Is it represented on a grid, or does it multiply some basis functions? Field $s$ must, presumably, be associated with the different categories of source field (a number $>40$ is mentioned in D2, e.g.).

8. Eq. (1): Given that the title of the paper talks about an ensemble, is Eq. (1) minimised with respect to each ensemble member? Or is the ensemble just used for the procedure described in Sect. 2.4 (see point 10 below)?

9. L111-112: "In the first term, the vector $y$ of observed concentrations is compared to the model prediction, which consists of the transported fluxes $Hs$ and the modeled far field $x^{\mathrm{ff}}$."

    (a) How can fluxes be transported? Do the authors mean "transported methane emitted by the fluxes"?

    (b) In Eq. (1) and in the above quote, $x^{\mathrm{ff}}$ is the "far field". Is this the same as field $c$ in Appendix B? If so, I would recommend that the same symbol is used in all parts of the paper. If not, I would recommend that they are not referred to as the "far field".

10. Sect. 2.4: The $R'$ matrix in Eq. (2) is determined from an ensemble.

    (a) Is this the same ensemble that is (possibly) used in the main inversion (but see point 8 above)? How is it initialised?

    (b) L132: "... added to each observation accounting for any representativity error." I think representativity error is something different, unless sub-grid-scale processes are included in the divergence of the ensemble members.

11. L163-168: "Plumes caused by high emissions in a small area require special treatment to avoid a potential bias in the inversion due to the so-called double penalty issue (Vanderbecken et al., 2023). In cases where our model falsely predicts that the plume reaches an observation site, the inversion will reduce the emissions to improve the agreement with the observation. In the opposite case, when the model fails to predict that a plume reaches the observation, the inversion will not change the plume emission amount but will wrongly increase emissions in other areas instead. This can cause systematic underestimation of fluxes from localized plumes." I very much like this interpretation.

12. L177-178: "The meteorological initial and lateral boundary conditions used to drive our transport model are taken from the archive of DWD's operational numerical weather prediction (NWP), which also employs the ICON model." This statement partially addresses point 1 above, but I am still unsure whether it is ICON itself – or another model that just uses ICON-derived winds – that is used for the $H$ operator in Eq. (1).

13. L184-185: "In contrast to the meteorological fields, the $CH_4$ concentrations are only transported and never re-initialized." Is it just transportation, or are chemical reactions included too?

14. L188: "We ensured mass conservation when interpolating to our model grid." How is this done? Is it by multiplying the interpolated fluxes by a factor to ensure that the total flux is the same after interpolation?

15. L189-190: "Anthropogenic fluxes excluding LULUCF are split further into 12 GNFR sectors (gridded aggregated NFR, nomenclature for reporting, Veldeman et al. (2013)), . . ."

    (a) GNFR is not defined (I assume GNFR = Gridded NFR, but I don't know what NFR stands for).

    (b) I don't really understand the text in the brackets.

16. L197: "These emissions are missing in our a priori estimate." Does this represent a low bias in the a-priori?

17. L219-221: "Observations within the planetary boundary layer are most representative in the afternoon hours whereas measurements at high mountains have less local influence at night time (Bergamaschi et al., 2015). We therefore use the time windows 23 h to 5 h (local mean time) for stations on high mountains and 11 h to 17 h for all other stations." If observations on high mountains have less influence over night, why only use them during 23 h to 5 h?

18. L227-229: "In the last filtering step – step 5 in Table 2 – we exclude data points with extreme mismatch between model and observations of more than 200 ppb. Data points where the observations are more than 20 ppb below the model-predicted far field are also discarded."

    (a) I assume by "model", the authors are referring to the a-priori?

    (b) For the first (extreme) condition is this written mathematically as $\left|y - Hs - x^{\mathrm{ff}}\right| > 200\mathrm{ppb}$ or $|y - Hs| > 200\mathrm{ppb}$ to discard? Perhaps this is better explained as an explicit inequality?

    (c) For the second condition is this, mathematically $y - x^{\mathrm{ff}} < -20\mathrm{ppb}$ to discard? Again, perhaps this is better explained as an explicit inequality?

19. Section 4.1.2, point (ii):

    (a) By advecting a tracer, how is it possible to set a lifetime for that tracer? One would know only the tracer concentration in any grid box, not how long it has been since that tracer was released (and anyway is likely to be comprised of tracers that have a variety of emission times).

(b) The statement that tracers have a lifetime and that "no CH$_4$ is lost" seem contradictory to me.

20. Section 4.2: I'm very confused as there seems to be two prescriptions for how the R-matrix is determined – this section and Sect. 2.4. See also point 27 below.

21. Section 4.3: Is any cycling done between the monthly inversions? That is, does the posterior of one month become the prior of the next?

22. Section 4.4:

    (a) L316: "In each inversion time window, we consider uncorrelated a priori scaling factors ..." and L319-320: "... , and within Germany categories describing the same sector have an a priori uncertainty correlation of 50 %". These two statements seem contradictory.

    (b) L316-320: "In each inversion time window, we consider uncorrelated a priori scaling factors with a two standard deviation ($2\sigma$) uncertainty of 80 % for most flux categories, corresponding to a 95 % confidence interval of $\pm80$ %. Throughout this paper, uncertainties will denote two standard deviations or 95 % confidence intervals. Categories resolving emission sectors have a higher prior $2\sigma$ uncertainty of $\pm100$ %, and within Germany categories describing the same sector have an a priori uncertainty correlation of 50 % (e.g., ...)" There are (presumably) two quantities represented by percentages here, the confidence intervals (which represent percentage of the PDF volume), and the methane quantity itself (presumably represented as a percentage of some value). If this is correct could there be a better way to describe these to avoid confusion?

23. L333-345: "Additionally, we combine the two variants of inversion (prior-R and posterior-R, see Sect. 2.4.2) by taking the arithmetic mean of the two separate inversion results, arriving at the combined scaling factors." The posterior error statistics should just be a function of the inverse Hessian of the inversion. Why the need to refer to the R-matrices? A similar mention of the R-matrices is made in L4-5 of the caption of Fig. 5.

24. L348-349: "Figure 4 presents an overview of (a) the a priori CH4 fluxes, (c) the resulting scaling factors, and the respective uncertainties (b, d), all accumulated over the year 2021." Presumably "averaged" rather than "accumulated".

25. L416-417: "Other filtering parameters such as the number of sampling heights per station (case 202) and ..." I don't understand what is meant by, "the number of sampling heights per station", given that the height of an observation station is (presumably) fixed.

26. L418-420: "Neglecting extreme outliers has only a small effect (cases 206, 207), but limiting the influence of outliers by increasing their uncertainty has a considerable impact (cases 208, 209)." One would expect that neglecting extreme outliers is just the limiting case of increasing their uncertainty to infinity. It might be counter-intuitive then that increasing their uncertainties has a large impact, then a small impact when their uncertainties are increased further to infinity.

27. L428-429: "... and the uncorrelated additive uncertainty $\sigma_{\mathrm{const}}$ of each data point (cases 309, 310)." Perhaps I have missed it, but I cannot find any reference to $\sigma_{\mathrm{const}}$ in Sects. 2.4 or 4.2. This adds to my confusion about how the R-matrix is determined. See also point 20 above.

28. L471: " Each vector of scaling factors defines  a synthetic truth, ..." (These suggested changes distinguishes considering all scaling factors together in one ensemble-based inversion.)

29. L484-485: "Within Germany, we distinguish agriculture from other emissions. The ability to distinguish sectors can be described by averaging kernel matrices which estimate ..." Putting the B-matrix aside, how can different sectors be distinguished from observations of methane? See also point 40a below.

30. Section 5.6: This section is about simulated transport errors, but I cannot find a description of how transport errors are simulated. Is noise added to the winds e.g.?

31. Figure 10: I'd recommend adding the 1:1 line as a guide. See also point 39c below.

32. L515-516: "Localized sources that cause a strong plume are underestimated by both methods, though the bias is reduced in the posterior-R inversion as predicted ..." I assume it is the posterior fluxes that are plotted in Fig. 10, using the two R-inversion methods, with and without the far-field correction (and not their errors). Figure 10 (and Fig. F2, see point 39c below) therefore compares the posterior values for each relative to the prior. How is it possible therefore to tell whether a strong plume is underestimated and what the biases are?

33. L537: "Firstly, we find that our top-down $CH_4$ emission estimates are significantly higher than reported for Germany." Although, looking at Fig. 5, the NIR estimate is well within the 95% confidence interval of the posterior.

34. L619-620: "The increasing availability of satellite data is especially interesting for constraining concentrations and emissions in less observed regions, such as near the boundaries of our domain." This seems an obvious extension of the work given the wide range of total column methane retrievals available from satellites. This may require careful tuning though to remove potential biases.

35. Appendix B, Eq.(B2): I am concerned that the formula (B2) is not formally correct in one detail (in the final step). Let's go through the maths to demonstrate

$$\text{penalty}(c) = (x + c - y)^{\mathsf{T}} P^{\mathsf{T}}(P\tilde{R}P^{\mathsf{T}})^{-1}P(x + c - y) + c^{\mathsf{T}}P^{\mathsf{T}}(P\tilde{C}P^{\mathsf{T}})^{-1}Pc.$$

(Incidentally, all other cost functions in the paper have a factor of 1/2 except this one. This does not make any difference to the outcome, but the authors might wish to be consistent by including the 1/2 factor here.) The gradient is

$$\nabla\text{penalty}(c) = 2P^{\mathsf{T}}(P\tilde{R}P^{\mathsf{T}})^{-1}P(x + c - y) + 2P^{\mathsf{T}}(P\tilde{C}P^{\mathsf{T}})^{-1}Pc = 0 \text{ for min.}$$

This agrees with (B3) of the paper. This can be rearranged:

$$\left(P^{\mathsf{T}}(P\tilde{R}P^{\mathsf{T}})^{-1}P + P^{\mathsf{T}}(P\tilde{C}P^{\mathsf{T}})^{-1}P\right)c = P^{\mathsf{T}}(P\tilde{R}P^{\mathsf{T}})^{-1}P(y - x)$$
$$P^{\mathsf{T}}\left((P\tilde{R}P^{\mathsf{T}})^{-1} + (P\tilde{C}P^{\mathsf{T}})^{-1}\right)Pc = P^{\mathsf{T}}(P\tilde{R}P^{\mathsf{T}})^{-1}P(y - x)$$

We cannot simply cancel $P^{\mathsf{T}}$ from the left, but because $PP^{\mathsf{T}}$ is full rank we can multiply from the left with $P$ and then cancel $PP^{\mathsf{T}}$. Doing this and then developing:

$$\cancel{PP^{\mathsf{T}}}\left((P\tilde{R}P^{\mathsf{T}})^{-1} + (P\tilde{C}P^{\mathsf{T}})^{-1}\right)Pc = \cancel{PP^{\mathsf{T}}}(P\tilde{R}P^{\mathsf{T}})^{-1}P(y - x)$$
$$(P\tilde{R}P^{\mathsf{T}})\left((P\tilde{R}P^{\mathsf{T}})^{-1} + (P\tilde{C}P^{\mathsf{T}})^{-1}\right)Pc = P(y - x)$$
$$\left(I + (P\tilde{R}P^{\mathsf{T}})(P\tilde{C}P^{\mathsf{T}})^{-1}\right)Pc = P(y - x)$$
$$\left[P\tilde{C}P^{\mathsf{T}} + P\tilde{R}P^{\mathsf{T}}\right](P\tilde{C}P^{\mathsf{T}})^{-1}Pc = P(y - x)$$
$$P(\tilde{C} + \tilde{R})P^{\mathsf{T}}(P\tilde{C}P^{\mathsf{T}})^{-1}Pc = P(y - x)$$
$$Pc = P\tilde{C}P^{\mathsf{T}}\left[P[\tilde{C} + \tilde{R})P^{\mathsf{T}}\right]^{-1}P(y - x).$$

If we could eliminate $P$ from the left of each side then we would get (B2). We do the trick similar to above: first multiply by $P^\mathsf{T}$:

$$P^\mathsf{T} Pc = P^\mathsf{T} P\tilde{C}P^\mathsf{T} \left[ P[\tilde{C} + \tilde{R})P^\mathsf{T} \right]^{-1} P\left(y - x\right).$$

Unfortunately, $P^\mathsf{T} P$ is not full rank so we cannot obtain a unique equation for $c$ (but we can for $Pc$). This makes sense because the constraint described by (B1) applies only to those observation points selected by $P$. Since there is no a-priori constraint (to allow information to be available to not-selected observations), the not-selected observations will have no result from the above analysis. I think this requires some attention in the paper.

36. Appendix C: I didn't follow the reasoning in this appendix. For example, I note the following points.

   (a) The left hand side of (C1) seems strange to me. It is the probability of a real number (the mismatch) being realised. This doesn't make any sense to me. Normally a probability *density* would be used, e.g. $P(\mu = y - Hs^{\mathrm{prior}} - x^{\mathrm{ff}})d\mu$ to mean the probability between values of $\mu$ and $\mu + d\mu$.

   (b) What is the variable $dP_s$ and how does one arrive at (C1)?

   (c) The line in (C2) appears to integrate over the posterior. How does this relate to (C1)?

   (d) The term $H\tau + \mu$, which appears in (C2) has the following form, given that $\tau = s - s^{\mathrm{prior}}$ and $\mu = y - Hs^{\mathrm{prior}} - x^{\mathrm{ff}}$:

   $$H\tau + \mu = Hs - Hs^{\mathrm{prior}} + y - Hs^{\mathrm{prior}} - x^{\mathrm{ff}} = Hs - 2Hs^{\mathrm{prior}} + y - x^{\mathrm{ff}}.$$

   This doesn't seem correct to me.

   (e) The rest of the section does contain some useful analysis about the expected value and range of $\chi^2/N_{\mathrm{dof}}$, but the appendix should be explained more and a reference added.

37. Appendix D

   (a) L692: "We estimate the model uncertainty using a meteorological ensemble." Which model uncertainty and which quantity? I assume the authors are referring to the uncertainty in $s$?

   (b) L693: "Stronger emissions lead to stronger spatial gradients in the model concentrations ..." (suggested change to make distinct from gradients in state space).

   (c) L712-715: "When using a priori scaling factors to estimate the model uncertainty, we need only the total concentration $x_i^m(s^{\mathrm{prior}})$ for each ensemble member. Thus, only a single tracer field is required in the ensemble transport simulation. To fully compute $x_i^m(s)$ as function of s, the tracer categories need to be distinguished for each ensemble member, resulting in >40 tracer fields in the ensemble simulation." The only difference between the calculation of $x_i^m(s^{\mathrm{prior}})$ and $x_i^m(s)$ is that $s^{\mathrm{prior}}$ is replaced by $s$. How does this require that >40 different tracer fields are required? I guess that >40 different values of $s$ are chosen. What is the significance of the number 40? The difficulty to understand this may be connected to point 7f above.

   (d) In the above, what does $i$ stand for?

   (e) I don't really understand Eq. (D3). What is the distinction between $x_i^{mg} = x_i^m(P_g s^{\mathrm{prior}})$ and $HP_g s^{\mathrm{prior}}$ (they seem to be the same thing)? Making this distinction is probably the key to understanding (D3).

38. Table E1

(a) ID 101: "no time averaging", given the explanation column, shouldn't this read "change in time averaging"?

(b) Some of the test cases (rows in table) are ambiguous to me. For example, "200, fewer hours of day, use time window 12 h–16 h (0 h–4 h for high mountains)". Does it mean that the reference case used observations at all hours? Then again "201, all hours of day, no filtering by time of day, increase uncertainty inflation by factor 1.5" seems to imply that using observation at all hours is a test case different from the reference. I would recommend going through the entire list and making sure that each test case is unambiguous to the reader.

39. Appendix F

(a) Figure F1: I am confused why the truth (horizontal lines) should change with the test ID.

(b) L755-757: "Next, we test the effect of an underestimation or overestimation of all emissions. In case 20 of Fig. F1, all natural and LULUCF fluxes are reduced by 40 % in the truth, and cases 21 and 22 change all anthropogenic emissions excluding LULUCF by −20 % and +20 %, respectively." Are these experiments simply changing the true emissions (and the synthetic observations and a-priori values) and then repeating the inversion? I would've thought that such an experiment would be expected to have posterior fluxes that are consistent with the truth. Wouldn't a more interesting experiment be one with the underestimation or overestimation of all emissions in the a-priori, but with the truth (and synthetic observations) unchanged?

(c) Fig. F2, and text: It would be useful to draw the 1:1 line in each panel (see also point 31 above). It would then be seen that the when far-field correction is applied the posterior R-inversion does better than the prior R-inversion. I think it is best to describe fully the experiments being done and then show the results in the Figs. In the case of the text around Fig. F2 it is not until the end of the appendix that the reader learns that the "prior is underestimated compared to the synthetic truth."

40. Appendix G

(a) L765-766: "When observations can detect a change in total emissions but cannot distinguish between different emission sectors, the sector-resolving inversion will change the sectoral distribution based on the prior uncertainties." Out of curiosity, how could observations of methane make this distinction? Also mention is made in the main text at L365.

(b) L768: "The a priori probability density for an emission ..."

(c) L771: The symbol $P(e)$ is used to represent the posteriori probability density, but the same symbol is used for the prior density in Eq. (G1). Although this is often done when describing probabilities, one can distinguish the prior from the posterior by its argument. The prior is often written as $P(e)$ and posterior as the conditional density $P(e|y)$. The statement on L770-771 needs to refer to the posterior.

41. Appendix H

(a) L786-787: "We first estimate the sensitivity of the posterior scaling factor to the true emissions under the assumption that the transport model, far field, and the flux pattern within each flux category are perfect." Flux patterns are referred to, but not defined anywhere. I suspect addressing this point will help the reader understand point 7e above also.

(b) L788: $\mu = y - Hs^{\mathrm{prior}} - x^{\mathrm{ff}}$. Let $y = Hs^{\mathrm{truth}} + x^{\mathrm{ff}} + \epsilon^y$ (where $\epsilon^y$ is the observation error). Then, $\mu = H(s^{\mathrm{prior}} - s^{\mathrm{truth}}) + \epsilon^y$. The expression given in the paper doesn't have $\epsilon^y$, which suggests that the authors are making the additional assumption that the observations are perfect.

(c) Eq. (H1): Given that the above expression is used to substitute for $y - Hs - x^{\text{ff}}$ (and not $y - Hs^{\text{prior}} - x^{\text{ff}}$ as used in point 41b above), why not simply present the previous expression with $s^{\text{prior}} \rightarrow s$?

42. Appendix I

(a) Fig. 11: There is no explanation of the distinction between the two panels in the caption. The only difference is (a) is labeled $A^{\text{emissions}}$ and (b) is labeled $A^{\text{scaling factors}}$. I assume that (a) is computed from (b)? What are the smaller matrices at the bottom of each panel?

**3  Presentational points and very minor points**

1. The appendices are referenced in a different order to their placement. For example, appendix B is referenced first. It makes logical sense to place the appendices in the order that they are first referenced in the main text.

2. Table 1: As there are many acronyms, the caption of Table 1 might be a good place to (re)define acronyms. Incidentally, I cannot see where GNFR is defined anywhere.

3. L201: reference to Fig. 3, I would recommend moving Fig. 3 to Fig. 1 as it referenced first. Also in Fig. 3: "A white  ellipse marks the Upper Silesian Coal Basin, ..."

4. Sections 4.1.1 and 4.1.2: I got very lost trying to follow exactly what was done here. I would recommend that these sections are rewritten, possibly with a table to help the reader see exactly how different regions and sectors are combined, etc.

5. L466, suggested change: "We therefore use experiments with  synthetic data, i.e., define truth and ..."

6. L110: "...  penalizes ..."

7. Table E1 caption: "Overall, we see that most test_s_ have an impact ..."

8. L194: "For Germany, the a priori LULUCF fluxes obtained from _the_ Thünen Institute cover ..."

9. L207: "... in model coordinates  are derived from the station sampling height_s_ and the modeled station elevation_s_, depending on the ..."

10. L214: "We start by  smoothing both observations and ..."

11. L372: "In France, Belgium, and Switzerland, the inversion scales flux categories _that_ overlap multiple countries[2] ."

12. Table E1, caption, L4: "Overall, we see that most test_s_ have an impact ..."

13. L568-569: "Though simulating emissions and transport in a large domain, we can only provide reliable emission estimates for selected countries (compare Fig. 5)." I think there is a slight problem with the beginning of the sentence. Possibly the authors mean either, "Th_r_ough simulating emissions and transport in a large domain, we can only provide reliable emission estimates for selected countries (compare Fig. 5)." or, " Although _we are_ simulating emissions and transport in a large domain, we can only provide reliable emission estimates for selected countries (compare Fig. 5)."

14. L579: "... is more likely to increase the far field rather than  to decrease it, leading to a bias towards ..."

15. L588: "Moreover, adjusting few_er_ degrees of freedom may ..."

---

## Author Comment (AC1)

**Reply to Referee #1 of egusphere-2025-1464**

**Valentin Bruch**

October 23, 2025

We thank the referee for the detailed reading and constructive comments which helped improve the manuscript. To best address the referees' comments and after consolidation with the editor, the manuscript was split into a Part 1 and Part 2, explained below.

The structure of this reply follows the structure of document provided by the referee, but we added Sect. 4 on a mistake we noticed in the initially submitted manuscript.

**1 Overview**

- 1. Referee comment: "The system description is not in the best order. The paper would benefit from a system flow chart of the steps and a more logical order of descriptions."
  - This is a very valuable comment. We have adjusted the order of the system description by merging the Sections 2 and 4 of the manuscript in Section 2 of the new Part 1. We added a flow chart of the data streams in Part 1 (Fig. 5).
- 2. Referee comment: "There is arguably too much information given in the appendices (there are nine appendices). The work shown in some of the appendices is important, such as additional experiments and sensitivity tests. In fact I think that the paper is too long. I would suggest having a two-part paper, part I for the system description, and part II for the results. I feel that the appendices should just contain things like derivations and supplementary tables. The mixture of results between the main part of the paper and the appendices caused me to lose track of the flow of the paper."

We gladly took up this suggestion and split the paper into two parts. **Part 1** now represents the system description, as suggested. Here, we also included the sensitivity tests and most of the synthetic experiments, addressing the uncertainties of the system. Moreover, we now also discuss a diagonal R matrix, responding to a question by Referee #2. **Part 2** presents the results from the application of the system, as suggested. The appendices were reduced and relevant information was included into the main text of both Part 1 and Part 2 as follows:

- **Appendix A** is incorporated into the new appendices (Appendix C of Part 1 and Appendix A of Part 2) since it only contains supplementary tables and figures.
- **Appendix B** is now partially contained in Sect. 2.3 of Part 1, keeping only the purely technical part in what is now Appendix A of Part 1.
- **Appendix C** is kept as Appendix D in Part 1, since it presents auxiliary technical derivations.
- **Appendix D** is now partially contained in Sect. 2.5.3 of Part 1. Only the technical scheme of tracers in the ensemble modeling is kept in what is now Appendix B of Part 1.
- **Appendix E** is kept as Appendix E in Part 1, since it contains supplementary tables and figures.

- **Appendix F** is now contained in the results (Sect. 4) of Part 1, and has been rewritten in response to the referees' suggestions.
- **Appendices G, H, and I** consist of technical derivations and are kept (now Appendices C, B, and D of Part 2).
- 3. Referee comment: "There appears to be some contradictory statements made in the paper. This may be just my misunderstanding, but an example is the description of how the model uncertainties are found. There seems to be one explanation in Section 2.4, another one in Sect. 4.2, and another in Sect. 5.4.2. See also detailed points 10, 20, and 27. More examples are given in my detailed comments."

We thank the referee for pointing out these inconsistencies. To improve clarity, we have now merged the former Section 2 and 4 into a single, coherent Section 2 in Part 1, describing the system.

The mentioned reoccurring discussion of matrix R in Sect. 5.4.2 (now Sect. 4.5.2 of Part 1) appears because the sensitivity tests are discussed separately from the system description. We have added cross-references to ensure consistency and make a logical connection between text passages where R is mentioned.

4. Referee comment: "I'm not completely sure how the ensemble comes into the main inversion. I can see how an ensemble is used to help define the model uncertainties in Sect. 2.4, and in some of the supplementary statistics, but I am not sure how it is used in the main inversion (Eq. (1)), given the paper's title suggests that the ensemble is used for that. See also detailed points 8 and 10."

We appreciate this question and specified that the ensemble improves the main inversion through estimates of the model uncertainty. We have clarified this point when introducing the ensemble in Sect. 2.1.2 of Part 1: "The ensemble will only be used to estimate model uncertainties and error covariances (see Sect. 2.5), and to generate pseudo-observations (Sect. 3.4)." The uncertainties and correlations in the error covariance matrix R are essential for the inversion. The ensemble used to estimate R therefore plays a central role in the inversion. The construction of the R matrix from the ensemble is also indicated in the newly added flow chart (Fig. 5 of Part 1).

5. Referee comment: "I am unclear about the structure of the main control variable, s, especially as these are called scaling factors. I cannot find what these factors actually multiply in order to lead to predictions of the model's observations via Hs. See also detailed points 2, 7e, 7f, and 41a."

We thank the referee for noting that there is a lack of clarity in this important aspect. We added the formal definition of s at the beginning of Sect. 2.4 (Part 1): "We define a vector of scaling factors – in our application  $s \in \mathbb{R}^{46}$  – consisting of one prefactor for each flux category."

In the new Sect. 2.7 (Part 1) we summarize the inversion output: "The inversion results consist of one vector  $s^{\text{post}} \in \mathbb{R}^{46}$  of scaling factors and the corresponding error covariance matrix for each month."

6. Referee comment: "I am unclear how individual flux categories can be distinguished from the observations (apart from using information in the ratio of the background error statistics, which divides the posterior fluxes according to how these are prescribed). See also detailed points 29 and 40a."

This question is related to the previously mentioned scaling factors. The flux categories are the basis vectors for the field of fluxes and the scaling factors are their coefficients. In the transport simulation we compute the contribution of each flux category to the  $CH_4$  concentration of each observation. We then scale each of these contributions to optimize the agreement between model and observations. We do so by adjusting the vector  $s \in \mathbb{R}^{46}$  of scaling factors using approximately  $10^4$  observations per inversion time window. Thus, the flux categories are distinguished based on the model-observation mismatch and the expected contribution of the flux categories to the observations. We have adjusted the introduction of the method in Sect. 2 and Sect. 2.1.3 of Part 1 to make this point clearer.

This synthesis inversion method relies on the assumption that the a priori spatial distribution of fluxes within each category is realistic. This is especially important when trying to distinguish different emission sectors. Formally, the separation of sectors is very similar to separating different areas, but this does not guarantee that the results are meaningful. We therefore include a detailed discussion of the sector discrimination (Sect. 5.5.2 in the first version, now Sect. 3.4.2 of Part 2). This aspect is discussed further in the detailed point 29 below.

7. Referee comment: "It is not clear to me how the ICON model is used in the inversion itself, and whether any chemical processes are simulated as part of the forward model, H."

As now stated in Sect. 2.1.1 of Part 1: "The ICON model simulates the meteorology and the tracer transport." It is only used as a forward transport model that predicts how much methane emitted from each flux category is transported to the observation sites. This defines the matrix H. The inversion only uses the output of the transport simulation.

We agree that we missed an explanation of the involvement of chemical processes. We included a sentence into Sect. 2.1.1 of Part 1 to clarify this: "We do not simulate any chemical reactions, because the typical lifetime of CH4 in the atmosphere is much longer than the time that an air parcel typically spends in our modeling domain."

**2 Scientific points**

**1. L49-51:**

(a) Referee comment: "I'm just wondering how the ICON model is used to provide the transport winds. As well as providing tracer transport winds, is the ICON model actually simulating the winds (on-line) at the same time as advecting the methane, or is the ICON model used to pre-determine winds, which are then used by the inversion system to do the transport? The former would provide very high-temporal resolution winds (and include sub-grid-scale transport), without the need to store them, which would be ideal, but very expensive. If ICON is simulating the winds on-line, then it must be run with a data assimilation scheme to keep them realistic. I think the authors could say a little about how this is done (although see also point 12 below)."

We gladly take up this suggestion and extended Sect. 2.1 on the transport simulation (now Sect. 2.1.1 in Part 1): "The ICON model simulates the meteorology and the tracer transport. Re-initialization of the meteorological fields every 24 h ensures that the meteorology stays close to reality."

Thus, ICON simulates the winds online. The transport is computed with the full temporal resolution and includes sub-grid-scale parametrizations. To avoid using a data

assimilation scheme for the meteorology, we update the meteorological fields at 0 UTC every night using archived data from the operational numerical weather prediction at DWD.

(b) Referee comment: "Are any chemical processes (such as the reaction with OH and methane oxidation included? L398-399: 'Another potential contribution to the seasonal cycle could arise from neglecting the OH sink of CH4 in our limited domain.' suggests not. Could this be an issue over the timescale of the inversions? The authors should at least say that this is an assumption made by their system"

We added to Sect. 2.1.1 of Part 1: "We do not simulate any chemical reactions, because the typical lifetime of CH4 in the atmosphere is much longer than the time that an air parcel typically spends in our modeling domain."

Chemical processes are important in global simulations. But in our regional simulation, we have a constant inflow of fresh air from the lateral boundaries. The relevance of chemical processes is determined by the amount of methane that is removed while an air parcel is transported from the lateral boundaries to an observation site. Assuming that this transport usually takes less than 10 days and that methane in the atmosphere has a typical lifetime of 10 years, we can expect an effect of less than 5.5 pbb at the observation sites, given a background concentration of 2000 ppb. The possible bias due to the neglected chemistry is further reduced by the far-field correction.

2. L72-74: Referee comment: "The categorized fluxes are scaled to minimize the mismatch between model prediction and observed concentrations. Thus, the inversion result consists of one scaling factor for each flux category. The a priori fluxes multiplied by the scaling factors yield the a posteriori fluxes.' By '... one scaling factor for each flux category.', do the authors mean one for all positions and time, or one per flux category per position and per time (obviously with relevant correlation scales)?"

We thank the referee for this point. The scaling factors are simply one number for each flux categories. These numbers are computed independently for each inversion time window. There is no position involved.

To clarify this, we added at the beginning of Sect. 2.4 (Part 1): "We define a vector space of scaling factors – in our application  $s \in \mathbb{R}^{46}$  – consisting of one prefactor for each flux category.", and in the new Sect. 2.7 (Part 1): "The inversion results consist of one vector  $s^{\text{post}} \in \mathbb{R}^{46}$  of scaling factors and the corresponding error covariance matrix for each month."

3. L91-92: Referee comment: "Each ensemble member uses slightly different but equally likely parametrizations and meteorological initial and boundary conditions.' Does each member use slightly different driving winds too?"

The ensemble members have different winds because the initial and lateral boundary conditions for the wind differ. We describe the ensemble in the new Sect. 2.1.2 (Part 1) with the conclusion:

Since our meteorological input fields and the transport model setup are taken from operational NWP at DWD, the ensemble provides a reasonable estimate for the meteorological uncertainty in our model, including uncertainties in the simulated wind field and atmospheric stability.

When checking the ensemble construction again, we noted a mistake in the configuration. Differing from what we described in the initially submitted manuscript, the ensemble does

not use any perturbation of physical model parameters. We have corrected this in the new version. But even though every ensemble member used exactly the same transport model, the differing meteorological initial and lateral boundary conditions are sufficient to provide a reasonable ensemble spread with different winds leading to different transport.

4. Sect. 2.2 and Appendix B: Referee comment: "I'm not really sure what is meant by the far field. I assume it's a smooth correction to the methane field, determined before the inversion, and is a function of space and time. Is that right? Looking at Eqs (1) and (B1), it looks like it exists only at observation locations. See also point 35 below."

We define the far field in Sect. 2.2 (now Sect. 2.3 of Part 1): "For cases where the model predicts almost no influence from our categorized emissions (i.e., clean air cases), deviations between model and observations point to the need for correcting the CH4 advected across the lateral boundaries – here referred to as 'far field'" Thus, the far field is the contribution of CH4 from the lateral boundaries and defined in each grid cell, justifying the name "field".

The far-field correction can be defined at each location in space and time, but we only compute it at the observation locations. To simplify the notation and focus on the practical application, the formal definition of the far-field correction in Appendix B (now Appendix A of Part 1) assumes that this correction is only needed at the location of observations. But this definition can be expanded to include arbitrary points in space and time. Note that the input vectors x and y of Eqs. (B1) and (B2) are only evaluated in a projected space as P(y-x). In the Gaussian localization matrix  $\tilde{C}$ , we can include arbitrary coordinates. When applying this generalized definition of the far-field correction, one will indeed obtain a smooth field in space and time.

Thus, the far-field correction is a smooth function of space and time that is determined and added to the methane field before the inversion. But for simplicity we only compute it at the observations.

5. L100-102: Referee comment: "[...] The terms 'far-field' and 'whole domain' sound contradictory to me."

We define the far field as the contribution from outside the domain, interpreted as far distance. The wind transports CH4 from outside the domain to every location in the domain. Since the background concentration is much higher than the typical contribution of emissions within the domain, the far field contributes the domainant part to the CH4 concentration everywhere in the domain.

6. L102-103: Referee comment: "'We require this correction field to be smooth on large length and time scales, chosen in our case as ...' This sounds like a tautology to me: surely 'smooth' means 'large scale'?"

The referee is right that "smooth" implies large scales. We have adjusted the formulation accordingly (now in Sect. 2.3): "We require this correction field to be smooth on spatial and temporal scales 320 km (horizontal), 1 km (vertical), and 16 h (time)."

**7. Sect. 2.3:**

(a) Referee comment: "How is B determined as used in Eq. (1)?"

B is defined by the a priori uncertainties and correlations of the scaling factors. We have added a link to the section describing the definition of B (which was 4.4 in the submitted manuscript).

(b) Referee comment: "How are the observations of? Appendix A lists the stations, but I cannot find anywhere whether the observations are of methane or of the flux itself. What are the instrument types and instrument precisions? This would be good to know even though they are claimed to be negligible to the model error (L125). I think it should be stated as early as possible in the paper that the observations are of methane."

We use observations of methane concentrations and have added this information at multiple location of the manuscript, specifically when introducing the observations in the introduction of Part 1. We thank the referee for this suggestion.

We use observations from the European Obspack, which is a collection of observations in a standardized format curated by ICOS. These observations may use different instruments. Most of the observation sites are ICOS sites, for which a precision and repeatability of < 0.5 ppb is required under test conditions [ICOS RI (2020): ICOS Atmosphere Station Specifications V2.0. ICOS ERIC. doi:10.18160/GK28-2188]. We added this citation to the manuscript to support the claim of negligible observation error compared to the model error.

(c) Referee comment: "What is the time window and how is information propagated from one time window to the next?"

We use a time window of one month. No information from the inversion is propagated from one time window to the next. However, the transport simulation is carried out for the whole year plus spin-up. This is now explicitly stated in Sect. 2.7 of Part 1 (corresponds to old Sect. 4.3):

We simulate the transport for the whole year 2021 without any interruption. The inversion is then applied to each month separately by selecting only observations within one month. The scaling factors of the months are treated as independent, always starting with the same a priori scaling factors [...].

(d) Referee comment: "As the initial conditions of the methane field are not mentioned in the cost function, Eq. (1), it looks like the initial conditions are not adjusted as part of the inversion. Is that right? Perhaps this is the purpose of the far-field correction step? If so, please make this clear. If the initial conditions are sufficiently wrong, then this would have an impact on the quality of the inversion."

We have extended the explanation of the inversion time windows (now Sect. 2.7 of Part 1) to address these questions:

The continuous transport simulation over the whole year implies that the initial CH4 concentration is hardly relevant after the spin-up. At the beginning of each month, the modeled CH4 concentration already consists of the far field – the contribution of the lateral boundaries – and the contribution of the fluxes, which will be adjusted by the inversion.

Thus, we can discuss initial conditions from two perspectives: The initial  $CH_4$  concentration when starting the spin-up of the simulation is virtually irrelevant for the inversion. The  $CH_4$  concentration at the beginning of each inversion time window is implicitly adjusted by the inversion, because it already consists of the far field and the categorized fluxes. The adjustment of the initial concentration is implicit because we define inversion time windows in observation space and thereby avoid any explicit dependence on the  $CH_4$  concentration at the beginning of the inversion time window.

(e) Referee comment: "Given that the control variable in Eq. (1) is s (a scaling factor), this suggests that it should multiply something. What is the field that it multiplies? See also point 41a below."

The scaling factors multiply the fluxes. More precisely, each entry  $s_k \in \mathbb{R}$  of the vector  $s \in \mathbb{R}^{46}$  of scaling factors is multiplied by one flux category.

(f) Referee comment: "What is the structure of the field s? Is it represented on a grid, or does it multiply some basis functions? Field s must, presumably, be associated with the different categories of source field (a number > 40 is mentioned in D2, e.g.)."

We consider a vector  $s \in \mathbb{R}^{46}$  of 46 scaling factors. Thus, s is not a field. We parametrize the field of fluxes using 46 basis vectors (called "flux categories"). The scaling factors are the coefficients of these basis vectors.

To clarify this, we added an introduction to Sect. 2.1.3 of Part 1:

Estimating  $CH_4$  fluxes in  $> 10^5$  grid cells based on 50 observation sites seems impossible without reducing the number of degrees of freedom of the fluxes. Here, we reduce the degrees of freedom drastically by parametrizing the fluxes using only 46 basis vectors. A basis vector in this parametrization is a flux category that contains all fluxes from one region, possibly limited to specific emission sectors.

and extended Sect. 2.4 of Part 1:

We use a Bayesian inversion to optimize the agreement of model and observations by scaling the flux categories. We define a vector of scaling factors – in our application  $s \in \mathbb{R}^{46}$  – consisting of one prefactor for each flux category.

8. Eq. (1): Referee comment: "Given that the title of the paper talks about an ensemble, is Eq. (1) minimised with respect to each ensemble member? Or is the ensemble just used for the procedure described in Sect. 2.4 (see point 10 below)?"

In the inversion, the transport ensemble is only used to construct the matrix R (old Sect. 2.4, new Sect. 2.5.2 of Part 1). Eq. (1) is minimized for the so-called deterministic (i.e., non-ensemble) simulation. Besides that, we use the ensemble to generate pseudo-observations for synthetic experiments. This is now clarified when introducing the ensemble (Sect. 2.1.2 of Part 1).

**9. L111-112:**

(a) Referee comment: "How can fluxes be transported? Do the authors mean 'transported methane emitted by the fluxes'?"

The correction "transported methane emitted by the fluxes" is indeed what we meant. We now use the formulation (Sect. 2.4 of Part 1): "In the first term, the vector y of observed concentrations is compared to the model prediction, which consists of the transported fluxes Hscontribution Hs of fluxes within the model domain and the modeled far field xff including the far-field correction."

(b) Referee comment: "In Eq. (1) and in the above quote,  $x^{ff}$  is the 'far field'. Is this the same as field c in Appendix B? If so, I would recommend that the same symbol is used in all parts of the paper. If not, I would recommend that they are not referred to as the 'far field'."

 $x^{\text{ff}}$  refers to the modeled far field including the far-field correction. This is now stated explicitly after Eq. (1), as quoted in the previous point.

We distinguish the far field which could be called  $x_{\text{uncorrected}}^{\text{ff}}$  (not used in the manuscript) and its correction c. For the inversion, we only need the corrected far field,  $x^{\text{ff}} = x_{\text{uncorrected}}^{\text{ff}} + c$ .

- 10. Sect. 2.4: Referee comment: "The R' matrix in Eq. (2) is determined from an ensemble."
  - (a) Referee comment: "Is this the same ensemble that is (possibly) used in the main inversion (but see point 8 above)? How is it initialised?"

The ensemble is only used for the main inversion by generating R' (aside from the construction of pseudo-observations). See also the response to the referee's point No. 4 in Sect. 1 of this response. The initialization of the ensemble is now described in Sect. 2.1.2 of Part 1.

(b) L132: Referee comment: "'...added to each observation accounting for any representativity error.' I think representativity error is something different, unless sub-grid-scale processes are included in the divergence of the ensemble members."

By representativity error we mean the error in predicting the observation for a particular observation site when assuming that the model works perfectly on the grid scale. This can be because of local topography or other local effects. We currently do not have a method to systematically estimate the representativity error. Instead, we include a sufficiently large uncorrelated error for each observation data point when constructing R'. This is a very simplified view and discards that the representativity error may be correlated over long times.

To clarify this point, we adjusted the formulation (now in Sect. 2.5.2): "With this uncorrelated uncertainty  $\sigma_{\rm const}$ , we account for additional uncertainties, such as representativity errors inherent to a simulation at finite resolution."

11. L163-168: Referee comment: "'Plumes caused by high emissions in a small area [...]' I very much like this interpretation."

We thank the referee for the positive feedback.

12. L177-178: Referee comment: "'The meteorological initial and lateral boundary conditions used to drive our transport model are taken from the archive of DWD's operational numerical weather prediction (NWP), which also employs the ICON model.' This statement partially addresses point 1 above, but I am still unsure whether it is ICON itself – or another model that just uses ICON-derived winds – that is used for the H operator in Eq. (1)."

We hope this question became sufficiently clear in point 1(a) above. We use ICON itself to compute H.

13. L184-185: Referee comment: "In contrast to the meteorological fields, the CH4 concentrations are only transported and never re-initialized.' Is it just transportation, or are chemical reactions included too?"

As stated for point 1(b), we do not include any chemical reactions in the simulation.

14. L188: Referee comment: "'We ensured mass conservation when interpolating to our model grid.' How is this done? Is it by multiplying the interpolated fluxes by a factor to ensure that the total flux is the same after interpolation?"

The interpolation algorithm ensures that mass is conserved. To determine the flux in one target grid cell, we draw the target grid cell on the input flux data. The input flux field within the target grid cell is then averaged to obtain the flux in the grid cell. This may smoothen the input, but it will not change the mass on scales larger than the grid scale as long as the output grid covers the input grid. We regard this algorithm as a technical implementation detail that does not need to be described in the publication.

- 15. L189-190: Referee comment: "'Anthropogenic fluxes excluding LULUCF are split further into 12 GNFR sectors (gridded aggregated NFR, nomenclature for reporting, Veldeman et al. (2013)), ...'
  - (a) GNFR is not defined (I assume GNFR = Gridded NFR, but I don't know what NFR stands for).
  - (b) I don't really understand the text in the brackets.

We agree that the abbreviation GNFR was not introduced properly. In the new structure, the term "GNFR" is introduced in Sect. 2.1.3 of Part 1: "When distinguishing emission sectors, we stay close to the national reporting by using definitions from the gridded aggregated nomenclature for reporting (GNFR, Veldeman et al., 2013)."

Lines 189-190 of the old version are now part of Sect. 3.3 of Part 1, where we make clear that we only use the convention named GNFR:

Since the input datasets for anthropogenic emissions are based on reporting to the UNFCCC, these distinguish between GNFR sectors following the reporting conventions (Veldeman et al., 2013). For the inversion, we combine these sectors and only distinguish between agriculture and the sum of all other sectors as described in Sect. 2.1.3.

16. L197: Referee comment: "'These emissions are missing in our a priori estimate.' Does this represent a low bias in the a-priori?"

Yes, the missing natural fluxes in Germany represent a low bias in the a priori. We now include this clarification: "These emissions are missing in our a priori estimate, leading to a low bias in the a priori." (Sect. 3.2 in Part 1)

17. L219-221: Referee comment: "'Observations within the planetary boundary layer are most representative in the afternoon hours whereas measurements at high mountains have less local influence at night time (Bergamaschi et al., 2015). We therefore use the time windows 23 h to 5 h (local mean time) for stations on high mountains and 11 h to 17 h for all other stations.' If observations on high mountains have less influence over night, why only use them during 23 h to 5 h?"

We aim to use observations that are representative on scales larger than the grid scale. When observations on high mountains are influence by local emissions and convection during day time, it is more likely that our model cannot correctly predict the concentrations. When selecting observations, our aim is to used those observations that our model could predict correctly if the modeled fluxes were correct.

To clarify that this is a standard procedure in atmospheric inversions, we have extended the explanation (now Sect. 3.3 of Part 1):

Observations within the planetary boundary layer are most representative in the afternoon hours whereas measurements at high mountains have less local influence are less influenced by very local fluxes at night time. Inversions therefore commonly use afternoon observations for flat land stations and night times at mountain sites (Bergamaschi et al., 2015; Steiner et al., 2024b). We therefore use the time windows 23 h to 5 h (local mean time) for stations on high mountains and 11 h to 17 h for all other stations.

- 18. L227-229: "In the last filtering step step 5 in Table 2 we exclude data points with extreme mismatch between model and observations of more than 200 ppb. Data points where the observations are more than 20 ppb below the model-predicted far field are also discarded."
  - (a) Referee comment: "I assume by 'model', the authors are referring to the a-priori?" Yes, we now explicitly state that we use the far-field corrected a priori.
  - (b) Referee comment: "For the first (extreme) condition is this written mathematically as  $|y-Hs-x^{ff}| > 200 \text{ ppb}$  or |y-Hs| > 200 ppb to discard? Perhaps this is better explained as an explicit inequality?"
    - We follow the suggestion to use the mathematical formulation, which is  $|y Hs x^{\text{ff}}| > 200 \text{ ppb}$  (now included in Sect. 3.3 of Part 1)
  - (c) Referee comment: "For the second condition is this, mathematically  $y x^{ff} < -20 \,\mathrm{ppb}$  to discard? Again, perhaps this is better explained as an explicit inequality?"

    Also here we follow the suggestion and use the mathematical formulation,  $y x^{ff} < -20 \,\mathrm{ppb}$ .

**19. Section 4.1.2, point (ii):**

- (a) Referee comment: "By advecting a tracer, how is it possible to set a lifetime for that tracer? One would know only the tracer concentration in any grid box, not how long it has been since that tracer was released (and anyway is likely to be comprised of tracers that have a variety of emission times)."
  - ICON-ART supports passive tracers that decay exponentially, similar to radioactive decay. We use this feature to avoid the accumulation of categorized tracers over long times. Since the exponential decay starts immediately, a small part of the emitted  $\mathrm{CH}_4$  will always be attributed to the background before reaching an observation site. This usually small fraction of the  $\mathrm{CH}_4$  concentration will not be scaled.
- (b) Referee comment: "The statement that tracers have a lifetime and that 'no CH4 is lost' seem contradictory to me."

We agree that this seems contradictory. We compute the total contribution of all fluxes within our domain separately from the categorized fluxes. Thus, the a priori concentration is modeled without loosing any CH4. In the a posteriori concentration, the lifetime of the categorized tracers leads to a small change in the total CH4 concentration. We have clarified the statement accordingly.

As recommended and following point 3 in Sect. 3 of this response, we have rewritten Sect. 4.1.2 (now Sect. 2.1.4 in Part 1) and emphasize that the prior concentration is computed independent of the artificial lifetime. We furthermore include the following explanation:

After emission, the concentration in these tracer fields decays exponentially with a mean lifetime of five days. [...] The artificial decay rate affects the posterior concentration and the sensitivity of the inversion to changes in the emissions. However, assuming that the typical time between emission and observation is short compared to the artificial lifetime and in the presence of transport model errors, we expect that this feature of our inversion system leads to more robust results.

20. Section 4.2: Referee comment: "I'm very confused as there seems to be two prescriptions for how the R-matrix is determined – this section and Sect. 2.4. See also point 27 below."

We have merged sections 2 and 4 to avoid this confusion. The description of determining R is now contained in Sections 2.5 and 2.6 of Part 1. See also our reply to point 3 of Sect. 1 of this response.

21. Section 4.3: Referee comment: "Is any cycling done between the monthly inversions? That is, does the posterior of one month become the prior of the next?"

We do not use any cycling between the monthly inversions. The months are treated independently as stated now in Sect. 2.7 of Part 1:

The inversion is then applied to each month separately by selecting only observations within one month. The scaling factors of the months are treated as independent, always each month starting with the same a priori scaling factors ( $s_k^{\text{prior}} = 1$  for all k) and the same a priori scaling uncertainties (B matrix).

**22. Section 4.4:**

(a) Referee comment: "L316: 'In each inversion time window, we consider uncorrelated a priori scaling factors ...' and L319-320: '..., and within Germany categories describing the same sector have an a priori uncertainty correlation of 50%'. These two statements seem contradictory."

We agree that our statement about the correlations in B, meant as an exception of the otherwise diagonal B, can be misleading. We have adjusted the formulation to avoid contradictory statements, for L316: "In each inversion time window, we consider uncorrelated a priori scaling factors with  $[\ldots]$ " and for L319-320: " $[\ldots]$ , and within Germany categories describing the same sector have an a priori uncertainty correlation of 0.5 (e.g.,  $[\ldots]$ ). All other categories are treated as uncorrelated in the a priori."

(b) L316-320: Referee comment: "In each inversion time window, we consider uncorrelated a priori scaling factors with a two standard deviation (2σ) uncertainty of 80 % for most flux categories, corresponding to a 95 % confidence interval of ±80 %. Throughout this paper, uncertainties will denote two standard deviations or 95 % confidence intervals. Categories resolving emission sectors have a higher prior 2σ uncertainty of ±100 %, and within Germany categories describing the same sector have an a priori uncertainty correlation of 50 % (e.g., ...)' There are (presumably) two quantities represented by percentages here, the confidence intervals (which represent percentage of the PDF volume), and the methane quantity itself (presumably represented as a percentage of some value). If this is correct could there be a better way to describe these to avoid confusion?"

The interpretation of the referee is correct. We have adjusted the notation and now use percentage values only to describe the confidence interval to avoid confusion.

23. L333-345: Referee comment: "'Additionally, we combine the two variants of inversion (prior-R and posterior-R, see Sect. 2.4.2) by taking the arithmetic mean of the two separate inversion results, arriving at the combined scaling factors.' The posterior error statistics should just be a function of the inverse Hessian of the inversion. Why the need to refer to the R-matrices? A similar mention of the R-matrices is made in L4-5 of the caption of Fig. 5."

We use the terms "prior-R" and "posterior-R" as labels to distinguish two variants of the inversion that differ by the construction of R. Both variants produce slightly different results. In the new structure, we explicitly introduce the terms "prior R" and "posterior R" in Sect. 2.5 of Part 1 and use these terms more consequently throughout Part 1 to avoid similar confusion.

24. L348-349: Referee comment: "'Figure 4 presents an overview of (a) the a priori CH4 fluxes, (c) the resulting scaling factors, and the respective uncertainties (b, d), all accumulated over the year 2021.' Presumably 'averaged' rather than 'accumulated'."

Yes, results are averaged over the year. We have adjusted the formulation accordingly: "Figure 2 presents an overview of (a) the a priori CH4 fluxes accumulated over the year 2021, (c) the resulting scaling factors averaged over 2021, and the respective uncertainties (b, d), all accumulated over the year 2021." (Sect. 3.1 of Part 2, the numbering of the figures has changed)

25. L416-417: Referee comment: "'Other filtering parameters such as the number of sampling heights per station (case 202) and ...' I don't understand what is meant by, 'the number of sampling heights per station', given that the height of an observation station is (presumably) fixed."

Some observation sites are tall towers with up to five inlet heights. We use up to three of these sampling heights per observation site. Using many observations can be beneficial, but when using co-located observations, error correlations must be taken into account. The maximum number of sampling heights used per observation site is therefore a tuning parameter of our inversion system.

In response to the referee's question, we have rewritten the explanation of this aspect in Sect. 3.3 (now of Part 1): "For tower observations, we use up to three sampling heights per station, preferring the highest three sampling heights and discarding observations below 50 m above ground level to reduce the influence of very local emissions."

26. L418-420: Referee comment: "Neglecting extreme outliers has only a small effect (cases 206, 207), but limiting the influence of outliers by increasing their uncertainty has a considerable impact (cases 208, 209).' One would expect that neglecting extreme outliers is just the limiting case of increasing their uncertainty to infinity. It might be counterintuitive then that increasing their uncertainties has a large impact, then a small impact when their uncertainties are increased further to infinity."

We define extreme outliers by  $|y - Hs - x^{ff}| > 200 \,\mathrm{ppb}$  or  $y - x^{ff} < -20 \,\mathrm{ppb}$ . This definition differs from the definition of outliers, which are all observations which deviate from the a priori model prediction by > 3 standard deviations. Since the latter definition affects more data points, it has a stronger impact on the results. To avoid confusion, we have added these definitions in the discussion (now in Sect. 4.5.1 of Part 1):

"Limiting the influence of outliers with model—observation mismatch  $|\mu_i| > 3\sqrt{R'_{ii}}$  by increasing their uncertainty (see Sect. 2.6.2) has a considerable impact (cases 208, 209). Completely neglecting extreme outliers – defined by  $|y - Hs - x^{\rm ff}| > 200\,{\rm ppb}$  or  $y - x^{\rm ff} < -20\,{\rm ppb}$  – has only a small effect (cases 206, 207)."

To indicate how many observations are affected by neglecting extreme outliers, we have added the total number of observations used by the inversion in the sensitivity tests in Table E1.

27. L428-429:Referee comment: "'... and the uncorrelated additive uncertainty σconst of each data point (cases 309, 310).' Perhaps I have missed it, but I cannot find any reference to σconst in Sects. 2.4 or 4.2. This adds to my confusion about how the R-matrix is determined. See also point 20 above."

We now refer to Eq. (2) to clarify where  $\sigma_{\text{const}}$  was introduced. Following point 10(b) above, the text below Eq. (2) now includes the explanation: "With this uncorrelated uncertainty  $\sigma_{\text{const}}$ , we account for any other source of uncertainties, including the representativity error."

28. L471: Referee comment: "'These Each vector of scaling factors defines the a synthetic truth, ...' (These suggested changes distinguishes considering all scaling factors together in one ensemble-based inversion.)"

We agree with the referee and use the suggested formulation (now in Sect. 3.4.1 of Part 2).

29. L484-485: Referee comment: "Within Germany, we distinguish agriculture from other emissions. The ability to distinguish sectors can be described by averaging kernel matrices which estimate ....' Putting the B-matrix aside, how can different sectors be distinguished from observations of methane? See also point 40a below."

To distinguish sectors, we use their different a priori spatial distribution of emissions. This involves large scales (e.g., strong agriculture emissions in the north west of Germany) and smaller scales (e.g., different dominant sectors in urban and rural areas). For example, agriculture emissions in Western and North-Western Germany lead to strong signals at the observation sites Steinkimmen and Torfhaus, whereas other emissions from the same region lead to stronger signals at Jülich, Heidelberg and KIT. The pattern of the model–observation mismatch in space and time indicates where emissions are overestimated or underestimated. By adjusting the scaling factors (prefactors) for the corresponding flux categories, we optimize the agreement with observations and distinguish sectors.

We added the following explanation in Sect. 3.4.2 of Part 2:

The discrimination of emission sectors works in the same way as we distinguish emissions from different areas. Each sector has a specific spatial distribution of emissions, which we assume to be correct in the a priori. The predicted CH4 concentration at the observation sites will therefore depend on how the individual sectors are scaled. In the inversion, the sector emissions are scaled to find optimal agreement of model prediction and observations.

One remaining challenge is that the described method has large uncertainties. This motivates the detailed discussion of the ability to distinguish sectors in Part 2, Sect. 3.4.2 (Sect. 5.5.2 in the initial submission), which allows us to conclude that we can obtain some information on the sector emissions.

30. Section 5.6: Referee comment: "This section is about simulated transport errors, but I cannot find a description of how transport errors are simulated. Is noise added to the winds e.g.?"

To clarify this issue and because these synthetic experiments are now a central part of Part 1, we introduce a new Sect. 3.4 on "Synthetic observation experiments" in Part 1:

To test our setup and analyze biases, we use synthetic experiments in which observation data are replaced by model-generated pseudo-observations. These synthetic experiments use exactly the same setup and the same observation coordinates. Only the observation values are replaced by the simulation result of one of our 12 ensemble members. We thus obtain 12 separate datasets of pseudo-observations, in which a transport error is simulated by the transport ensemble members.

As mentioned in the reply to point 3 above, the ensemble members have different winds as a central component for different transport.

31. Figure 10: Referee comment: "I'd recommend adding the 1:1 line as a guide. See also point 39c below."

We agree that adding the 1:1 line improves the figure. To also address the point by Referee #2 on the comparison of our R matrix construction to a "standard approach", we included additional results into Figure 10 (now Fig. 6 of Part 1). This led to a new visualization, taking all referee comments into account:

Figure 6. Mean (a, c) and standard deviation (b, d) of monthly flux estimates relative to the prior in synthetic experiments for diagonal R (blue), prior R (orange), and posterior R inversion (green). Each bar represents the posterior fluxes for 144 inversions, obtained from 12 datasets of pseudo-observations, each covering 12 monthly time windows. Black horizontal lines indicate the  $2\sigma$  statistical uncertainty estimate. Panels (a, c) show the bias as the relative deviation of the mean posterior from the prior, which is equal to the synthetic truth. The standard deviation (b, d) among the 144 emission estimates indicates the random error expected in each monthly inversion. Colored lines in (b, d) show the mean posterior  $1\sigma$  uncertainty, which is similar for all three methods.

In the figure, we also removed the German sector emissions because these are discussed in Part 2.

32. L515-516: Referee comment: "'Localized sources that cause a strong plume are underestimated by both methods, though the bias is reduced in the posterior-R inversion as predicted . . . ' I assume it is the posterior fluxes that are plotted in Fig. 10, using the two R-inversion methods, with and without the far-field correction (and not their errors). Figure 10 (and Fig. F2, see point 39c below) therefore compares the posterior values for each relative to the prior. How is it possible therefore to tell whether a strong plume is underestimated and what the biases are?"

We thank the referee for this remark on Figure 10. The figure shows posterior emissions relative to the true emissions in a synthetic observation experiment. In the synthetic experiment, we define the true emissions and compute pseudo-observations that agree with this defined truth. The inversion only sees the pseudo-observations and tries to reconstruct the true emissions. From the result we can then see whether emissions are underestimated compared to the synthetic truth.

To prevent misinterpretation, we improved the visualization of the data and expanded the explanation of the synthetic experiments in the new Sect. 3.4 of Part 1 (see also point 30 above).

33. L537: Referee comment: "'Firstly, we find that our top-down CH4 emission estimates are significantly higher than reported for Germany.' Although, looking at Fig. 5, the NIR estimate is well within the 95% confidence interval of the posterior."

We appreciate this comment. While the uncertainty ranges of prior and posterior have some overlap, the reported value for Germany (1.89 Mt, light blue/cyan in Fig. 5) is not within the uncertainty range of the posterior (2.15 Mt to 2.88 Mt, red, 95 % confidence interval).

34. L619-620: Referee comment: "The increasing availability of satellite data is especially interesting for constraining concentrations and emissions in less observed regions, such as near the boundaries of our domain.' This seems an obvious extension of the work given the wide range of total column methane retrievals available from satellites. This may require careful tuning though to remove potential biases."

We fully agree with the referee's comment. To emphasize this point and for clarification, we have adjusted the text: "The increasing availability of satellite data is especially interesting for constraining concentrations and emissions in less observed regions regions with few or no ground-based observations, such as near the boundaries of our domain, which is an aspect to be addressed in future studies."

35. Appendix B, Eq. (B2): Referee comment: "I am concerned that the formula (B2) is not formally correct in one detail (in the final step). [...]"

We thank the referee for spotting this. The arguments and calculations presented by the referee are correct. We are aware that Eq. (B2) is one possible solution of Eq. (B1), but this solution is not unique. We have adjusted the formulation in Appendix B to highlight this aspect.

Referee comment: "I think this requires some attention in the paper."

We have extended the appendix and included a short proof that the chosen (non-unique) solution is optimal in the sense that it minimizes  $c^{\top}\tilde{C}^{-1}c$  under the constraint that is solves Eq. (B1).

We want to show that the following solution is optimal:

$$c = \tilde{C}P^{\top} \left[ P(\tilde{C} + \tilde{R})P^{\top} \right]^{-1} P(y - x). \tag{A2}$$

As derived by the referee, we know that

$$Pc = P\tilde{C}P^{\top} \left[ P(\tilde{C} + \tilde{R})P^{\top} \right]^{-1} P(y - x). \tag{A6}$$

The extended appendix (now Appendix A of Part 1) continues:

One can furthermore show that Eq. (A2) is optimal in the sense that it minimizes  $c^{\top}\tilde{C}^{-1}c$  under constraint that c is a solution of Eq. (A1) [or Eq. (A6)]. Thus, this solution is as close as possible to zero under the constraint of smoothness (quantified by  $\tilde{C}$ ). By defining  $\xi = \left[P(\tilde{C} + \tilde{R})P^{\top}\right]^{-1}P(y-x)$  and introducing Lagrange multipliers  $\lambda$ , we obtain

$$f(c,\lambda) = c^{\mathsf{T}} \tilde{C}^{-1} c + \lambda^{\mathsf{T}} (Pc - P\tilde{C}P^{\mathsf{T}}\xi), \quad \frac{\partial f}{\partial c_i} = 0, \frac{\partial f}{\partial \lambda_i} = 0,$$
 (A7)

$$c = -\tilde{C}P^{\top}\lambda \quad \text{from } \partial_{c_i} f(c, \lambda) = 0,$$
 (A8)

$$Pc = P\tilde{C}P^{\mathsf{T}}\xi \quad \text{from } \partial_{\lambda_j} f(c,\lambda) = 0.$$
 (A9)

Since  $P\tilde{C}P^{\top}$  has full rank, combining Eqs. (A8) and (A9) implies that  $\lambda = -\xi$  and thereby  $c = \tilde{C}P^{\top}\xi$  is the unique solution of the optimization problem  $\arg\min_c f(c,0)$  under the constraint that  $Pc = P\tilde{C}P^{\top}\xi$ .

Given that we aim for a smooth solution that should only deviated from zero when force by observations, we consider Eq. (A2) (old Eq. (B2)) the optimal solution and need not worry about non-uniqueness of Eq. (A6) (or old Eq. (B1)). We thank the referee for checking our calculations in detail and for motivating this more formal justification for the construction of our far-field correction.

36. Appendix C: Referee comment: "I didn't follow the reasoning in this appendix. For example, I note the following points."

We agree that the derivations in this appendix (now Appendix D of Part 1) need more clarification. We have rewritten the mathematical derivation of the central result, Eq. (C4), of the appendix.

Remark: In the new derivation, the matrix Q may seem to differ from the old appendix at first sight, but one can show that both forms are equivalent.

- (a) Referee comment: "The left hand side of (C1) seems strange to me. It is the probability of a real number (the mismatch) being realised. This doesn't make any sense to me. Normally a probability density would be used, e.g.  $P(\mu = y Hs^{prior} x^{ff})d\mu$  to mean the probability between values of  $\mu$  and  $\mu + d\mu$ ."
  - (C1) describes a probability density and not a probability, as correctly noted by the referee. We have corrected this aspect.
- (b) Referee comment: "What is the variable dPs and how does one arrive at (C1)?"

  As recommended, we have reformulated this step and avoid the ambiguous notation used in (C1):

We start from the probability density of observations y under the assumption that s describes the true emissions:

$$P(y|s) \propto \exp\left[-\frac{1}{2}(y - Hs - x^{\text{ff}})^{\top}R^{-1}(y - Hs - x^{\text{ff}})\right].$$
 (D1)

Like in the inversion, R describes uncertainties in the transport, in the corrected far-field contribution  $x^{\mathrm{ff}}$ , and in the observations y. By a change of variables we obtain the probability for the a priori model—observation mismatch  $\mu^{\mathrm{prior}} = y - Hs^{\mathrm{prior}} - x^{\mathrm{ff}}$ :  $P(\mu^{\mathrm{prior}}|s)d\mu = P(y|s)|_{y=Hs^{\mathrm{prior}}+x^{\mathrm{ff}}+\mu^{\mathrm{prior}}}dy$ .

To estimate whether a given  $\mu^{\text{prior}}$  is realistic, we need to integrate out the scaling factors s to obtain  $P(\mu^{\text{prior}})$ . We denote the integral over the vector space of scaling factors s with probability measure  $dP_s$  by  $\int_s \bullet dP_s = \int_s P(s) \bullet d^n s$  for  $s \in \mathbb{R}^n$ . Using the above definitions in Eq. (D1), we obtain [Berchet et al., 2015, doi:10.5194/gmd-8-1525-2015]

$$P(\mu^{\text{prior}})$$

$$= \int_{s} P(\mu^{\text{prior}}|s) dP_{s}$$

$$\propto \int_{s} \exp\left[-\frac{1}{2}(y - Hs - x^{\text{ff}})^{\top} R^{-1}(y - Hs - x^{\text{ff}}) - \frac{1}{2}(s - s^{\text{prior}})^{\top} B^{-1}(s - s^{\text{prior}})\right]_{y = Hs^{\text{prior}} + x^{\text{ff}} + \mu^{\text{prior}}} d^{n}s$$

$$(D3)$$

$$\tau = s - s^{\text{prior}} \int \exp\left[-\frac{1}{2}(\mu^{\text{prior}} - H\tau)^{\top} R^{-1}(\mu^{\text{prior}} - H\tau) - \frac{1}{2}\tau^{\top} B^{-1}\tau\right] d^{n}\tau$$

$$(D4)$$

(c) Referee comment: "The line in (C2) appears to integrate over the posterior. How does this relate to (C1)?"

We have changed the notation and added another step (Eq. (D3) above) in which we explicitly write the assumed Gaussian probability densities for y and s to make this step clearer. The integral over s originates from basic Bayesian probability theory (Eq. (D2) above).

- (d) Referee comment: "The term  $H\tau + \mu$ , which appears in (C2) has the following form [...]. This doesn't seem correct to me."
  - (C2) contained a sign mistake. This mistake did not change the result. We are thankful for spotting this mistake.
- (e) Referee comment: "The rest of the section does contain some useful analysis about the expected value and range of  $\chi^2/N_{dof}$ , but the appendix should be explained more and a reference added."

We have extended the explanation in the first part of the appendix and added some references (Greenwood and Nikulin (1996) for a general discussion of  $\chi^2$  tests, Berchet et al. (2015) for  $P(\mu^{\text{prior}})$  and Abramowitz and Stegun (1964) for the approximation of  $\chi^2$ ).

- 37. Appendix D: We have included most of this appendix in the main text (Sect. 2.4.3 of Part 1). This part of the appendix was completely rewritten. Only Appendix D2 remains in what is now Appendix B of Part 1.
  - (a) Referee comment: "'We estimate the model uncertainty using a meteorological ensemble.' Which model uncertainty and which quantity? I assume the authors are referring to the uncertainty in s?"

The model uncertainty is the transport uncertainty described by R. To clarify this point, we now write "model uncertainty in R" when referring to this uncertainty, unless it is clear from the context. The mentioned sentence in line 692 has been removed when rewriting the appendix.

- (b) Referee comment: "'Stronger emissions lead to stronger spatial gradients in the model concentrations...' (suggested change to make distinct from gradients in state space)."

  We follow this suggestion and refer to spatial gradients when discussing this issue (now in Sect. 4.2 of Part 1).
- (c) L712-715: Referee comment: "When using a priori scaling factors to estimate the model uncertainty, we need only the total concentration  $x_i^m(s^{prior})$  for each ensemble member. Thus, only a single tracer field is required in the ensemble transport simulation. To fully compute  $x_i^m(s)$  as function of s, the tracer categories need to be distinguished for each ensemble member, resulting in > 40 tracer fields in the ensemble simulation.' The only difference between the calculation of  $x_i^m(s^{prior})$  and  $x_i^m(s)$  is that  $s^{prior}$  is replaced by s. How does this require that > 40 different tracer fields are required? I guess that > 40 different values of s are chosen. What is the significance of the number 40? The difficulty to understand this may be connected to point 7f above."

To compute  $x_i^m(s)$  for one fixed vector  $s \in \mathbb{R}^{46}$  (e.g.,  $s = s^{\text{prior}}$ ), we can use a single tracer that contains all CH4 emissions scaled by s. But we are interested in the function  $x_i^m(s)$  and want to evaluate this for arbitrary s without re-running the transport model. This is only possible if we know  $x_i^m(s)$  for a complete set of basis vectors of  $\mathbb{R}^{46}$  and additionally for the background concentration  $x_i^m(0)$ . Thus, we need to simulate the transport for 47 tracers, which is numerically expensive.

We have adjusted the explanation to clarify this point (now Appendix B of Part 1):

When using a priori scaling factors to estimate the model uncertainty in R, we need only the total concentration  $x_i^m(s^{\text{prior}})$  for each ensemble member m and each observation i, where  $s^{\text{prior}}$  is known. Thus, only a single tracer field is required in the ensemble transport simulation. To compute  $x_i^m(s)$  for arbitrary  $s \in \mathbb{R}^{46}$ , the flux categories need to be distinguished for each ensemble member, resulting in > 40 tracer fields in the ensemble simulation.

- (d) Referee comment: "In the above, what does i stand for?"

  i labels the observations as introduced in Sect. 2.4. We added a reminder on this notation (see previous point 37c for the new text) and adjusted the notation throughout the manuscript to avoid using i differently.
- (e) Referee comment: "I don't really understand Eq. (D3). What is the distinction between  $x_i^{mg} = x_i^m(P_g s^{prior})$  and  $HP_g s^{prior}$  (they seem to be the same thing)? Making this distinction is probably the key to understanding (D3)."

We thank the referee fo this valuable comment because it points to a mistake in the notation of Appendix D2: It should be  $x_i^{mg} = H^m P_g s^{\text{prior}}$  and not  $x_i^{mg} = x_i^m (P_g s^{\text{prior}})$ . By  $x_i^{mg}$  we denote the contribution of emissions from flux-category-group g to the concentration at observation i in ensemble member m. The new formulation of the appendix should clarify this point:

From the deterministic model run, we know the operator H mapping scaling factors s to a model prediction  $Hs+x^{\mathrm{ff}}$  for the concentrations. For ensemble member m, we would ideally know  $H^m$  and  $x^{\mathrm{ff},m}$  yieldingto compute a model prediction  $H^ms+x^{\mathrm{ff},m}$ . In lack of computational resources to compute  $H^m$  for every ensemble member, we combine information from the deterministic run (H) and selected tracers for the ensemble run to approximate  $H^m$ . To avoid calculating the full matrix  $H^m$ , We group the flux categories into groups  $\{g\}$  and denote by  $P_g$  the projector of scaling vectors s on the subspace spanned by the flux categories in group g. In the ensemble members, we compute the total concentration from group g,  $x_i^{mg} = H^m P_g s^{\mathrm{prior}}$ . We distribute the 46 flux categories to only three groups and thereby reduce the computational effort considerably. To estimate the full dependence on the scaling factors in the ensemble, we approximate:

$$x_i^m(s) \approx \sum_g \frac{(HP_g s)_i}{(HP_g s^{\text{prior}})_i} x_i^{mg} + x_i^{\text{ff},m}.$$
 (B1)

 $[\dots]$

**38. Table E1:**

(a) Referee comment: "ID 101: 'no time averaging', given the explanation column, shouldn't this read 'change in time averaging'?"

We agree that this is misleading and adjusted the description to "no additional time averaging". The observations include some type of time averaging over 1 h. By choosing the same window for the time averaging in the model, we avoid any time averaging beyond what is required to compare to the available observation data.

(b) Referee comment: "Some of the test cases (rows in table) are ambiguous to me. For example, '200, fewer hours of day, use time window 12h-16h (0h-4h for high mountains)'. Does it mean that the reference case used observations at all hours? Then again

'201, all hours of day, no filtering by time of day, increase uncertainty inflation by factor 1.5' seems to imply that using observation at all hours is a test case different from the reference. I would recommend going through the entire list and making sure that each test case is unambiguous to the reader."

We have checked the list and added more information. For many cases, like the mentioned case 200, we have added the configuration of the reference case for comparison. Furthermore, when changing how observations are selected, we add the total number of considered observations as an indicator for the impact of the changes in tuning parameters. We also added more references to those parts in the manuscript where the considered tuning parameters are introduced.

**39. Appendix F:**

(a) Referee comment: "I am confused why the truth (horizontal lines) should change with the test ID."

The test IDs represent different scenarios. These scenarios include changes in the observation bias, noise, and changes in the true fluxes. Since these are synthetic experiments with pseudo-observations, we can define the true emissions and construct pseudo-observations accordingly. By keeping the prior fixed and changing the truth, we test how the inversion system is expected to reacts to changes in the emissions.

(b) Referee comment: "Next, we test the effect of an underestimation or overestimation of all emissions. In case 20 of Fig. F1, all natural and LULUCF fluxes are reduced by 40% in the truth, and cases 21 and 22 change all anthropogenic emissions excluding LULUCF by -20% and +20%, respectively.' Are these experiments simply changing the true emissions (and the synthetic observations and a-priori values) and then repeating the inversion? I would've thought that such an experiment would be expected to have posterior fluxes that are consistent with the truth. Wouldn't a more interesting experiment be one with the underestimation or overestimation of all emissions in the a-priori, but with the truth (and synthetic observations) unchanged?"

The synthetic experiments change the true emissions and thereby the pseudo-observations, but the a priori emissions are not changed. The inversion is repeated with the different pseudo-observations. Since the a priori emissions are not adjusted to the new synthetic truth, we can analyze how well the inversion can determine the modified synthetic truth. One could also keep the truth fixed and change the a priori. But in our setup it is simpler to change the truth and leave the a priori unchanged. In both cases, the a priori is an underestimation or overestimation of the synthetic truth.

In response to the referees' comments, the old Appendix F has been rewritten and is now included in the main text of Part 1. We introduce the mentioned test cases as follows (Sect. 4.4, Part 1): "For the last three test cases (20–22), we scale either the natural and LULUCF fluxes or all other emissions in the synthetic truth while leaving the a priori emissions unchanged."

(c) Fig. F2 and text: Referee comment: "It would be useful to draw the 1:1 line in each panel (see also point 31 above). It would then be seen that the when far-field correction is applied the posterior R-inversion does better than the prior R-inversion. I think it is best to describe fully the experiments being done and then show the results in the Figs. In the case of the text around Fig. F2 it is not until the end of the appendix that the reader learns that the 'prior is underestimated compared to the synthetic truth.'"

This part of the old Appendix F is now Sect. 4.3 and Fig. 7 of Part 1. Like Fig. 10 (point 31 above), the representation of the data has changed such that no 1:1 line is needed to compare the different methods. In response to the referee's comment, we now state that: "The a priori emissions remain unchanged and are thus lower than the synthetic truth." before referring to the figure with the results.

**40. Appendix G (now Appendix C of Part 2):**

(a) Referee comment: "'When observations can detect a change in total emissions but cannot distinguish between different emission sectors, the sector-resolving inversion will change the sectoral distribution based on the prior uncertainties.' Out of curiosity, how could observations of methane make this distinction? Also mention is made in the main text at L365."

As explained in the above point 29, the different spatial distribution of emissions from different sectors leads to different patterns in the observations. The analysis in Appendix G shows one possible problem when trying to estimate sector emissions.

- (b) Referee comment: "'The a priori probability density for an emission ...'" We thank the referee for this correction, which has been incorporated.
- (c) Referee comment: "The symbol P(e) is used to represent the posteriori probability density, but the same symbol is used for the prior density in Eq. (G1). Although this is often done when describing probabilities, one can distinguish the prior from the posterior by its argument. The prior is often written as P(e) and posterior as the conditional density P(e|y). The statement on L770-771 needs to refer to the posterior."

Eq. (G1) refers to the a priori probability density P(e) but remains valid for P(e|y). We have adjusted the notation accordingly:

The inversion will yield a result for estimate the total emissions  $e_{\text{tot}}^{\text{post}}$  that maximizes the probability P(s) when including information from the observations such that the a posteriori probability density P(e|y) is maximized. But by assumption, these observations do not distinguish between sectors such that Eq. (G1) remains valid the a posteriori probability density fulfills  $P(e|y) \propto P(e)$  as long as  $\sum_i e_i$  is fixed. We thus obtain the posterior emissions of the sectors by maximizing [Eq. (G1)] with the constraint  $\sum_i e_i = e_{\text{tot}}^{\text{post}}$ .

**41. Appendix H (now Appendix B of Part 2):**

(a) L786-787: Referee comment: "'We first estimate the sensitivity of the posterior scaling factor to the true emissions under the assumption that the transport model, far field, and the flux pattern within each flux category are perfect.' Flux patterns are referred to, but not defined anywhere. I suspect addressing this point will help the reader understand point 7e above also."

We agree and gladly take up this suggestion to improve clarity: We now avoid the term "flux pattern" and adjusted this at multiple locations in the manuscript. The mentioned part now reads: "We first estimate the sensitivity of the posterior scaling factor to the true emissions under the assumption that the transport model, far field, observations, and the flux pattern a priori spatial distribution within each flux category are perfect."

(b) L788: Referee comment: " $\mu = y - Hs^{prior} - x^{ff}$ . Let  $y = Hs^{truth} + x^{ff} + \epsilon^y$  (where  $\epsilon^y$  is the observation error. Then,  $\mu = H(s^{prior} - s^{truth} + \epsilon^y)$ . The expression given in the paper

doesn't have  $\epsilon^y$ , which suggests that the authors are making the additional assumption that the observations are perfect."

Yes, we thank the referee for this correction. We now include the assumption of perfect observations in the explanation in the new version of the manuscript (see previous point 41a for the new text).

(c) Eq. (H1): Referee comment: "Given that the above expression is used to substitute for  $y-Hs-x^{ff}$  (and not  $y-Hs^{prior}-x^{ff}$  as used in point 41b above), why not simply present the previous expression with  $s^{prior} \rightarrow s$ ?"

Yes, the notation proposed by the referee  $(s^{\text{prior}} \to s)$  is better. We adapt this in the new version.

- 42. Appendix I (now Appendix D of Part 2):
  - (a) Fig. I1: Referee comment: "There is no explanation of the distinction between the two panels in the caption. The only difference is (a) is labeled Aemissions and (b) is labeled Ascaling factors. I assume that (a) is computed from (b)? What are the smaller matrices at the bottom of each panel?"

Fig. I1 (now Fig. D1 of Part 2) uses the same representation as Fig. 9 (now Fig. 7 of Part 2). We extended the explanation in the figure caption, but do not completely repeat the explanation of the representation from Fig. 9:

Averaging kernel matrices of German sector emissions (a) and the corresponding scaling factors (b) when trying to distinguish sectors waste, public power and other, estimated using the posterior error covariance matrix. Small matrices at the bottom indicate the ideal result. See Fig. 7 for an explanation of the representation. Panel (a), third row, shows that increasing true emissions in any sector is expected to cause higher posterior agriculture emissions with a false attribution of 46 % to 70 %. The same row in panel (b) shows that when looking at relative changes in the emissions, the influence of the false attribution on the agriculture sector is not very large.

**3 Presentational points and very minor points**

We thank the referee for correcting also typos and providing advices on making the text more understandable. Aside from typos and simple corrections, the referee mentioned the following points:

1. Referee comment: "The appendices are referenced in a different order to their placement. For example, appendix B is referenced first. It makes logical sense to place the appendices in the order that they are first referenced in the main text."

This is an important remark. In the restructuring we made sure that the appendices are ordered as recommended.

2. Table 1: Referee comment: "As there are many acronyms, the caption of Table 1 might be a good place to (re)define acronyms. Incidentally, I cannot see where GNFR is defined anywhere."

The acronyms used in Table 1 are partially product names (e.g., CAMS-REG or ICON), for which the acronym may be more common than the expanded name. For the abbreviation

"GNFR", which names a convention used in reporting, we added the information to the table caption that is most relevant to the reader: "The national reporting distinguishes emissions by GNFR sectors of which A–M include all anthropogenic emissions excluding land use, land use change and forestry (LULUCF)." We thereby also define "LULUCF" in the updated caption.

4. Sections 4.1.1 and 4.1.2: Referee comment: "I got very lost trying to follow exactly what was done here. I would recommend that these sections are rewritten, possibly with a table to help the reader see exactly how different regions and sectors are combined, etc."

As suggest, these sections have been rewritten and are now Sections 2.1.3 and 2.1.4 of Part 1. We added a short introduction with a simple example and a table of flux categories to Sect. 2.1.3 (old Sect. 4.1.1). In Sect. 2.1.4 (old Sect. 4.1.2), we have added an explanation how the a priori concentrations are computed. We have rewritten the most complicated part relating to flux categories.

15. L588: Referee comment: "'Moreover, adjusting fewer degrees of freedom may . . . '"

We thank the referee for this comment. The suggested correction does not perfectly fit what we meant here, but motivates the following clarification: "Moreover, adjusting only a few degrees of freedom may not be sufficient ..."

**4 Mistake in the manuscript**

Figures 10, F2 and E1 in the submitted manuscript were affected by a mistake: Those inversions for which the far-field correction was disabled used wrong tuning parameters with underestimated uncertainties due to a bug in the code. In sensitivity test 400 and Figure 10, this led to an overestimation of the relevance of the far-field correction. The mistake has been corrected and the discussion of the far-field correction has been adjusted accordingly. The conclusions of our manuscript remain unaffected.

(This paragraph is included in the replies to both referees.)

---

## Author Comment (AC2)

**Reply to Referee #2 of egusphere-2025-1464**

**Valentin Bruch**

October 23, 2025

We thank the referee for the detailed reading, positive assessment and valuable comments. Following a suggestion by the other referee and after consolidation with the editor, the manuscript has been split into a Part 1 and Part 2. In this new structure, we address the questions raised by the referee below.

The structure of this document follows the report of the referee. For transparency and convenience, we added Sect. 3 below to explain a mistake we noticed in the initially submitted manuscript during the review process.

**1 Comparison to "standard" R values**

Referee comment: "My main point is that I would like to see better visualised how the R matrix as made in this paper improves the inversion. The calculation is quite complex, and covers a large part of the paper. Nevertheless, some arbitrary choices remain (like the 10 ppb standard, and the inflation factor). I now wonder if 'standard' R values (like in Steiner et al., 2024b (reference as in manuscript): 'we use a model-data mismatch of 2 ppb + 40 % of the anthropogenic signal') would give similar results. [...]"

This is a very interesting question, which is now discussed in detail in Section 4 of the new Part 1. One main outcome of Part 1 is that the ensemble-based R matrix leads to a lower random error in the inversion results than the described "standard" R. But for comparing the different results, we choose a different focus than what the referee proposed:

Referee comment: "[...] For this, the  $\chi^2$ -innovation can be used.  $\chi^2$  is used to assess the uncertainty, relative to the model-observation mismatch (i.e.  $((y-Hx)^2)/HPH^T$ , where y are observations, Hx are transported (prior mean) fluxes and P is the prior covariance matrix of the fluxes). One would like the  $\chi^2$ -innovation to be as close as possible to 1.0 (which means uncertainty and model error are balanced). It would help the paper to include a figure of the  $\chi^2$ -innovation to show that the 'new' R is an improvement over e.g. Steiner et al., 2024b. A run with a fixed (diagonal) R can also be included in Figure 7."

We agree that comparing different methods for constructing the R matrix based on the  $\chi^2$  innovation can be very helpful. But as pointed out by the referee, our method contains many tuning parameters. A simple comparison of the  $\chi^2$  innovation, as suggested by the referee, would only highlight differences between two configurations of tuning parameters but provide little insight into the method itself. Since we are interested in the potential of the ensemble method, we need comparable configurations and a way of assessing the inversion quality.

In the submitted manuscript, we already considered two methods for constructing R (denoted prior R and posterior R). In Part 1 of the revised manuscript, we add the "diagonal R" as a simple approach as also proposed by the referee (now in Sect. 2.5.1). One main focus of Part 1 is the comparison of these three ways of constructing R (prior R, posterior R, and diagonal R). However,

we consider the  $\chi^2$  method as not suitable for comparing the prior R and posterior R inversion. In particular, these methods have a different bias, which is not detectable from the  $\chi^2$  innovation. We therefore focus on different ways of comparing the methods but also mention the  $\chi^2$  innovation (now in Sect. 2.6.5 of Part 1, see comment below).

We contrast the methods using synthetic experiments with a simulated transport uncertainty (Sect. 5.6 and Appendix F in the first version, now Sections 4.2–4.4 of Part 1). Using these synthetic experiments with a known truth, we can compare how well a method can estimate the synthetic truth from pseudo-observations and thereby assess the quality of the inversion. We find that the diagonal R approximation without any information from the ensemble leads to a significantly larger random error in the inversion results. These results are visualized in the new Figures 6 and 7 of Part 1, which replace Figures 10 and F2. To make the inversion results comparable, we choose the tuning parameters such that all methods have approximately equal posterior uncertainties.

Referee comment: "It would help the paper to include a figure of the  $\chi^2$ -innovation to show that the 'new' R is an improvement over e.g. Steiner et al., 2024b."

As mentioned above, we discuss the  $\chi^2$  innovation in Sect. 2.6.5 of Part 1. Instead of a figure, we chose a table (Table 2 of Part 1) as the best format to summarize  $\chi^2$  for the different methods for real and synthetic observations:

**Table 2.** Median of  $\chi^2/N_{\text{dof}}$  for different configurations.  $\chi^2/N_{\text{dof}}$  for the prior R inversion also serves as an approximation for the posterior R inversion. Synthetic observations are generated using the ensemble simulation, assuming that the a priori fluxes and the CH4 concentration on lateral boundaries are known exactly.

| observations | far-field correction | $\chi^2/N_{\rm dof}$ , diagonal $R$ | $\chi^2/N_{\rm dof}$ , prior $R$ |
|--------------|----------------------|-------------------------------------|----------------------------------|
| real         | yes                  | 0.18                                | 0.16                             |
| real         | no                   | 0.21                                | 0.18                             |
| synthetic    | yes                  | 0.05                                | 0.03                             |
| synthetic    | no                   | 0.06                                | 0.03                             |

We find that the simple diagonal R has a larger  $\chi^2$  than the prior R, while  $\chi^2$  remains well below 1.0 for all cases. This result is closely linked to the constraint that the posterior uncertainties shall be similar, on which we have based the tuning parameters for the diagonal R matrix. This result indicates that our ensemble-based approach would allow for tuning to smaller posterior uncertainties than the diagonal R, then leading to larger  $\chi^2$ . Nevertheless, this comparison does not imply the superiority of any particular method.

Referee comment: "A run with a fixed (diagonal) R can also be included in Figure 7." We thank the referee for this constructive comment. Instead of adding only the diagonal R run to Figure 7, we compare all three considered ways of constructing R in the newly added supplementary Figure A4 in Part 2:

**Figure A4.** Statistics of the relative (a) and absolute (b) improvement of the model—observation mismatch at independent validation stations for different choices of the error

covariance matrix R discussed in Part 1. The figure is analogous to Fig. 5 [previously Fig. 7], where the visualization and the data selection is explained. Here, we distinguish three inversion methods that differ in how R is constructed, as introduced in Sect. 2.5 of Part 1. No clear advantage of one method over the others can be seen. The diagonal R inversion has the lowest posterior RMSE at validation sites, followed by the posterior R and prior R inversion, but the differences are not statistically significant.

In the new structure, suggested by the other referee, the discussion of the method is separated from the discussion of the main results. Since the validation (Figure 7) belongs to the results in Part 2, we have included this as a supplementary figure of Part 2 and not in the main text.

**2 Other comments**

1. Referee comment: "Almost all IDs tested are within 15% in their flux-solution. To me, this seems like the posterior uncertainty is under-estimated. Can the authors comment on this?"

The sensitivity tests typically show variations within 15% of the posterior  $2\sigma$  uncertainty. This indicates that the uncertainty due to the choice of tuning parameters is small compared to the posterior uncertainty. Based on this finding, we do not try to include the uncertainty due to the tuning parameters when estimating the posterior uncertainties. We see no evidence that the posterior uncertainty might be considerably underestimated.

2. Referee comment: "Can the authors explain to me the strange boundaries in Fig. 4b and d? With this, I mean the darker lines that run through e.g. France."

When defining flux categories by area, we used a smooth transition at boundaries between different categories. For example, we split Poland into two flux categories (west and east), but some emissions in the center are assigned to 50% to western Poland and 50% to eastern Poland. This setup helps us reduce sharp spatial concentration gradients in our transport simulation. However, when assuming uncorrelated a priori uncertainties for eastern and western Poland, this implies that emissions that belong to both categories have a lower relative uncertainty. These boundaries between flux categories are therefore visible in the uncertainties. We added a brief explanation in the figure caption: "The smooth boundaries between two regions with separate scaling factors appear as darker lines because these scaling factors are assumed to be initially uncorrelated."

3. Referee comment: "in Section 5.5.1, a gaussian noise of 2 ppb random error is added to the pseudo-observations. However, the σconst is already 10 ppb, which means the added white noise is quite small compared to the uncertainty associated to these observations. Can the authors explain this choice?"

In the synthetic experiments with randomized true emissions, we work with idealized pseudo-observations that have an underestimated error. The simulation of a transport error for these pseudo-observations would require an impractically high computational effort. Thus, these pseudo-observations inevitably only include a very simplified error. But this merely impacts the analysis, since the focus of these experiments lies on testing how the full inversion system – including observation filtering and far-field correction – can determine the synthetic truth from idealized pseudo-observations.

In lack of a good estimate for the error on the pseudo-observations, we choose the values 2 ppb which is larger than the observation uncertainty (of usually < 1 ppb) but sufficiently small to

not impact the outcome of the synthetic experiments. To clarify this aspect, we have extended the explanation in Sect. 5.5.1 (now Sect. 3.4.1 of Part 2):

This construction of pseudo-observations clearly underestimates the true error in the model—observation comparison, but it allows us to test the interplay of farfield correction and inversion in a controlled setup. Synthetic experiments with a simulated transport uncertainty are discussed in Part 1.

In this work, we consider different types of synthetic experiments which either consider a simulated transport error or a random variation of the emissions. Combining both would be an interesting extension, but goes beyond the focus of the current study.

**3 Mistake in the manuscript**

Figures 10, F2 and E1 in the submitted manuscript were affected by a mistake: Those inversions for which the far-field correction was disabled used wrong tuning parameters with underestimated uncertainties due to a bug in the code. In sensitivity test 400 and Figure 10, this led to an overestimation of the relevance of the far-field correction. The mistake has been corrected and the discussion of the far-field correction has been adjusted accordingly. The conclusions of our manuscript remain unaffected.

(This paragraph is included in the replies to both referees.)